# Exact Rates and Saturation Effect of Kernel Ridge Regression over Unbounded Input Space in Large Dimensions

## Abstract

This work presents a theoretical analysis of Kernel Ridge Regression (KRR) in a large dimensional regime where both the input dimension $d$ and sample size $n$ grow, satisfying $n \asymp d^\gamma$ for some $\gamma > 0$. We extend prior studies, which focus on inner product kernels on the sphere $\mathbb{S}^{d-1}$, to broader classes of kernels defined on unbounded domains $\mathcal{X} \subseteq \mathbb{R}^d$. Suppose that the true function $f_\rho^* \in [\mathcal{H}]^s$, where $[\mathcal{H}]^s$ denotes the interpolation space of RKHS $\mathcal{H}$ with source condition $s > 0$. Our primary contribution is a precise characterization of the generalization error of KRR, treating the cases $s \geq 1$ and $s < 1$ separately.

Surprisingly, after adopting the result to the Gaussian kernel in large dimensions and deriving precise asymptotics for the corresponding eigenvalues, our analysis of Gaussian kernel ridge regression reveals that it: $i)$ achieves minimax optimality when $0 < s \leq 1$; $ii)$ fails to attain the minimax lower bound for $s > 1$, demonstrating a *saturation effect* where additional smoothness beyond this point does not improve the convergence rate. Furthermore, we identify two phenomena unique to the large dimensional regime: a *periodic plateau phenomenon* in the convergence rate and a *multiple-descent behavior* with respect to the sample size $n$.

## 1 Introduction

The recent success of deep learning has brought the traditional kernel methods (such as kernel ridge regression(KRR)) back to spotlight, particularly through the connection between wide neural networks and neural tangent kernels (NTKs) established by Jacot et al. (2018). Under the fixed dimensional setting, the behavior of KRR is well-characterized under standard assumptions by Caponnetto & De Vito (2007); Fischer & Steinwart (2020). Also, a line of work (Simon et al. (2023)) considered the eigenframework, deriving conservative equations to generalization metrics of KRR. However, modern applications increasingly involve large dimensional data, where the dimension $d$ ranges from thousands to millions. This shift towards large dimensional applications calls for a thorough understanding of how kernel methods perform when both the sample size $n$ and dimension $d$ grow, particularly in the large dimensional regime where $n \asymp d^\gamma$ for some $\gamma > 0$.

In large dimensions, classical analyses based on polynomial eigendecay and source conditions often break down, complicating the theoretical landscape. Several recent works have begun exploring this challenging setting. For instance, Ghorbani et al. (2021) identified approximation barriers for large-dimensional kernel regression, while Lu et al. (2023) and Zhang et al. (2024) derived minimax optimal convergence rates for inner product kernels on the sphere, revealing intriguing phenomena such as periodic plateau and multiple descent behaviors in the excess risk. Also, a benign overfitting behavior arises in the large dimensional setting, which has been discussed by Misiakiewicz (2022); Barzilai & Shamir (2023); Zhang et al. (2025).These findings suggest that large-dimensional kernel regression exhibits fundamentally different characteristics than its fixed-dimensional counterpart.

Despite these advances, existing studies under large dimensional setting mostly focus on bounded input space. In fact, current theoretical understanding is largely confined to inner product kernels defined on sphere, leaving unbounded domains, which are critical for widely used kernels like the Gaussian kernel, largely unexplored. In this paper, we bridge this gap by investigating the general-

ization error of KRR in large dimensional, general domains. We establish exact convergence rates without restrictive eigenfunction assumptions. By adopting the exact convergence rates to large dimensional Gaussian kernel, we demonstrate that phenomena previously observed for spherical kernels also manifest in Gaussian kernel settings.

## 1.1 CONTRIBUTIONS AND RELATED WORKS

**The exact rates of the generalization error of KRR with general input space** $\mathcal{X}$ In this paper, we perform an investigation on the kernel ridge regression in large dimensional setting (i.e., $n \propto d^{\gamma}$) where the kernel may defined on a (possibly) unbounded domains. In particular, we derive the exact rates of the generalization error of KRR for target functions $f_{\rho}^* \in [\mathcal{H}]^s$ in this setting.

Most recent work on large dimensional kernels focuses on inner product kernels on sphere (e.g.,Lu et al. (2023),Zhang et al. (2024)), with some studies (e.g., Misiakiewicz & Saeed (2024)) extending to other kernels defined on domains beyond the sphere. However, these studies universally impose explicit assumptions ( *e.g. hypercontractivity*) on the properties of the kernel's eigenfunctions (see Remark 3.8 for details). While such assumptions are known to hold in specific cases (such as kernels on the hypercube, kernels on the sphere, or settings where eigenfunctions are polynomials under Gaussian measure), their applicability to the most commonly used kernel, the Gaussian kernel on Gaussian measure, remains unverified. Recently, Pandit et al. (2024) considered large dimensional KRR on potentially unbounded domains under the setting $n \asymp d^2$, thereby partially addressing the problem.

In this paper, we establish the exact convergence rate without imposing any assumptions on eigenfunctions. We believe this result offers a more widely applicable and theoretically accessible framework for analyzing kernel ridge regression in broad settings.

**Minimax optimality and saturation Effect of large dimensional Gaussian kernel** Though whether the results is applicable for the Gaussian kernel remains unclear, our results can be easily verified for Gaussian kernel. Our analysis shows that when the source condition $s \leq 1$, the exact rates of the generalization error of large dimensional Gaussian kernel is in accordance with the minimax rate, indicating the minimax optimality of KRR in such regime. In contrast, when $s > 1$, a gap between the exact rates of the generalization error of large dimensional Gaussian kernel and the minimax rate exists, which is known as the saturation effect.

Under fixed dimension setting, the minimax optimality and saturation effect of KRR have been well studied. Fischer & Steinwart (2020); Zhang et al. (2023) derived the minimax optimal rate $n^{-\frac{s\beta}{s\beta+1}}$ of KRR when $0 < s \leq 2$; Bauer et al. (2007); Li et al. (2023a) reported the saturation effect when $s > 2$. However, these phenomena remain largely unexplored in the large dimensional regime. The recent work of Zhang et al. (2024) made progress by establishing minimax optimality for inner product kernels on the sphere when $s < 1$, and reporting the saturation effect when $s > 1$.

Going beyond inner product kernels on spheres, this paper provides the corresponding results for the Gaussian kernel in large dimensional setting. Our results suggest that saturation and optimality are not peculiar to spherical settings, but reflect broader behavior of KRR in large dimensions. We believe this finding significantly expands our understanding of kernel methods in modern large dimensional learning.

**Similar phenomena and differences between large dimensional Gaussian kernel and inner product kernels on sphere** Our analysis reveals that, similar to the inner product kernel regression (Ghorbani et al., 2021; Lu et al., 2023; Zhang et al., 2024), Gaussian kernel regression exhibits the similar *periodic plateau phenomenon* and *multiple descent behavior* (Fig. 1 and Fig. 2).

Although these phenomena are shared between large dimensional Gaussian kernels and inner product kernels on the sphere, we emphasize that the two kernel classes differ significantly in several key aspects. These include the asymptotic properties of high-order moments of the samples $x_i$, as well as the differences in their respective eigenfunctions (see Section A.1 for details). These distinctions make the analysis of large dimensional Gaussian kernels more challenging than that of inner product kernels on the sphere.

**Notations** Let $\mathcal{X} \subseteq \mathbb{R}^d$ be the input space, and $\mathcal{Y} \subseteq \mathbb{R}$ be the output space. Let $\rho = \rho_d$ be an unknown probability distribution on $\mathcal{X} \times \mathcal{Y}$ satisfying $\int_{\mathcal{X} \times \mathcal{Y}} y^2 \mathrm{d}\rho(x,y) < \infty$ and denote the

corresponding marginal distribution on $\mathcal{X}$ as $\mu = \mu_d$. For the sake of conciseness, we may use $L^2$ as the abbreviation of $L^2(\mathcal{X}, \mu)$, and $\|\cdot\|_{\mathcal{H}}, \|\cdot\|_{L^2}, \|\cdot\|_{L^\infty}, \|\cdot\|_2$ denotes the norm of Hilbert space $\mathcal{H}$, the norm of $L^2(\mathcal{X}, \mu)$, the norm of $L^\infty(\mathcal{X}, \mu)$, the Frobenius norm, respectively.

Throughout the paper, we shall use the asymptotic notations $o(\cdot), O(\cdot), \omega(\cdot), \Omega(\cdot)$. Also, we shall define the following notations: $a \lesssim b$ if and only if $a = O(b)$; $a \gtrsim b$ if and only if $b = O(a)$; $a \asymp b$ (or $a = \Theta(b)$) if and only if $a \lesssim b$ and $a \gtrsim b$. We may also define the notations $o_{\mathbb{P}}(\cdot), O_{\mathbb{P}}(\cdot)$ as follows. $a_n = o_{\mathbb{P}}(b_n)$ if and only if $a_n/b_n$ converges to 0 in probability; $a_n = O_{\mathbb{P}}(b_n)$ if and only if for any $\epsilon > 0$, there exist constants $C_\epsilon, N_\epsilon$ such that for all $n \geq N_\epsilon$, $P(|a_n| > C_\epsilon|b_n|) < \epsilon$.

## 2 PRELIMINARIES

Suppose that $\{(x_i, y_i)\}_{i=1}^n$ are i.i.d. sampled from the model $y = f_\rho^*(x) + \epsilon$, where $f_\rho^*(x)$ is the true function, while $\epsilon$ is the noise. Throughout the paper, we shall consider the large dimensional setting, where $n \asymp d^\gamma, \gamma > 0$. We consider the following assumptions.

**Assumption 2.1.** Suppose that $\mathcal{H} = \mathcal{H}_d$ is a separable RKHS on $\mathcal{X} \subset \mathbb{R}^d$ with respect to a continuous kernel function $k = k_d$ satisfying

$$\sup_{x \in \mathcal{X}} k_d(x, x) \leq \kappa^2,$$

where $\kappa$ is an absolute constant, which does not depend on the dimension $d$. In the remaining paper, we may omit the subscript $d$ without causing any ambiguity.

*Remark* 2.2. In most kernel related literature, (see e.g., Caponnetto (2006); Caponnetto & De Vito (2007); Raskutti et al. (2014); Beaglehole et al. (2023); Buchholz (2022); Lai et al. (2023); Li et al. (2023c)), the input space $\mathcal{X}$ is assumed to be bounded. However, such assumptions are not made in this paper. Actually, we shall consider the Gaussian kernel in Section 4, whose input space is $\mathbb{R}^d$. The relieved restriction on the input space $\mathcal{X}$ allows us to apply our results to more general kernels.

By the celebrated Mercer's theorem (see,e.g., Steinwart & Scovel (2012)), when Assumption 2.1 holds, the kernel $k$ satisfies $k(x, x') = \sum_{i \in N} \lambda_i e_i(x) e_i(x')$, where $N$ is an at most countable set, $\{\lambda_i, i = 0, 1, \cdots\}$ are the non-increasing eigenvalues of $k$, while $\{e_i(x), i = 0, 1, \cdots\}$ are the corresponding eigenfunctions.

Next, we shall introduce the interpolation space of $\mathcal{H}$. For any $s \geq 0$, the interpolation space $[\mathcal{H}]^s$ can be defined as

$$[\mathcal{H}]^s := \left\{ \sum_{i \in N} \lambda_i^{s/2} a_i e_i \mid \sum_{i \in N} a_i^2 < \infty \right\},$$

with the inner product deduced from $\langle \lambda_i^{s/2} e_i, \lambda_j^{s/2} e_j \rangle_{[\mathcal{H}]^s} = \delta_{ij}$. It is easy to show that $[\mathcal{H}]^s$ is also a separable Hilbert space. Moreover, if we assume $s = 1$ or $s = 0$, the interpolation space norm $\|\cdot\|_{[\mathcal{H}]^s}$ will be reduced to $\|\cdot\|_{\mathcal{H}}$ and $\|\cdot\|_{L^2}$ respectively.

Kernel Ridge Regression (KRR) aims at minimizing the quadratic cost with a penalty term. Specifically, the estimator $\hat{f}_\lambda$ of KRR is defined by the following equation:

$$\hat{f}_\lambda = \arg\min_{f \in \mathcal{H}} \left( \frac{1}{n} \sum_{i=1}^n (y_i - f(x_i))^2 + \lambda \|f\|_{\mathcal{H}}^2 \right), \tag{1}$$

where $\lambda > 0$ is the regularization parameter.

Denote the samples as $\boldsymbol{X} = (x_1, \cdots, x_n)$ and $\boldsymbol{y} = (y_1, \cdots, y_n)^\top$. The explicit expression of the KRR estimator $\hat{f}_\lambda$ is given by the representer theorem (see, e.g., Steinwart & Christmann 2008):

$$\hat{f}_\lambda(x) = \mathbb{K}(x, \boldsymbol{X})(\mathbb{K}(\boldsymbol{X}, \boldsymbol{X}) + n\lambda I)^{-1}\boldsymbol{y}, \tag{2}$$

where

$$\mathbb{K}(\boldsymbol{X}, \boldsymbol{X}) = (k(x_i, x_j))_{n \times n}, \quad \mathbb{K}(x, \boldsymbol{X}) = (k(x, x_1), \cdots, k(x, x_n)).$$

Throughout the paper, we are interested in the generalization error (excess risk) of $\hat{f}_\lambda$:

$$\mathbb{E}_{x \sim \mu} \left[ \left( \hat{f}_\lambda(x) - f_\rho^*(x) \right)^2 \right] = \left\| \hat{f}_\lambda - f_\rho^* \right\|_{L^2}^2, \tag{3}$$

and our aim is to derive the exact convergence rate of the generalization error of KRR.

## 3 MAIN RESULTS

In this section, we shall provide the exact convergence rate of the generalization error of $\hat{f}_\lambda$. Before giving the main result, we shall introduce the following quantities, which will be frequently used in the following paper.

**Definition 3.1.** Given $\{\lambda_i, i = 0, 1, \cdots\}$ are the non-increasing eigenvalues of $k$, denoting $f_\rho^* = \sum_{i=1}^\infty f_i e_i(x) \in L^2(\mathcal{X}, \mu)$, we shall define the following quantities.

$$\mathcal{N}_k(\lambda) := \sum_{i \in N} \left(\frac{\lambda_i}{\lambda + \lambda_i}\right)^k, \quad \mathcal{R}_2(\lambda) := \sum_{i \in N} \left(\frac{\lambda}{\lambda_i + \lambda} f_i\right)^2, \quad k = 1, 2. \tag{4}$$

Also, we need the following assumptions, which are often used in kernel-related literature.

**Assumption 3.2.** Suppose that the noise $\epsilon$ satisfies $\mathrm{Var}(\epsilon) = \sigma_\epsilon^2 < \infty$.

**Assumption 3.3.** Suppose that there exist $s$ and an absolute constant $R_1$ such that $f_\rho^*$ satisfies $\|f_\rho^*\|_{[\mathcal{H}]^s} \leq R_1$.

Now we are prepared to give one of the main results of this paper.

**Theorem 3.4.** *When $s \geq 1$, suppose that Assumption 2.1, 3.2, 3.3 hold, and there exist absolute constants $c_1, c_2$ such that $c_1 < \lambda_0 < c_2$ regardless of $d$. Furthermore, assume that the following conditions hold for some $\lambda = \lambda(d, n) \to 0$:*

$$\frac{\mathcal{N}_1(\lambda)}{n} \ln n = o(1); \quad \frac{1}{n\lambda} = o(1); \quad \mathcal{N}_2(\lambda) = \Omega(1); \quad n^{-1}\frac{1}{\lambda^2} \ln n = o(\mathcal{N}_2(\lambda)). \tag{5}$$

*Then we have the following equation holds:*

$$\mathbb{E}[\|\hat{f}_\lambda - f_\rho^*\|_{L^2}^2 | \boldsymbol{X}] = \Theta_\mathbb{P}\left(\frac{\sigma_\epsilon^2 \mathcal{N}_2(\lambda)}{n} + \mathcal{R}_2(\lambda)\right). \tag{6}$$

**Elaboration of equation 6** Theorem 3.4 provides the precise order of the generalization of KRR when $s \geq 1$. In equation 6, $\frac{\sigma_\epsilon^2 \mathcal{N}_2(\lambda)}{n}$ represents the variance term of the generalization error, while $\mathcal{R}_2(\lambda)$ represents the bias term. The tradeoff between the variance term and bias term urges us to choose appropriate $\lambda$. In the following paper, we shall derive the appropriate $\lambda$ when considering large dimensional Gaussian kernel (see Theorem 4.8, 4.10 for details).

**Theorem 3.5.** *When $0 < s < 1$, suppose that Assumption 2.1, 3.2, 3.3 holds. Assume that the following conditions hold for some $\lambda = \lambda(d, n) \to 0$:*

$$\frac{\mathcal{N}_1(\lambda)}{n} \ln n = o(1); \quad \frac{1}{n\lambda} = o(1); \quad \mathcal{N}_2(\lambda) = \Omega(1); \quad n^{-1}\frac{1}{\lambda^2} \ln n = o(\mathcal{N}_2(\lambda)). \tag{7}$$

*Furthermore, define $f_\lambda = \sum_{i \in N} \frac{\lambda_i}{\lambda_i + \lambda} f_i e_i$, if there exists $\epsilon > 0$ such that*

$$\frac{\sqrt{\frac{1}{\lambda}}(n^{\frac{1-s}{2}+\epsilon} + \|f_\lambda\|_{L^\infty})}{n} = o(\frac{1}{\sqrt{n}}\mathcal{N}_2(\lambda)^{\frac{1}{2}} + \mathcal{R}_2(\lambda)^{\frac{1}{2}}). \tag{8}$$

*Then we have the following equation holds:*

$$\mathbb{E}[\|\hat{f}_\lambda - f_\rho^*\|_{L^2}^2 | \boldsymbol{X}] = \Theta_\mathbb{P}\left(\frac{\sigma_\epsilon^2 \mathcal{N}_2(\lambda)}{n} + \mathcal{R}_2(\lambda)\right). \tag{9}$$

*Remark* 3.6. Notice that the most significant distinction between Theorem 3.5 and Theorem 3.4 lies in the inclusion of condition 8. This requirement arises because $f_\rho^*$ no longer belongs to $\mathcal{H}$ when $s < 1$, thereby complicating the bounding procedure of the bias term.

**An approach to bound** $\|f_\lambda\|_{L^\infty}$ Unlike Theorem 3.4, we notice that $\|f_\lambda\|_{L^\infty}$ in Theorem 3.5 depends on the eigenfunctions $\{e_i\}_{i \in N}$. Thankfully, we are able to provide an upper bound for $\|f_\lambda\|_{L^\infty}$. Notice that $f_\lambda \in \mathcal{H}$, we can bound $\|f_\lambda\|_{L^\infty}$ by $\|f_\lambda\|_\mathcal{H}$, while the latter one can be

expressed as $(\sum_{i=0}^{\infty} \frac{\lambda_i}{(\lambda_i+\lambda)^2} f_i^2)^{\frac{1}{2}}$, which is independent of eigenfunctions. (Notice that $\|f_\rho^*\|_{[\mathcal{H}]^s} \le R_1, (\sum_{i=0}^{\infty} \frac{\lambda_i}{(\lambda_i+\lambda)^2} f_i^2)^{\frac{1}{2}}$ can be further bounded by functions of $\{\lambda_i\}_{i\in N}$ and $\lambda$, for a detailed upper bound of $\|f_\lambda\|_{L^\infty}$, see Lemma A.14.) Hence, we are able to claim that the conditions in our Theorem 3.5 can be easily verified without properties of eigenfunctions.

*Remark* 3.7. We notice that Theorem 1 in Zhang et al. (2024) provides a result similar to our results Theorem 3.4,3.5. The most important difference is that in our assumptions and conditions, we deliberately avoid reliance on eigenfunctions. On the contrary, Theorem 1 in Zhang et al. (2024) highly relies on the properties of eigenfunctions, such as the quantity $\mathcal{M}_1(\lambda) = \operatorname{ess\,sup}_{x\in\mathcal{X}} \left| \sum_{i=1}^{\infty} \left( \frac{\lambda}{\lambda_i+\lambda} f_i e_i(x) \right) \right|$ and the Assumption 3 in their paper. Although Zhang et al. (2024) verified their assumptions and conditions for spherical harmonics as eigenfunctions, such constraints may not hold for other general kernels, especially for those kernels defined on a unbounded domain. Our theorems depend solely on the eigenvalues rather than the eigenfunctions, which allows for a simpler verification and broader applicability across different settings.

*Remark* 3.8. Recently, Misiakiewicz & Saeed (2024) proposed an approximation of the generalization error via a deterministic equivalent $R_n(\boldsymbol{\beta}_*, \lambda) = \frac{\lambda_*^2 \langle \boldsymbol{\beta}_*, (\boldsymbol{\Sigma}+\lambda_*)^{-2} \boldsymbol{\beta}_* \rangle + \sigma_\varepsilon^2}{1 - \frac{1}{n} \operatorname{Tr}(\boldsymbol{\Sigma}^2(\boldsymbol{\Sigma}+\lambda_*)^{-2})}$, which appears to be valid in the large dimensional setting. Here $\boldsymbol{\Sigma}$ denotes the covariance operator, $\beta^*$ denotes the corresponding coefficients of $f^*$ under the basis $\{e_i\}$'s, $\lambda_*$ satisfies $n - \frac{\lambda}{\lambda_*} = \operatorname{Tr}\left(\boldsymbol{\Sigma}(\boldsymbol{\Sigma}+\lambda_*)^{-1}\right)$. While this approach offers a valuable theoretical tool for analyzing generalization performance, its applicability relies critically on certain structural assumptions—notably, a hypercontractivity condition (outlined in their Assumptions 3 and 4). This condition requires uniform bounds on higher-order moments of the eigenfunctions $e_i$ relative to their second moments, which can be challenging to verify for many commonly used kernels.

# 4 APPLICATIONS TO LARGE DIMENSIONAL GAUSSIAN KERNEL

In this section, we shall apply Theorem 3.4, 3.5 to the case of large dimensional Gaussian kernel. First, we shall provide the definition of Gaussian kernel in the large dimensional setting.

**Definition 4.1** (large dimensional Gaussian kernel)**.** Assume $(x_1, \cdots, x_n)$ are i.i.d. sampled from the large dimensional Gaussian distribution $N(0, \sigma^2 I_d)$, and $k_d(x, x') = \exp(-\|x-x'\|_2^2/(2\ell^2 d))$, where $\sigma$ and $\ell$ are absolute constants.

*Remark* 4.2. We consider the Gaussian kernel so that the explicit expression of the Mercer's decomposition exists (see Proposition 4.4 for details.) Notice that most works analyzing the spectral algorithm in large dimensional settings consider an inner product kernel on the sphere (Liang et al., 2020; Ghorbani et al., 2021; Misiakiewicz, 2022; Xiao et al., 2022; Lu et al., 2023; Zhang et al., 2024, etc.). We consider the Gaussian kernel rather than the inner product kernel on sphere to emphasize that the input space $\mathcal{X}$ does not need to be compact in our paper. For other kernels with explicit Mercer's decomposition (such as the often concerned inner product kernels on sphere, and positive definite kernels on homogeneous spaces), there may also exist similar derivation in this section. Also, we shall provide an applicable kernel on anisotropic Gaussian distribution in Section A.7.

*Remark* 4.3. Notice that by the Law of Large Numbers, $\|x-x'\|_2^2$ diverges to $\infty$ when $d$ diverges to $\infty$. Hence, we add the term $\frac{1}{d}$ in the Gaussian kernel to prevent that the kernel $k$ converges to 0.

The following proposition provides the explicit form of the Mercer's decomposition of Gaussian kernel, which can be found in Rasmussen & Williams (2005), Section 4.3.

**Proposition 4.4.** *Denote* $a^{-1} = 4\sigma^2, b^{-1} = 2\ell^2 d, c = \sqrt{a^2+2ab}, A = a+b+c, B = b/A$. *Then, the Mercer's decomposition of large dimensional Gaussian kernel is given as follows.*

$$k_d(z, z') = \sum_{k=0}^{\infty} \mu_k \sum_{l=1}^{N(d,k)} Y_{k,l}(z) Y_{k,l}(z'),$$

*where the eigenvalues $\{\mu_k, k = 0, 1, \cdots\}$ are* $\mu_k = (2a/A)^{\frac{d}{2}} \cdot B^k$, $N(d, k) = \binom{d-1}{k+d-1}$, *and the eigenfunctions* $\{Y_{k,l}; l = 1, \cdots, N_{d,k}\}$ *are*

$$\{Y_{k,l}; l = 1, \cdots, N_{d,k}\} = \{\prod_{j=1}^{d} c_{i_j} \exp\left(-(c-a)^2 z_j^2\right) H_{i_j}(\sqrt{2c}z_j); i_1, \cdots, i_d \in \mathbb{N}, i_1 + \cdots + i_d = k\},$$

*Here* $H_k(x) = (-1)^k \exp(x^2) \frac{d^k}{dx^k} \exp(-x^2)$ *denotes the* $k^{th}$ *Hermite polynomial,* $z_j$ *denotes the* $j^{th}$ *component of* $z$, *and* $\{c_k\}_{k \geq 0}$ *are constants satisfying* $\|c_k \exp\left(-(c-a)^2 z_j^2\right) H_k(\sqrt{2c}z_j)\|_{L^2} = 1$.

Proposition 4.4 provides the explicit form of the eigenvalues, and hence we can derive the asymptotic properties of the spectrum when $d$ diverges to $\infty$.

In order to calculate the quantities in Theorem 3.4, 3.5, we need the following assumption.

**Assumption 4.5.** Denote $q$ as the smallest integer such that $q > \gamma$ and $\mu_q \neq 0$. Define $\mathcal{I}_{d,k}$ as the index set satisfying $\lambda_i \equiv \mu_k, i \in \mathcal{I}_{d,k}$. Further suppose that there exists an absolute constant $c_0 > 0$ such that for any $d$ and $k \in \{0, 1, \cdots, q\}$ with $\mu_k \neq 0$, we have

$$\sum_{i \in \mathcal{I}_{d,k}} \mu_k^{-s} f_i^2 \geq c_0. \tag{10}$$

*Remark* 4.6. Assumption 4.5 actually implies that $f_\rho^* \notin [\mathcal{H}]^t$ for any $t > s$. Similar assumptions have been adopted when the lower bound of generalization error needs to be derived in the fixed-dimensional setting, such as equation (8) in Cui et al. (2021) and Assumption 3 in Li et al. (2023b).

Next we shall provide an insightful but informal proposition on the asymptotic properties of the eigenvalues of large dimensional Gaussian kernel. The formal form is deferred to Appendix A.4.1.

**Proposition 4.7** (informal). *Recall the definitions of* $\mu_k, N(d, k)$ *in Proposition 4.4,* $\mathcal{N}_1(\lambda), \mathcal{N}_2(\lambda)$, $\mathcal{R}_2(\lambda)$ *in equation 4. Define* $\tilde{s} = \min\{s, 2\}$. *Given* $k$, *If we choose* $\lambda = d^{-l}$ *for some* $l > 0$, *denote* $p \leq l \leq p + 1$ *for some* $p \in \{0, 1, 2 \cdots\}$, *when* $d$ *is large enough, we have:*

$$\mu_k \asymp d^{-k}, \quad N(d, k) \asymp d^k,$$

$$\mathcal{N}_1(\lambda) = O\left(\lambda^{-1}\right), \quad \mathcal{N}_2(\lambda) = \Theta\left(d^p + \lambda^{-2}d^{-(p+1)}\right),$$

$$\mathcal{R}_2(\lambda) = \Theta(\lambda^2 d^{(2-\tilde{s})p} + d^{-(p+1)\tilde{s}}),$$

$$\|f_\lambda\|_{L^\infty} = O\left(d^{\frac{(1-s)p}{2}} + \lambda^{-1}d^{-\frac{(1+s)(p+1)}{2}}\right) (\text{when } s < 1).$$

Now we are prepared to provide the exact rates of the generalization error under the large dimensional Gaussian kernel case, which are direct applications of Theorem 3.4, 3.5.

**Theorem 4.8** ($s \geq 1$). *Let* $c_1 d^\gamma \leq n \leq c_2 d^\gamma$ *for some fixed* $\gamma > 0$ *and absolute constants* $c_1, c_2$. *Consider* $\mathcal{X} = \mathbb{R}^d$ *and the marginal distribution* $\mu$ *to be the Gaussian distribution* $N(0, \sigma^2 I_d)$. *Let* $k = k_d$ *be a Gaussian kernel defined in Definition 4.1. Define* $\tilde{s} = \min\{s, 2\}$, *when* $s \geq 1$, *we have:*

(i) *When* $\gamma \in (p + p\tilde{s}, \ p + p\tilde{s} + 1]$ *for some* $p \in \{0, 1, 2 \cdots\}$, *by choosing* $\lambda = d^{-\frac{\gamma + p - p\tilde{s}}{2}} \cdot \mathbf{1}_{p>0} + d^{-\frac{\gamma}{2}} \ln d \cdot \mathbf{1}_{p=0}$, *we have*

$$\mathbb{E}\left[\left\|\hat{f}_\lambda - f_\rho^*\right\|_{L^2}^2 \mid \boldsymbol{X}\right] = \begin{cases} \Theta_{\mathbb{P}}\left(d^{-\gamma} \ln^2 d\right) = \Theta_{\mathbb{P}}\left(n^{-1} \ln^2 n\right), & p = 0, \\ \Theta_{\mathbb{P}}\left(d^{-\gamma+p}\right) = \Theta_{\mathbb{P}}\left(n^{-1+\frac{p}{\gamma}}\right), & p > 0; \end{cases} \tag{11}$$

(ii) *When* $\gamma \in (p + p\tilde{s} + 1, \ p + p\tilde{s} + 2\tilde{s} - 1]$ *for some* $p \in \{0, 1, 2 \cdots\}$, *by choosing* $\lambda = d^{-\frac{\gamma + 3p - p\tilde{s} + 1}{4}}$, *we have*

$$\mathbb{E}\left[\left\|\hat{f}_\lambda - f_\rho^*\right\|_{L^2}^2 \mid \boldsymbol{X}\right] = \Theta_{\mathbb{P}}\left(d^{-\frac{\gamma - p + p\tilde{s} + 1}{2}}\right) = \Theta_{\mathbb{P}}\left(n^{-\frac{\gamma - p + p\tilde{s} + 1}{2\gamma}}\right); \tag{12}$$

(iii) *When* $\gamma \in (p + p\tilde{s} + 2\tilde{s} - 1, \ (p+1) + (p+1)\tilde{s}]$ *for some* $p \in \{0, 1, 2 \cdots\}$, *by choosing* $\lambda = d^{-\frac{\gamma + (p+1)(1-\tilde{s})}{2}}$, *we have*

$$\mathbb{E}\left[\left\|\hat{f}_\lambda - f_\rho^*\right\|_{L^2}^2 \mid \boldsymbol{X}\right] = \Theta_{\mathbb{P}}\left(d^{-(p+1)\tilde{s}}\right) = \Theta_{\mathbb{P}}\left(n^{-\frac{(p+1)\tilde{s}}{\gamma}}\right). \tag{13}$$

*Also, there exists no $\lambda$ whose convergence rate of the generalization error of KRR is faster than the above choice.*

*Remark* 4.9. In the case $0 < \gamma \leq 1$, we choose $\lambda = d^{-\gamma/2} \ln d$ rather than $d^{-\gamma/2}$ in order to meet the conditions in Theorem 3.4. Correspondingly, the convergence rate is slightly smaller than $d^{-\gamma+p}$, which is the case when $2p < \gamma \leq 2p + 1$, $p > 0$. However, we shall show in Theorem 4.13 that both cases are in accordance with the minimax rate of KRR in large dimensional setting. In the following context we may ignore $\ln^2 n$ without causing any ambiguity.

**Theorem 4.10** ($s < 1$). *Let $c_1 d^\gamma \leq n \leq c_2 d^\gamma$ for some fixed $\gamma > 0$ and absolute constants $c_1, c_2$. Consider $\mathcal{X} = \mathbb{R}^d$ and the marginal distribution $\mu$ to be the Gaussian distribution $N(0, \sigma^2 I_d)$. Let $k = k_d$ be a Gaussian kernel defined in Definition 4.1.When $s < 1$, we have:*

- *If $\frac{1}{2} < s < 1$:*

  (i) *When $\gamma \in (p + ps, \ p + ps + s]$ for some $p \in \mathbb{N}$, by choosing $\lambda = d^{-\frac{\gamma+p-ps}{2}} \cdot \mathbf{1}_{p>0} + d^{-\frac{\gamma}{2}} \ln d \cdot \mathbf{1}_{p=0}$, we have*

  $$\mathbb{E}\left[\left\|\hat{f}_\lambda - f_\rho^*\right\|_{L^2}^2 \ \Big| \ \boldsymbol{X}\right] = \begin{cases} \Theta_{\mathbb{P}}\left(d^{-\gamma} \ln^2 d\right) = \Theta_{\mathbb{P}}\left(n^{-1} \ln^2 n\right), & p = 0, \\ \Theta_{\mathbb{P}}\left(d^{-\gamma+p}\right) = \Theta_{\mathbb{P}}\left(n^{-1+\frac{p}{\gamma}}\right), & p > 0; \end{cases} \tag{14}$$

  (ii) *When $\gamma \in (p + ps + s, \ (p+1) + (p+1)s]$ for some $p \in \mathbb{N}$, by choosing $\lambda = d^{-\frac{2p+s}{2}}$, we have*

  $$\mathbb{E}\left[\left\|\hat{f}_\lambda - f_\rho^*\right\|_{L^2}^2 \ \Big| \ \boldsymbol{X}\right] = \Theta_{\mathbb{P}}\left(d^{-(p+1)s}\right) = \Theta_{\mathbb{P}}\left(n^{-\frac{(p+1)s}{\gamma}}\right); \tag{15}$$

- *If $0 < s \leq \frac{1}{2}$: we have the same convergence rates as the case $s \in (\frac{1}{2}, 1)$ when $\gamma > \frac{3s}{2(s+1)}$.*

*Remark* 4.11. Note that in Theorem 4.10, we do not claim that the $\lambda$ we choose is optimal. However, we shall show in Theorem 4.13 that the minimax lower bound is in accordance with the rate given in Theorem 4.10. Hence, the rates in Theorem 4.10 are the fastest convergence rates KRR can achieve.

*Remark* 4.12. In Theorem 4.10, we only prove the case $\gamma > \frac{3s}{2(s+1)}$ when $s \leq \frac{1}{2}$. However, notice that $\frac{3s}{2(s+1)} \leq \frac{1}{2}$ when $s \leq \frac{1}{2}$, we actually provide the exact convergence rate for all $n \gtrsim d^{\frac{1}{2}}$.

Next, we provide the minimax lower bound of Gaussian kernel under large dimensional setting.

**Theorem 4.13** (minimax lower bound). *Let $c_1 d^\gamma \leq n \leq c_2 d^\gamma$ for some fixed $\gamma > 0$ and absolute constants $c_1, c_2$. Consider $\mathcal{X} = \mathbb{R}^d$ and the marginal distribution $\mu$ to be the Gaussian distribution $N(0, \sigma^2 I_d)$. Let $k = k_d$ be a Gaussian kernel defined in Definition 4.1. Let $\mathcal{P}$ consist of all the distributions $\rho$ on $\mathcal{X} \times \mathcal{Y}$ such that Assumption 3.2, 3.3, 4.5 hold. Then we have:*

  (i) *When $\gamma \in (p + ps, p + ps + s]$ for some $p \in \mathbb{N}$, for any $\epsilon > 0$, there exist constants $\mathfrak{C}_1$ and $\mathfrak{C}$ only depending on $s, \epsilon, \gamma, \sigma_\epsilon, \kappa, c_1$ and $c_2$ such that for any $d \geq \mathfrak{C}$, we have:*

  $$\min_{\hat{f}} \max_{\rho \in \mathcal{P}} \mathbb{E}_{(\boldsymbol{X}, \boldsymbol{y}) \sim \rho^{\otimes n}} \left\|\hat{f} - f_\rho^*\right\|_{L^2}^2 \geq \mathfrak{C}_1 d^{-\gamma+p-\epsilon}; \tag{16}$$

  (ii) *When $\gamma \in (p + ps + s, (p+1) + (p+1)s]$ for some $p \in \mathbb{N}$, there exist constants $\mathfrak{C}_1$ and $\mathfrak{C}$ only depending on $s, \gamma, \sigma_\epsilon, \kappa, c_1$ and $c_2$ such that for any $d \geq \mathfrak{C}$, we have:*

  $$\min_{\hat{f}} \max_{\rho \in \mathcal{P}} \mathbb{E}_{(\boldsymbol{X}, \boldsymbol{y}) \sim \rho^{\otimes n}} \left\|\hat{f} - f_\rho^*\right\|_{L^2}^2 \geq \mathfrak{C}_1 d^{-(p+1)s}. \tag{17}$$

Figure 1 shows the theoretical rates of generalization error of KRR and the corresponding minimax rate for different $s$. From the figure we shall draw the following conclusions:

**Minimax optimality and saturation effect** From Figure 1, we conclude that:

- When $s \leq 1$, the convergence rate of generalization error of KRR reaches the minimax lower bound for most $(s, \gamma)$. (Notice that we only derive the convergence rate of the generalization error of KRR for $\gamma > \frac{3s}{2s+2}$ when $s \leq \frac{1}{2}$.) Hence, when $s \leq 1$, with a little abusement, we shall claim that KRR is minimax optimal.

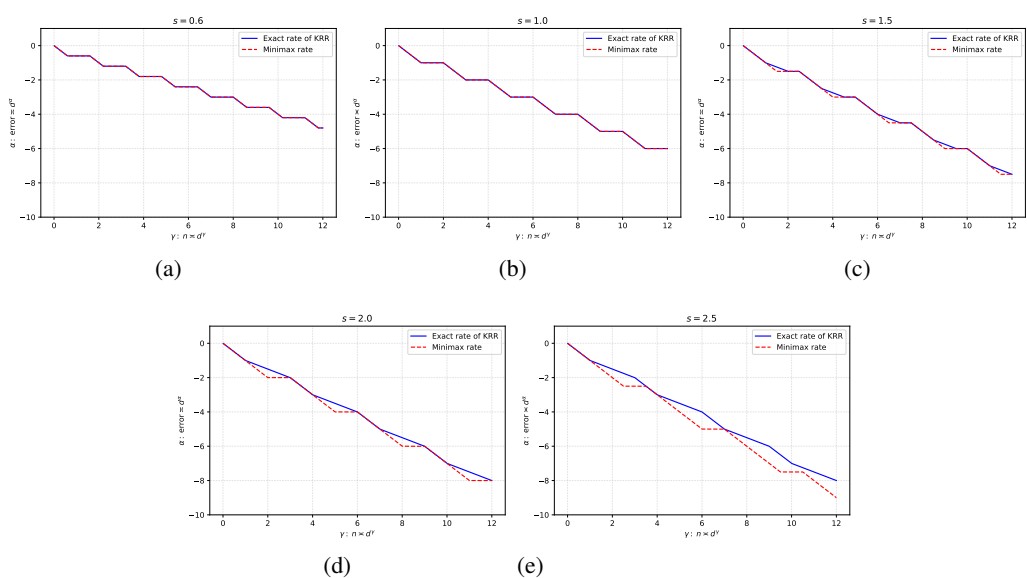

Figure 1: The exact rates of generalization error of KRR and the corresponding minimax rate with respect to $d$, $s = 0.6, 1, 1.5, 2, 2.5$.

- When $1 < s \leq 2$, the convergence rate of generalization error of KRR is in accordance with the minimax lower bound for certain ranges of $\gamma$ ($\gamma \in (p + ps, \ p + ps + 1] \cup (p + ps + 2s - 1, \ (p + 1) + (p + 1)s]$), while fails to reach the minimax rate for other $\gamma$. Such gap between the exact rate of KRR and the minimax rate is called the saturation effect, which was reported by Li et al. (2023a) when $s \geq 2$ in the fixed dimensional setting, Zhang et al. (2024) for inner product kernels on sphere when $s > 1$ in the large dimensional setting.

- When $s > 2$, we find that as $\gamma$ increases, the gap between the exact rate of KRR and the minimax rate tends to increase. This is due to the reason that the exact rate of KRR no longer changes when $s \geq 2$, while the minimax rate remain to decrease as $s$ increases. As a result, when $\gamma$ is sufficiently large, the convergence rate of generalization error of KRR shall never reach the minimax lower bound, hence we claim that the saturation effect occurs for most of the $\gamma$.

**Periodic plateau phenomenon** For the minimax rate, we observe a periodic plateau phenomenon under the large dimensional Gaussian kernel case: the rate of the excess risk remains unchanged over certain intervals of $\gamma$, and grows faster in other intervals of $\gamma$. Similar periodic plateau phenomenon was reported by Ghorbani et al. (2021); Lu et al. (2023); Zhang et al. (2024), when considering large dimensional spectral algorithms on the sphere.

For the exact convergence rate of generalization error of KRR, the periodic plateau phenomenon can also be observed when $0 < s < 2$. When $s \geq 2$, the plateau period degenerates.

Such periodic plateau phenomenon implies that in order to increase the convergence rate of the excess risk, it might be necessary to increase $\gamma$ (or equivalently, the sample size $n$) beyond a specific threshold.

We now present Figure 2, which illustrates the convergence rate of the large dimensional Gaussian kernel as a function of sample size $n$ (as opposed to dimension $d$). This figure enables us to draw the following conclusion.

**Multiple descent behavior** In Figure 2, we find that both the exact convergence rates and minimax rates experience vibration when $\gamma$ varies. For the minimax rate, the curve achieves its local maxima (the slowest rate) at $\gamma = p + ps, p \in \mathbb{N}^+$, and achieve its local minima (the fastest rate) at $\gamma = p + ps + s, p \in \mathbb{N}^+$. For the exact convergence rates of KRR, when $0 < s \leq 1$, the curve coincides with that of the minimax lower bound. When $1 < s < 2$, the curve achieves its local maxima at

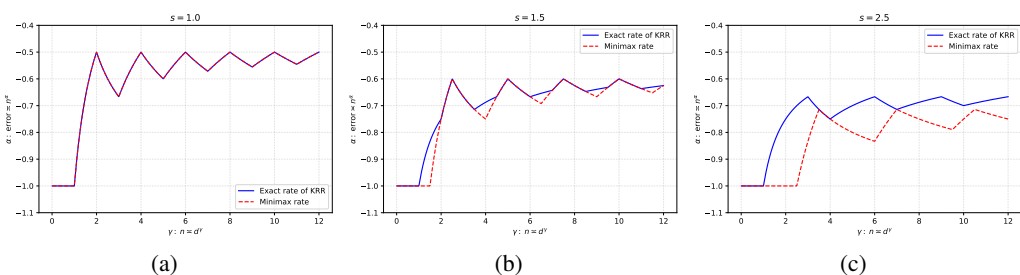

(a)  (b)  (c)

Figure 2: The exact rates of generalization error of KRR and the corresponding minimax rate with respect to $n$, $s = 1, 1.5, 2.5$.

$\gamma = p + p\tilde{s}, p \in \mathbb{N}^+$, and achieve its local minima at $\gamma = p + p\tilde{s} + 1, p \in \mathbb{N}^+$. When $s \geq 2$, the curve does not change with $s$, which achieves its local maxima at $\gamma = 3p, p \in \mathbb{N}^+$, and achieve its local minima at $\gamma = 3p + 1, p \in \mathbb{N}^+$. We call this vibration of the convergence rate the multiple descent behavior, which is also reported in Zhang et al. (2024); Lu et al. (2023). Generally speaking, we may conclude that when $\gamma$ increases, the convergence rate with respect to $n$ is slower, implying that increasing $n$ is less worthwhile when the sample size is already large enough.

## 5 CONCLUSION

In this paper, we consider KRR under the large dimensional setting. Unlike most large dimensional kernel related literature, we relieve the constraint that the input space $\mathcal{X}$ is compact, and provide an exact rate of the generalization error. Furthermore, we consider the large dimensional Gaussian kernel as an example, which is not applicable in the previous work, either due to the bounded input space constraint (Zhang et al., 2024) or a hypercontractivity condition (Misiakiewicz & Saeed, 2024). As a result, we prove the minimax optimality of large dimensional Gaussian KRR when $s \leq 1$, and report a saturation effect when $s > 1$. Furthermore, we identify phenomena such as *periodic plateau phenomenon* and *multiple descent behavior*, similar to those found in previous studies by Lu et al. (2023); Zhang et al. (2024), which considered the inner product kernel on the sphere. This similarity suggests that these phenomena may be general to large dimensional kernel methods, and can provide insights into the trade-off between sample size $n$ and convergence rates in practical applications.

Future work may extend this analysis to other kernel related algorithms under the large dimensional setting with unbounded input space. We believe such researches would significantly advance the understanding of kernel methods in the large dimensional setting, generalizing the restrictive case of inner product kernels on the sphere to a broader class of kernel functions. Recently, Li & Lin (2025) considers a diagonal adaptive kernel model under the fixed dimension setting. They claim that by introducing over-parametrization to the learning of kernels, the model can significantly improve its generalization properties compared with the fixed-kernel method. It would be of great interest to determine whether such phenomenon exists in the large dimensional setting for KRR. Understanding the underlying mechanism of how over-parameterization affect the learning of adaptive kernels in large dimensional setting could shed light on the potential advantages of adaptive kernels in large dimensional environments.

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

# A  APPENDIX

LLM is used in this paper at the sentence level.

## A.1  DIFFERENCES BETWEEN LARGE DIMENSIONAL GAUSSIAN KERNELS AND INNER PRODUCT KERNELS ON SPHERE

In this section, we shall point out the differences between large dimensional Gaussian kernels and inner product kernels on sphere, and hence emphasizing the necessity of separate theoretical treatment for Gaussian kernels. The large dimension Gaussian kernel has been defined in Definition 4.1, while the inner product kernels on sphere $k^{\mathrm{in}} : \mathbb{S}^{d-1} \times \mathbb{S}^{d-1} \to \mathbb{R}$ satisfies $k^{\mathrm{in}}(x,y) = \Phi(\langle x,y \rangle)$, where $\Phi : [-1,1] \to \mathbb{R}$.

**The asymptotic properties of high order moments** Assume that $X \sim N(0, \sigma^2 I_d)$. Although the Central Limit Theorem implies that $\|X\|_2 / \sqrt{d}$ converges to $\sigma$ in distribution, such observation does not imply the similarity between large dimensional Gaussian kernel and inner product kernel on sphere. In fact, the asymptotic properties of the high-order moments of $X$ differ fundamentally from those of random vectors distributed uniformly on the sphere. It is easy to show that when $d$ diverges to $\infty$,

$$\mathbb{E}\left[\left\|\frac{X}{\sigma\sqrt{d}}\right\|_2^{d^\gamma}\right] \longrightarrow \begin{cases} 1, & \gamma < \frac{1}{2} \\ e^{\frac{1}{4}}, & \gamma = \frac{1}{2} \\ \infty, & \gamma > \frac{1}{2} \end{cases},$$

whose proof is deferred to Appendix A.8. Hence, given $d$, when $r = \Omega(d^{\frac{1}{2}})$, $\mathbb{E}\left[\left\|\frac{X}{\sigma\sqrt{d}}\right\|_2^r\right]$ may diverge to infinity, implying that a uniform bound on the moments of $X$ does not exist. However, when $X$ is distributed on the sphere, we have $\mathbb{E}\|X\|_2^r = 1$ for all $d, r$. The divergence in moment behavior

underscores fundamental differences in the structure of data distributions between unbounded Gaussian models and spherical settings. These differences suggest that the large dimensional Gaussian kernel may exhibit unique statistical behaviors not shared by inner product kernels on the sphere, necessitating a separate theoretical treatment.

**The difference on eigenfunctions** By Proposition 4.4, the eigenfunctions of large dimensional Gaussian kernels can be induced from Hermite polynomials, while those of inner product kernels on sphere are spherical harmonic polynomials. An important difference lies in the domain of eigenfunctions. For large dimensional Gaussian kernels, the domain is $\mathbb{R}^d$, while for inner product kernels on sphere, the domain is $\mathbb{S}^{d-1}$. Due to such difference, results for inner product kernels on sphere are not applicable for large dimensional Gaussian kernels (see Remark 3.7,3.8 for details).

**Structural Differences Between the RKHSs** Another important difference lies in the structural properties of the RKHS. The RKHS of Gaussian kernel possesses a nested property, which can be described as follows. Let $\{i_1, \cdots, i_p\}$ be a subset of $\{1, 2, \cdots, d\}$, and for any $u = (u_1, \cdots, u_d)$, define its projection as $u' = (u_{i_1}, \cdots, u_{i_p})$. Consider a Gaussian kernel

$$k(u, v) = \exp\left(-\frac{\|u - v\|_2^2}{2\ell^2 d}\right).$$

By Proposition 4.4, if we set $u_i = 0$ for all $i \notin \{i_1, i_2, \ldots, i_p\}$ for every element in the RKHS of $k$, the resulting RKHS naturally coincides with that of a lower-dimensional Gaussian kernel

$$k(u', v') = \exp\left(-\frac{\|u' - v'\|_2^2}{2\ell'^2 p}\right)$$

with an appropriate $\ell'$.

This nested structure implies that if the target function $f_\rho^*$ possesses a low dimensional structure, i.e., $f_\rho^*(u_1, \cdots, u_d) = g(u_{i_1}, \cdots, u_{i_p})$, a low dimensional Gaussian kernel can be used to effectively approximate $f_\rho^*$. In contrast, the RKHS of inner-product kernels on the sphere generally lacks such nested RKHS properties. Therefore, a thorough understanding of large-dimensional Gaussian kernels is essential for analyzing such problems involving low-dimensional structure and sparsity.

### A.2 PROOF OF THEOREM 3.4

The proof of Theorem 3.4 relies on the classical bias-variance decomposition framework. We begin by establishing key notations and definitions.

Denote $T_\lambda = T + \lambda$ and $T_{\mathbf{X}\lambda} = T_{\mathbf{X}} + \lambda$, where $\lambda > 0$ is the regularization parameter. Define $S_k$ as the embedding operator $S_k : \mathcal{H} \to L^2(\mathcal{X}, \mu)$, and its adjoint operator as $S_k^* : L^2(\mathcal{X}, \mu) \to \mathcal{H}$, satisfying

$$(S_k^* f)(x) = \int_{\mathcal{X}} k(x, x') f(x') \, d\mu(x').$$

For operator norms, we rewrite $L^2(\mathcal{X}, \mu)$ as $L^2$ and $L^\infty(\mathcal{X}, \mu)$ as $L^\infty$ for brevity, and use the following conventions:

(i) $\|\cdot\|_{\mathcal{B}(B_1, B_2)}$ denotes the operator norm between Banach spaces $B_1$ and $B_2$, and is defined as $\|A\|_{\mathcal{B}(B_1, B_2)} = \sup_{\|f\|_{B_1}=1} \|Af\|_{B_2}$. When the context is clear, we write $\|\cdot\|$ for the operator norm.

(ii) $\operatorname{tr} A$ and $\|A\|_1$ denote the trace and trace norm of operator $A$, while $\|A\|_2$ represents the Hilbert-Schmidt norm.

Next, we shall introduce the following operators. Given the sample set $\mathbf{Z} = \{(x_i, y_i)\}_{i=1}^n$, we define the sampling operator $K_x : \mathbb{R} \to \mathcal{H}, y \mapsto yk(x, \cdot)$. Its adjoint operator $K_x^* : \mathcal{H} \to \mathbb{R}$ satisfies $K_x^* f = f(x)$. Also, we define the operator $T_x = K_x K_x^*$, and the empirical covariance operator is defined as:

$$T_{\mathbf{X}} := \frac{1}{n} \sum_{i=1}^n K_{x_i} K_{x_i}^*$$

Note that $\|T_{\mathbf{X}}\| \leq \|T_{\mathbf{X}}\|_1 \leq \kappa^2$, and $T_{\mathbf{X}}$ is a compact trace-class operator.

Further, if we define the sample basis function as:

$$g_\mathbf{Z} := \frac{1}{n} \sum_{i=1}^n K_{x_i} y_i \in \mathcal{H},$$

following Caponnetto & De Vito (2007), the KRR estimator has the operator representation:

$$\hat{f}_\lambda = (T_\mathbf{X} + \lambda)^{-1} g_\mathbf{Z}.$$

To analyze the bias term, we introduce conditional expectations:

$$\tilde{g}_\mathbf{Z} := \mathbb{E}(g_\mathbf{Z}|\mathbf{X}) = \frac{1}{n} \sum_{i=1}^n K_{x_i} f_\rho^*(x_i),$$

$$\tilde{f}_\lambda := \mathbb{E}(\hat{f}_\lambda|\mathbf{X}) = (T_\mathbf{X} + \lambda)^{-1} \tilde{g}_\mathbf{Z}.$$

Also, the expectation of $g_\mathbf{Z}$ can be defined as:

$$g := \mathbb{E}g_\mathbf{Z} = \int_\mathcal{X} k(x, \cdot) f_\rho^*(x) d\mu(x) = S_k^* f_\rho^*,$$

and

$$f_\lambda = (T + \lambda)^{-1} g = (T + \lambda)^{-1} S_k^* f_\rho^*.$$

Now we are ready to give the bias-variance decomposition. The estimation error decomposes as:

$$\hat{f}_\lambda - f_\rho^* = \frac{1}{n} (T_\mathbf{X} + \lambda)^{-1} \sum_{i=1}^n K_{x_i} y_i - f_\rho^*$$

$$= (T_\mathbf{X} + \lambda)^{-1} \left[ \frac{1}{n} \sum_{i=1}^n K_{x_i} (f_\rho^*(x_i) + \epsilon_i) \right] - f_\rho^*$$

$$= (T_\mathbf{X} + \lambda)^{-1} \tilde{g}_\mathbf{Z} + \frac{1}{n} \sum_{i=1}^n (T_\mathbf{X} + \lambda)^{-1} K_{x_i} \epsilon_i - f_\rho^*$$

$$= (\tilde{f}_\lambda - f_\rho^*) + \frac{1}{n} \sum_{i=1}^n (T_\mathbf{X} + \lambda)^{-1} K_{x_i} \epsilon_i$$

Taking conditional expectation with respect to the noise $\epsilon$ given $\mathbf{X}$, and noting that $\{\epsilon_i|x_i\}$ are independent with $\mathbb{E}[\epsilon_i] = 0$ and $\text{Var}(\epsilon_i) = \sigma_\epsilon^2$, we obtain:

$$\mathbb{E}\left[ \|\hat{f}_\lambda - f_\rho^*\|_{L^2}^2 \big| \mathbf{X} \right] = \text{Bias}^2(\lambda) + \text{Var}(\lambda)$$

where

$$\text{Bias}^2(\lambda) = \|\tilde{f}_\lambda - f_\rho^*\|_{L^2}^2, \text{Var}(\lambda) = \frac{\sigma_\epsilon^2}{n^2} \sum_{i=1}^n \|(T_\mathbf{X} + \lambda)^{-1} k(x_i, \cdot)\|_{L^2}^2 \tag{18}$$

The subsequent analysis will establish upper bounds for both $\text{Bias}^2(\lambda)$ and $\text{Var}(\lambda)$.

### A.2.1 VARIANCE TERM

In this subsection, we aim at deriving Theorem A.6, which establishes the upper bound of the variance term. We shall proceed through several preparatory steps. First, we define the sample subspace

$$\mathcal{H}_n = \text{span}\{k(x_1, \cdot), \ldots, k(x_n, \cdot)\} \subset \mathcal{H}.$$

Also, recall the notation for kernel matrices:

$$\mathbb{K}(\mathbf{X}, \mathbf{X}) = (k(x_i, x_j))_{n \times n}, \quad \mathbb{K}(\mathbf{X}, \cdot) = \{k(x_1, \cdot), \ldots, k(x_n, \cdot)\},$$

and define the normalized sample kernel matrix as

$$K = \frac{1}{n}\mathbb{K}(\mathbf{X}, \mathbf{X}).$$

Notice that $\mathrm{Ran}\,(T_{\mathbf{X}}) = \mathcal{H}_n$, and $K$ represents $T_{\mathbf{X}}$ under the basis $\{k\,(x_1, \cdot)\,, \ldots, k\,(x_n, \cdot)\}$. Hence, for any continuous function $\varphi$, the following equation holds:

$$\varphi\,(T_{\mathbf{X}})\,\mathbb{K}(\mathbf{X}, \cdot) = \varphi(K)\mathbb{K}(\mathbf{X}, \cdot). \tag{19}$$

Consider the inner product elementwise between 19 and $f$, we get

$$(\varphi\,(T_{\mathbf{X}})\,f)\,[\mathbf{X}] = \varphi(K)f[\mathbf{X}]. \tag{20}$$

Also, we shall define the following inner product:

$$\langle f, g\rangle_{L^2, n} = \frac{1}{n}\sum_{i=1}^{n} f\,(x_i)\,g\,(x_i) = \frac{1}{n}f[\mathbf{X}]^{\top}g[\mathbf{X}].$$

Throughout the proof, we shall keep the abbreviation $k_x(\cdot) = k(x, \cdot)$ for brevity. The following lemma provides a transformation of the variation term.

**Lemma A.1.** *The variance term satisfies:*

$$\mathrm{Var}(\lambda) = \frac{\sigma_\epsilon^2}{n}\int_{\mathcal{X}}\left\|(T_{\mathbf{X}} + \lambda)^{-1}\,k_x(\cdot)\right\|_{L^2, n}^2\,d\mu(x).$$

*Proof.* The original derivation structure is maintained but slightly reorganized:

$$\begin{aligned}
\mathrm{Var}(\lambda) &= \frac{\sigma_\epsilon^2}{n^2}\sum_{i=1}^{n}\left\|(T_{\mathbf{X}} + \lambda)^{-1}\,k\,(x_i, \cdot)\right\|_{L^2}^2 \\
&= \frac{\sigma_\epsilon^2}{n^2}\left\|(T_{\mathbf{X}} + \lambda)^{-1}\mathbb{K}(\mathbf{X}, \cdot)\right\|_{L^2(\mathbb{R}^n)}^2 \\
&= \frac{\sigma_\epsilon^2}{n^2}\left\|(K + \lambda)^{-1}\mathbb{K}(\mathbf{X}, \cdot)\right\|_{L^2(\mathbb{R}^n)}^2 \quad \text{(by equation 19)} \\
&= \frac{\sigma_\epsilon^2}{n^2}\int_{\mathcal{X}}\mathbb{K}(x, \mathbf{X})(K + \lambda)^{-2}\mathbb{K}(\mathbf{X}, x)d\mu(x).
\end{aligned}$$

Using equation 20, noticing that $k_x[\mathbf{X}] = \mathbb{K}(\mathbf{X}, x)$, we get:

$$\left((T_{\mathbf{X}} + \lambda)^{-1}\,k_x\right)[\mathbf{X}] = (K + \lambda)^{-1}\mathbb{K}(\mathbf{X}, x),$$

and hence

$$\begin{aligned}
\frac{1}{n}\mathbb{K}(x, \mathbf{X})(K + \lambda)^{-2}\mathbb{K}(\mathbf{X}, x) &= \frac{1}{n}\left\|(K + \lambda)^{-1}\mathbb{K}(\mathbf{X}, x)\right\|_{\mathbb{R}^n}^2 \\
&= \left\|(T_{\mathbf{X}} + \lambda)^{-1}\,k_x\right\|_{L^2, n}^2.
\end{aligned}$$

Thus we conclude:

$$\mathrm{Var}(\lambda) = \frac{\sigma_\epsilon^2}{n}\int_{\mathcal{X}}\left\|(T_{\mathbf{X}} + \lambda)^{-1}\,k_x(\cdot)\right\|_{L^2, n}^2\,d\mu(x).$$

$\square$

In the remaining part of this subsection, we shall derive the following two approximations.

$$\begin{aligned}
\int_{\mathcal{X}}\left\|(T_{\mathbf{X}} + \lambda)^{-1}k_x\right\|_{L^2, n}^2\,d\mu(x) &= \int_{\mathcal{X}}\left\|T_{\mathbf{X}\lambda}^{-1}k_x\right\|_{L^2, n}^2\,d\mu(x) \\
&\stackrel{A}{\approx}\int_{\mathcal{X}}\left\|T_\lambda^{-1}k_x\right\|_{L^2, n}^2\,d\mu(x) \stackrel{B}{\approx}\int_{\mathcal{X}}\left\|T_\lambda^{-1}k_x\right\|_{L^2}^2\,d\mu(x).
\end{aligned}$$

### A.2.2 APPROXIMATION B

**Lemma A.2** (Approximation B). *Suppose that Assumption 2.1, 3.2, 3.3 hold. Furthermore, suppose that*

$$\frac{\sup_{i \in N} \frac{\lambda_i}{(\lambda_i + \lambda)^2}}{n} = o(\mathcal{N}_2(\lambda)),$$

*in which $\lambda = \lambda(d, n) \to 0$. Then for any fixed $\delta \in (0, 1)$, when $n$ is sufficiently large, with probability at least $1 - \delta$, we have*

$$\frac{1}{2} \int_{\mathcal{X}} \left\| T_\lambda^{-1} k_x \right\|_{L^2}^2 \, d\mu(x) - \Delta_2 \leq \int_{\mathcal{X}} \left\| T_\lambda^{-1} k_x \right\|_{L^2, n}^2 \, d\mu(x) \leq \frac{3}{2} \int_{\mathcal{X}} \left\| T_\lambda^{-1} k_z \right\|_{L^2}^2 \, d\mu(x) + \Delta_2$$

*where*

$$\Delta_2 = \frac{5\kappa^2 \sup_{i \in N} \frac{\lambda_i}{(\lambda_i + \lambda)^2}}{3n} \ln \frac{2}{\delta}$$

*Proof.* We define the function $f(z)$ as follows:

$$\begin{aligned}
f(z) &= \int_{\mathcal{X}} \left( T_\lambda^{-1} k_x(z) \right)^2 d\mu(x) \\
&= \int_{\mathcal{X}} \sum_{i=1}^{\infty} \left( \frac{\lambda_i}{\lambda_i + \lambda} \right)^2 e_i^2(x) e_i^2(z) d\mu(x) \\
&= \sum_{i=1}^{\infty} \left( \frac{\lambda_i}{\lambda_i + \lambda} \right)^2 e_i^2(z)
\end{aligned} \tag{21}$$

The norms of $f$ satisfy:

$$\begin{aligned}
\|f\|_{L^\infty} &\leq \sup_{i \in \mathbb{N}} \frac{\lambda_i}{(\lambda_i + \lambda)^2} \| \sum_{i=1}^{\infty} \lambda_i e_i^2(x) \|_{L^\infty} \leq \kappa^2 \sup_{i \in \mathbb{N}} \frac{\lambda_i}{(\lambda_i + \lambda)^2} \\
\|f\|_{L^1} &= \sum_{i=1}^{\infty} \left( \frac{\lambda_i}{\lambda_i + \lambda} \right)^2 = \mathcal{N}_2(\lambda)
\end{aligned} \tag{22}$$

Applying Proposition A.23 to $\sqrt{f}$ and noting that:

$$\|\sqrt{f}\|_{L^\infty} = \sqrt{\|f\|_{L^\infty}} = \left( \kappa^2 \sup_{i \in \mathbb{N}} \frac{\lambda_i}{(\lambda_i + \lambda)^2} \right)^{\frac{1}{2}}$$

we obtain with probability at least $1 - \delta$:

$$\frac{1}{2} \|\sqrt{f}\|_{L^2}^2 - \frac{5\kappa^2 \sup_{i \in \mathbb{N}} \frac{\lambda_i}{(\lambda_i + \lambda)^2}}{3n} \ln \frac{2}{\delta} \leq \|\sqrt{f}\|_{L^2, n}^2 \leq \frac{3}{2} \|\sqrt{f}\|_{L^2}^2 + \frac{5\kappa^2 \sup_{i \in \mathbb{N}} \frac{\lambda_i}{(\lambda_i + \lambda)^2}}{3n} \ln \frac{2}{\delta} \tag{23}$$

The empirical norm can be expressed as:

$$\begin{aligned}
\|\sqrt{f}\|_{L^2, n}^2 &= \int_{\mathcal{X}} f(y) dP_n(y) \\
&= \int_{\mathcal{X}} \left[ \int_{\mathcal{X}} \left( T_\lambda^{-1} k_x(y) \right)^2 d\mu(x) \right] dP_n(y) \\
&= \int_{\mathcal{X}} \left[ \int_{\mathcal{X}} \left( T_\lambda^{-1} k_x(y) \right)^2 dP_n(y) \right] d\mu(x) \\
&= \int_{\mathcal{X}} \left\| T_\lambda^{-1} k_x \right\|_{L^2, n}^2 d\mu(x)
\end{aligned} \tag{24}$$

Similarly, the $L^2$ norm satisfies:

$$
\begin{aligned}
\|\sqrt{f}\|_{L^2}^2 &= \int_{\mathcal{X}} f(z) d\mu(z) \\
&= \int_{\mathcal{X}} \left[ \int_{\mathcal{X}} \left( T_\lambda^{-1} k_x(z) \right)^2 d\mu(x) \right] d\mu(z) \\
&= \int_{\mathcal{X}} \left\| T_\lambda^{-1} k_x \right\|_{L^2}^2 d\mu(x)
\end{aligned}
\tag{25}
$$

These equalities establish the desired results.

$\square$

### A.2.3 APPROXIMATION A

The proof of Approximation A relies on the following key observation by Li et al. (2023a):

**Proposition A.3.** *For any $f, g \in \mathcal{H}$, we have*

$$
\langle f, g \rangle_{L^2, n} = \langle T_{\mathbf{X}} f, g \rangle_{\mathcal{H}} = \langle T_{\mathbf{X}}^{1/2} f, T_{\mathbf{X}}^{1/2} g \rangle_{\mathcal{H}}.
\tag{26}
$$

*Proof.* Since $T_{\mathbf{X}} f = \frac{1}{n} \sum_{i=1}^n f(x_i) k(x_i, \cdot)$, we have:

$$
\begin{aligned}
\langle T_{\mathbf{X}} f, g \rangle_{\mathcal{H}} &= \frac{1}{n} \sum_{i=1}^n f(x_i) \langle k(x_i, \cdot), g \rangle_{\mathcal{H}} \\
&= \frac{1}{n} \sum_{i=1}^n f(x_i) g(x_i) \\
&= \langle f, g \rangle_{L^2, n}.
\end{aligned}
$$

The second equality follows from the definition of $T_{\mathbf{X}}^{1/2}$. $\square$

**Lemma A.4** (Approximation A). *Suppose that Assumption 2.1, 3.2, 3.3 hold. Define $\Delta_1 = \Delta_1(\lambda, X)$ as a function of $\lambda$ and $X$:*

$$
\Delta_1 := \left| \int_{\mathcal{X}} \left\| T_{\mathbf{X}\lambda}^{-1} k_x \right\|_{L^2, n}^2 d\mu(x) - \int_{\mathcal{X}} \left\| T_\lambda^{-1} k_x \right\|_{L^2, n}^2 d\mu(x) \right|.
\tag{27}
$$

*Suppose that $\lambda = \lambda(d, n)$ satisfies $\frac{\mathcal{N}_1(\lambda)}{n} \ln n = o(1)$, $\frac{\sup_{i \in N} \frac{\lambda_i}{(\lambda_i + \lambda)^2}}{n} = o(\mathcal{N}_2(\lambda))$. Then for any fixed $\delta \in (0, 1)$, when $n$ is sufficiently large, with probability at least $1 - \delta$, we have*

$$
\Delta_1 \leq 36 n^{-1} \mathcal{N}_1(\lambda)^2 \ln n + 12 n^{-\frac{1}{2}} \mathcal{N}_1(\lambda) \mathcal{N}_2(\lambda)^{\frac{1}{2}} (\ln n)^{\frac{1}{2}}.
\tag{28}
$$

*Proof.* We decompose $\Delta_1$ as:

$$
\begin{aligned}
\Delta_1 &= \left| \int_{\mathcal{X}} \left\| T_{\mathbf{X}\lambda}^{-1} k_x \right\|_{L^2, n}^2 d\mu(x) - \int_{\mathcal{X}} \left\| T_\lambda^{-1} k_x \right\|_{L^2, n}^2 d\mu(x) \right| \\
&\leq \int_{\mathcal{X}} \left| \left\| T_{\mathbf{X}}^{\frac{1}{2}} T_{\mathbf{X}\lambda}^{-1} k_x \right\|_{\mathcal{H}}^2 - \left\| T_{\mathbf{X}}^{\frac{1}{2}} T_\lambda^{-1} k_x \right\|_{\mathcal{H}}^2 \right| d\mu(x) \\
&= \int_{\mathcal{X}} \left| \left\| T_{\mathbf{X}}^{\frac{1}{2}} T_{\mathbf{X}\lambda}^{-1} k_x \right\|_{\mathcal{H}} - \left\| T_{\mathbf{X}}^{\frac{1}{2}} T_\lambda^{-1} k_x \right\|_{\mathcal{H}} \right| \cdot \left| \left\| T_{\mathbf{X}}^{\frac{1}{2}} T_{\mathbf{X}\lambda}^{-1} k_x \right\|_{\mathcal{H}} + \left\| T_{\mathbf{X}}^{\frac{1}{2}} T_\lambda^{-1} k_x \right\|_{\mathcal{H}} \right| d\mu(x). \\
&:= \int_{\mathcal{X}} |X_1 - X_2| \cdot |X_1 + X_2| \, d\mu(x).
\end{aligned}
$$

where we use Proposition A.3 in the second line.

**Part I: Bounding $|X_1 - X_2|$**

$$
\begin{aligned}
|X_1 - X_2| &\leq \left\| T_{\mathbf{X}}^{1/2} T_{\mathbf{X}\lambda}^{-1} \left( T - T_{\mathbf{X}} \right) T_\lambda^{-1} k_x \right\|_{\mathcal{H}} \\
&\leq \left\| T_{\mathbf{X}}^{1/2} T_{\mathbf{X}\lambda}^{-1/2} \right\| \cdot \left\| T_{\mathbf{X}\lambda}^{-1/2} T_\lambda^{1/2} \right\| \cdot \left\| T_\lambda^{-1/2} \left( T - T_{\mathbf{X}} \right) T_\lambda^{-1/2} \right\| \cdot \left\| T_\lambda^{-1/2} k_x \right\|_{\mathcal{H}}.
\end{aligned}
$$

**Step (i):** Denote $A_i = T_\lambda^{-1/2}(T - T_{\mathbf{x}_i})T_\lambda^{-1/2}$. By Lemma A.27:

$$\|A_i\| \leq \mathcal{N}_1(\lambda) + \kappa^2/\lambda, \quad \mu\text{-a.e. } x \in \mathcal{X}.$$

Again, by Lemma A.27, we have:

$$\mathbb{E}A_i^2 \preceq \mathbb{E}[T_\lambda^{-1/2}T_{x_i}T_\lambda^{-1/2}]^2 \preceq (\kappa^2/\lambda)\mathbb{E}[T_\lambda^{-1/2}T_{x_i}T_\lambda^{-1/2}] \preceq (\kappa^2/\lambda)T_\lambda^{-1}T.$$

Define $V := (\kappa^2/\lambda)T_\lambda^{-1}T$, we have:

$$\|V\| = \frac{\kappa^2}{\lambda}\frac{\lambda_0}{\lambda_0 + \lambda} = \frac{\kappa^2}{\lambda}\frac{\|T\|}{\|T\| + \lambda} \leq \frac{\kappa^2}{\lambda};$$

$$\text{tr}V = \frac{\kappa^2}{\lambda}\mathcal{N}_1(\lambda);$$

$$\frac{\text{tr}V}{\|V\|} = \frac{\mathcal{N}_1(\lambda)(\|T\| + \lambda)}{\|T\|},$$

where $\lambda_0$ is the biggest eigenvalue. Adopt Lemma A.24 to $A_i$, $V$, we get the following claim.

For any fixed $\delta \in (0, 1)$, with probability at least $1 - \delta$, we have

$$\|T_\lambda^{-\frac{1}{2}}(T - T_{\mathbf{x}})T_\lambda^{-\frac{1}{2}}\| \leq \frac{2\left(\mathcal{N}_1(\lambda) + \frac{\kappa^2}{\lambda}\right)}{3n}\beta + \sqrt{\frac{2\kappa^2}{n\lambda}\beta},$$

where

$$\beta = \ln\frac{4\mathcal{N}_1(\lambda)(\|T\| + \lambda)}{\delta\|T\|}.$$

Recall that the condition $\frac{1}{\lambda}\ln n/n = o(1)$ implies that $\frac{1}{\lambda} = O(n)$. Furthermore, notice that $\mathcal{N}_1(\lambda) \leq \sum_{i \in N}\frac{\lambda_i}{\lambda} \leq \frac{\kappa^2}{\lambda}$, we have $\beta = O(\ln n)$, so when $n$ is sufficiently large, we can conclude that

$$\|T_\lambda^{-\frac{1}{2}}(T - T_{\mathbf{x}})T_\lambda^{-\frac{1}{2}}\| \lesssim \sqrt{\frac{2\kappa^2}{n\lambda}\beta} \lesssim n^{-\frac{1}{2}}\frac{1}{\lambda^{\frac{1}{2}}}(\ln n)^{\frac{1}{2}}. \tag{29}$$

**Step (ii):** By Lemma A.26:

$$\|T_\lambda^{-1/2}k(x, \cdot)\|_{\mathcal{H}} \leq \sqrt{\kappa^2/\lambda}, \quad \mu\text{-a.e.} \tag{30}$$

**Step (iii):** We now bound the first two terms in equation 29. Under the assumption that $\frac{1}{\lambda}\frac{\ln n}{n} = o(1)$, equation equation 29 implies that for sufficiently large $n$:

$$a := \|T_\lambda^{-\frac{1}{2}}(T - T_{\mathbf{x}})T_\lambda^{-\frac{1}{2}}\| \leq \frac{2}{3}. \tag{31}$$

This allows us to establish the following key operator norm bounds:

**Lemma A.5.** *For sufficiently large $n$, the following inequalities hold with probability at least $1 - \delta$:*

*1.* $\|T_\lambda^{-\frac{1}{2}}T_{\mathbf{X}\lambda}^{\frac{1}{2}}\|^2 \leq 2$

*2.* $\|T_\lambda^{\frac{1}{2}}T_{\mathbf{X}\lambda}^{-\frac{1}{2}}\|^2 \leq 3$

*Proof.* For the first bound:

$$\|T_\lambda^{-\frac{1}{2}}T_{\mathbf{X}\lambda}^{\frac{1}{2}}\|^2 = \|T_\lambda^{-\frac{1}{2}}T_{\mathbf{X}\lambda}T_\lambda^{-\frac{1}{2}}\|$$
$$= \|T_\lambda^{-\frac{1}{2}}(T_{\mathbf{X}} + \lambda)T_\lambda^{-\frac{1}{2}}\|$$
$$= \|T_\lambda^{-\frac{1}{2}}(T_{\mathbf{X}} - T + T + \lambda)T_\lambda^{-\frac{1}{2}}\|$$
$$= \|T_\lambda^{-\frac{1}{2}}(T_{\mathbf{X}} - T)T_\lambda^{-\frac{1}{2}} + I\|$$
$$\leq a + 1 \leq 2.$$

For the second bound:

$$\|T_\lambda^{\frac{1}{2}} T_{\mathbf{X}\lambda}^{-\frac{1}{2}}\|^2 = \|T_\lambda^{\frac{1}{2}} T_{\mathbf{X}\lambda}^{-1} T_\lambda^{\frac{1}{2}}\|$$

$$= \|(T_\lambda^{-\frac{1}{2}} T_{\mathbf{X}\lambda} T_\lambda^{-\frac{1}{2}})^{-1}\|$$

$$= \|(I - T_\lambda^{-\frac{1}{2}}(T - T_{\mathbf{X}})T_\lambda^{-\frac{1}{2}})^{-1}\|$$

$$\leq \sum_{k=0}^{\infty} \|T_\lambda^{-\frac{1}{2}}(T - T_{\mathbf{X}})T_\lambda^{-\frac{1}{2}}\|^k$$

$$\leq \sum_{k=0}^{\infty} \left(\frac{2}{3}\right)^k \leq 3.$$

$\square$

Combining the estimates equation 29, equation 30, equation 32, and equation 32 into equation 29, we obtain that with high probability, the following inequality holds:

$$|X_1 - X_2| \lesssim 6n^{-\frac{1}{2}}\frac{1}{\lambda}(\ln n)^{\frac{1}{2}}, \quad \mu\text{-a.e. } \mathbf{x} \in \mathcal{X}. \tag{32}$$

**Part II: Bounding $\int X_2 d\mu$**

Now we shall bound $\int X_2 d\mu$. When $n$ is sufficiently large, with probability at least $1 - \delta$, we have

$$\int_{\mathcal{X}} X_2 d\mu(x) = \int_{\mathcal{X}} \|T_\lambda^{-1} k_x\|_{L^2,n} d\mu(x)$$

$$\leq \left[\int_{\mathcal{X}} \|T_\lambda^{-1} k_x\|_{L^2,n}^2 d\mu(x)\right]^{\frac{1}{2}}$$

$$\leq \left(\frac{3}{2}\mathcal{N}_2(\lambda) + \Delta_2\right)^{\frac{1}{2}}$$

$$\leq (2\mathcal{N}_2(\lambda))^{\frac{1}{2}},$$

where the third line follows from Lemma A.2, and the forth line follows from the assumption that $\frac{\sup_{i \in N} \frac{\lambda_i}{(\lambda_i + \lambda)^2}}{n} = o(\mathcal{N}_2(\lambda))$.

**Part III: Final Bound**

Combining equation 32 and equation 33, we get

$$\Delta_1 \leq \int_{\mathcal{X}} |X_1 - X_2| \cdot |X_1 + X_2| \, d\mu(x) \lesssim 36n^{-1}\lambda^{-2} \ln n + 24n^{-1/2}\lambda^{-1}\mathcal{N}_2(\lambda)^{1/2}(\ln n)^{1/2}.$$

$\square$

### A.2.4 FINAL PROOF OF THE VARIANCE TERM

Now we are ready to state the theorem about the variance term.

**Theorem A.6** (Variance term). *Suppose that Assumption 2.1, 3.2, 3.3 hold. If the following approximation conditions hold for some $\lambda = \lambda(d, n) \to 0$:*

$$\frac{\mathcal{N}_1(\lambda)}{n} \ln n = o(1); \quad \frac{\sup_{i \in N} \frac{\lambda_i}{(\lambda_i + \lambda)^2}}{n} = o(\mathcal{N}_2(\lambda)); \quad n^{-1}\frac{1}{\lambda^2}\ln n = o(\mathcal{N}_2(\lambda)), \tag{33}$$

*then we have*

$$\mathrm{Var}(\lambda) = \Theta_{\mathbb{P}}\left(\frac{\mathcal{N}_2(\lambda)}{n}\right). \tag{34}$$

*Proof.* Lemma A.1 has shown that

$$\mathrm{Var}(\lambda) = \frac{\sigma_\epsilon^2}{n} \int_{\mathcal{X}} \left\| (T_{\mathbf{X}} + \lambda)^{-1} k_x(\cdot) \right\|_{L^2,n}^2 d\mu(x).$$

Denote $\Delta_1$ as in Lemma A.4, then conditions equation 33 and Lemma A.4 imply that

$$\Delta_1 = o_{\mathbb{P}}\left(\mathcal{N}_2(\lambda)\right).$$

Further recall that in Lemma A.2, we have defined

$$\Delta_2 = \frac{5\kappa^2 \sup_{i \in N} \frac{\lambda_i}{(\lambda_i + \lambda)^2}}{3n} \ln \frac{2}{\delta} = o(\mathcal{N}_2(\lambda)).$$

Then for any $\delta \in (0,1)$, when $n$ is sufficiently large, with probability at least $1 - \delta$, we have

$$n \,\mathrm{Var}(\lambda)/\sigma_\epsilon^2 = \int_{\mathcal{X}} \left\| T_{\mathbf{X}\lambda}^{-1} k_x \right\|_{L^2,n}^2 d\mu(x) \leq \int_{\mathcal{X}} \left\| T_\lambda^{-1} k_x \right\|_{L^2,n}^2 d\mu(x) + \Delta_1$$

$$\leq \frac{3}{2} \int_{\mathcal{X}} \left\| T_\lambda^{-1} k_x \right\|_{L^2}^2 d\mu(x) + \Delta_1 + \Delta_2$$

$$= \frac{3}{2}\mathcal{N}_2(\lambda) + \Delta_1 + \Delta_2,$$

and

$$n \,\mathrm{Var}(\lambda)/\sigma_\epsilon^2 = \int_{\mathcal{X}} \left\| T_{\mathbf{X}\lambda}^{-1} k_x \right\|_{L^2,n}^2 d\mu(x) \geq \int_{\mathcal{X}} \left\| T_\lambda^{-1} k_x \right\|_{L^2,n}^2 d\mu(x) - \Delta_1$$

$$\geq \frac{1}{2} \int_{\mathcal{X}} \left\| T_\lambda^{-1} k_x \right\|_{L^2}^2 d\mu(x) - \Delta_1 - \Delta_2$$

$$= \frac{1}{2}\mathcal{N}_2(\lambda) - \Delta_1 - \Delta_2,$$

which further implies

$$n \,\mathrm{Var}(\lambda)/\sigma_\epsilon^2 = \Theta_{\mathbb{P}}\left(\mathcal{N}_2(\lambda)\right). \tag{35}$$

$\square$

### A.2.5 BIAS TERM

In this subsection, we shall derive Theorem A.9, which shows the upper bound of bias under some approximation conditions.

The triangle inequality implies that

$$\mathrm{Bias}(\lambda) = \left\| \tilde{f}_\lambda - f_\rho^* \right\|_{L^2} \leq \left\| f_\lambda - f_\rho^* \right\|_{L^2} + \left\| \tilde{f}_\lambda - f_\lambda \right\|_{L^2}, \tag{36}$$

and we shall bound the two terms separately.

The following lemma characterizes the first term of $Bias(\lambda)$.

**Lemma A.7.** *Suppose that Assumption 2.1, 3.2, 3.3 hold. Further more, suppose that there exist constants $c_1, c_2$ regardless of $d$ such that $c_1 \leq \lambda_0 \leq c_2$. Then for any $\lambda = \lambda(d,n) \to 0$, we have*

$$\left\| f_\lambda - f_\rho^* \right\|_{L^2} = \mathcal{R}_2(\lambda)^{\frac{1}{2}}. \tag{37}$$

*Proof.* Recall that we have defined $f_\rho^* = \sum\limits_{i=1}^{\infty} f_i e_i(x) \in L^2(\mathcal{X}, \mu)$ and $f_\lambda = (T + \lambda)^{-1} S_k^* f_\rho^*$. Therefore, we have

$$\left\| f_\lambda - f_\rho^* \right\|_{L^2}^2 = \left\| \sum_{i=1}^{\infty} f_i e_i(x) - \sum_{i=1}^{\infty} \frac{\lambda_i}{\lambda_i + \lambda} f_i e_i(x) \right\|_{L^2}^2$$

$$= \left\| \sum_{i=1}^{\infty} \frac{\lambda}{\lambda_i + \lambda} f_i e_i(x) \right\|_{L^2}^2$$

$$= \sum_{i=1}^{\infty} \left( \frac{\lambda}{\lambda_i + \lambda} f_i \right)^2$$

$$= \mathcal{R}_2(\lambda).$$

$\square$

The following lemma characterizes the second term of $Bias(\lambda)$.

**Lemma A.8.** *Suppose that Assumption 2.1, 3.2, 3.3 hold. If the following conditions hold for some* $\lambda = \lambda(d, n) \to 0$:

$$\frac{\mathcal{N}_1(\lambda)}{n} \ln n = o(1); \quad \frac{1}{n\lambda} = o(1); \quad \mathcal{N}_2(\lambda) = \Omega(1); \quad n^{-1}\frac{1}{\lambda^2} \ln n = o(\mathcal{N}_2(\lambda)). \tag{38}$$

*then we have*

$$\left\| \tilde{f}_\lambda - f_\lambda \right\|_{L^2} \le n^{-\frac{1}{2}} o((\mathcal{N}_2(\lambda))^{\frac{1}{2}}) + o(\mathcal{R}_2(\lambda)^{1/2}). \tag{39}$$

*Proof.* We begin by decomposing the $L^2$ norm difference:

$$\begin{aligned}
\left\| \tilde{f}_\lambda - f_\lambda \right\|_{L^2} &= \left\| S_k(\tilde{f}_\lambda - f_\lambda) \right\|_{L^2} \\
&= \left\| S_k T_\lambda^{-\frac{1}{2}} \cdot T_\lambda^{\frac{1}{2}} T_{\mathbf{X}\lambda}^{-1} T_\lambda^{\frac{1}{2}} \cdot T_\lambda^{-\frac{1}{2}} T_{\mathbf{X}\lambda}(\tilde{f}_\lambda - f_\lambda) \right\|_{L^2} \\
&\le \left\| S_k T_\lambda^{-\frac{1}{2}} \right\| \cdot \left\| T_\lambda^{\frac{1}{2}} T_{\mathbf{X}\lambda}^{-1} T_\lambda^{\frac{1}{2}} \right\| \cdot \left\| T_\lambda^{-\frac{1}{2}}(\tilde{g}_{\mathbf{z}} - T_{\mathbf{X}\lambda} f_\lambda) \right\|_{\mathcal{H}}.
\end{aligned}$$

**Part I: First Term Bound** For any $f \in \mathcal{H}$ with $\|f\|_{\mathcal{H}} = 1$, let $f = \sum_{i=1}^{\infty} a_i \lambda_i^{1/2} e_i$ where $\sum_{i=1}^{\infty} a_i^2 = 1$. Then:

$$\begin{aligned}
\left\| S_k T_\lambda^{-\frac{1}{2}} \right\| &= \sup_{\|f\|_{\mathcal{H}}=1} \left\| S_k T_\lambda^{-\frac{1}{2}} f \right\|_{L^2} \\
&\le \sup_{\|f\|_{\mathcal{H}}=1} \left\| \sum_{i \in N} \frac{\lambda_i^{1/2}}{(\lambda_i + \lambda)^{1/2}} a_i e_i \right\|_{L^2} \\
&\le \sup_{i \in N} \frac{\lambda_i^{1/2}}{(\lambda_i + \lambda)^{1/2}} \cdot \sup_{\|f\|_{\mathcal{H}}=1} \left\| \sum_{i \in N} a_i e_i \right\|_{L^2} \\
&\le 1.
\end{aligned}$$

**Part II: Second Term Bound** Under the assumption $\mathcal{N}_1(\lambda) \ln n/n = o(1)$, for any $\delta \in (0, 1)$ and sufficiently large $n$, with probability $\ge 1 - \delta$:

$$\left\| T_\lambda^{\frac{1}{2}} T_{\mathbf{X}\lambda}^{-1} T_\lambda^{\frac{1}{2}} \right\| \le \left\| T_\lambda^{\frac{1}{2}} T_{\mathbf{X}\lambda}^{-\frac{1}{2}} \right\| \cdot \left\| T_{\mathbf{X}\lambda}^{-\frac{1}{2}} T_\lambda^{\frac{1}{2}} \right\| \le \frac{3}{2}. \tag{40}$$

**Part III: Third Term Bound** The third term decomposes as:

$$\begin{aligned}
\left\| T_\lambda^{-\frac{1}{2}}(\tilde{g}_{\mathbf{z}} - T_{\mathbf{X}\lambda} f_\lambda) \right\|_{\mathcal{H}} &= \left\| T_\lambda^{-\frac{1}{2}}[(\tilde{g}_{\mathbf{z}} - (T_{\mathbf{X}} + \lambda + T - T) f_\lambda] \right\|_{\mathcal{H}} \\
&= \left\| T_\lambda^{-\frac{1}{2}}[(\tilde{g}_{\mathbf{z}} - T_{\mathbf{X}} f_\lambda) - (T + \lambda) f_\lambda + T f_\lambda] \right\|_{\mathcal{H}} \\
&= \left\| T_\lambda^{-\frac{1}{2}}[(\tilde{g}_{\mathbf{z}} - T_{\mathbf{X}} f_\lambda) - (g - T f_\lambda)] \right\|_{\mathcal{H}}.
\end{aligned} \tag{41}$$

Define $\xi_i = \xi(x_i) = T_\lambda^{-\frac{1}{2}}(K_{x_i} f_\rho^*(x_i) - T_{x_i} f_\lambda)$. For the $m$-th moment:

$$\begin{aligned}
\mathbb{E}\|\xi(x)\|_{\mathcal{H}}^m &= \mathbb{E}\|T_\lambda^{-\frac{1}{2}} K_x(f_\rho^* - f_\lambda(x))\|_{\mathcal{H}}^m \\
&\le \mathbb{E}\left( \|T_\lambda^{-\frac{1}{2}} k(x, \cdot)\|_{\mathcal{H}}^m \mathbb{E}[|f_\rho^* - f_\lambda(x)|^m | x] \right). 
\end{aligned} \tag{42}$$

From Lemma A.26, we have:

$$\|T_\lambda^{-\frac{1}{2}} k(x,\cdot)\|_{\mathcal{H}} \leq \sqrt{\kappa^2/\lambda}, \quad \mu\text{-a.e. } x \in \mathcal{X}$$

Using Lemma A.7, we are able to provide the following bound:

$$(42) \leq \left(\frac{\kappa^2}{\lambda}\right)^{m/2} \|f_\lambda - f_\rho^*\|_{L^\infty}^{m-2} \mathcal{R}_2(\lambda)$$

$$\leq \left(\sqrt{\frac{\kappa^2}{\lambda}} \|f_\lambda - f_\rho^*\|_{L^\infty}\right)^{m-2} \left(\sqrt{\frac{\kappa^2}{\lambda}} \mathcal{R}_2(\lambda)^{1/2}\right)^2.$$

Applying Lemma A.25 with $L = \sqrt{\frac{\kappa^2}{\lambda}} \|f_\lambda - f_\rho^*\|_{L^\infty}$ and $\sigma_\epsilon = \sqrt{\frac{\kappa^2}{\lambda}} \mathcal{R}_2(\lambda)^{1/2}$, with probability $\geq 1 - \delta$:

$$(41) \leq 4\sqrt{2} \log \frac{2}{\delta} \left(\frac{\sqrt{\frac{\kappa^2}{\lambda}} \|f_\lambda - f_\rho^*\|_{L^\infty}}{n} + \frac{\sqrt{\frac{\kappa^2}{\lambda}} \mathcal{R}_2(\lambda)^{1/2}}{\sqrt{n}}\right).$$

**Norm Bound on** $\|f_\lambda - f_\rho^*\|_{L^\infty}$: Notice that $\|f_\lambda - f_\rho^*\|_{L^\infty} \leq \|f_\lambda\|_{L^\infty} + \|f_\rho^*\|_{L^\infty}$. For every $f$ in $\mathcal{H}$, we have $f(x) = \langle k(x,\cdot), f\rangle_{\mathcal{H}} \leq \|k(x,\cdot)\|_{\mathcal{H}} \|f\|_{\mathcal{H}} \leq \kappa \|f\|_{\mathcal{H}}$. Hence, when $s \geq 1$, $f_\rho^* \in \mathcal{H}$, which indicates that there exists a constant $R_2$ such that $\|f_\rho^*\|_{L^\infty} \leq R_2$. Also,

$$\|f_\lambda\|_{L^\infty} \lesssim \|f_\lambda\|_{\mathcal{H}}$$

$$= \left\|\sum_{i=0}^\infty \frac{\lambda_i}{\lambda_i + \lambda} f_i e_i\right\|_{\mathcal{H}}$$

$$= \left(\sum_{i=0}^\infty \frac{\lambda_i^{1+s}}{(\lambda_i + \lambda)^2} \lambda_i^{-s} f_i^2\right)^{1/2}$$

$$\leq \left(\sup_{i \in N} \frac{\lambda_i^{1+s}}{(\lambda_i + \lambda)^2}\right)^{\frac{1}{2}} \cdot \left(\sum_{i=0}^\infty \lambda_i^{-s} f_i^2\right)^{1/2}$$

$$\leq \left(\sup_{i \in N} \frac{\lambda_i^{1+s}}{(\lambda_i + \lambda)^2}\right)^{\frac{1}{2}} \cdot R_1$$

$$\leq \lambda_0^{\frac{s-1}{2}} \cdot R_1$$

$$\lesssim 1.$$

Hence we get

$$\|f_\lambda - f_\rho^*\|_{L^\infty} \lesssim 1 + R_2$$

Thus the final bound becomes:

$$(41) \lesssim 4\sqrt{2} \log \frac{2}{\delta} \left(\frac{\sqrt{\frac{\kappa^2}{\lambda}} (1 + R_2)}{n} + \frac{\sqrt{\frac{\kappa^2}{\lambda}} \mathcal{R}_2(\lambda)^{1/2}}{\sqrt{n}}\right)$$

$$= O\left(\frac{1}{n\sqrt{\lambda}} + \frac{\mathcal{R}_2(\lambda)^{1/2}}{\sqrt{n\lambda}}\right)$$

$$= o(n^{-\frac{1}{2}}) + o(\mathcal{R}_2(\lambda)^{1/2})$$

$$= o\left(\left(\frac{\mathcal{N}_2(\lambda)}{n}\right)^{\frac{1}{2}}\right) + o(\mathcal{R}_2(\lambda)^{1/2}).$$

$\square$

*Final proof of the bias term.* Now we are ready to state the theorem about the bias term.

**Theorem A.9.** *Suppose that Assumption 2.1, 3.2, 3.3 hold. If the following condition holds for some* $\lambda = \lambda(d, n) \to 0$:

$$\frac{\mathcal{N}_1(\lambda)}{n} \ln n = o(1); \quad \frac{1}{n\lambda} = o(1); \quad \mathcal{N}_2(\lambda) = \Omega(1); \quad n^{-1}\frac{1}{\lambda^2} \ln n = o(\mathcal{N}_2(\lambda)), \quad (43)$$

*then we have*

$$\text{Bias}^2(\lambda) = \Theta_{\mathbb{P}}\left(\mathcal{R}_2(\lambda) + o\left(\frac{\mathcal{N}_2(\lambda)}{n}\right) + \frac{1}{n^2\lambda} + \frac{\mathcal{R}_2(\lambda)}{n\lambda}\right). \quad (44)$$

### A.2.6 FINAL PROOF OF THEOREM 3.4

Now we are ready to prove Theorem 3.4. Note that

$$\frac{\sup_{i \in N} \frac{\lambda_i}{(\lambda_i+\lambda)^2}}{n} \leq \frac{1}{4\lambda n} = o(1) = o(\mathcal{N}_2(\lambda)),$$

$\lambda = \lambda(d, n) \to 0$ in Theorem 3.4 satisfies all the conditions required in Theorem A.6 and Theorem A.9. Therefore, Theorem A.6 and Theorem A.9 show that

$$\text{Var}(\lambda) = \Theta_{\mathbb{P}}\left(\frac{\mathcal{N}_2(\lambda)}{n}\right); \quad \text{Bias}^2(\lambda) = \Theta_{\mathbb{P}}\left(\mathcal{R}_2(\lambda) + o\left(\frac{\mathcal{N}_2(\lambda)}{n}\right)\right).$$

Recalling the bias-variance decomposition , we finish the proof.

### A.3 PROOF OF THEOREM 3.5

The only difference lies in the **Part III: Third Term Bound** in Lemma A.8.

Define $\xi_i = \xi(x_i) = T_\lambda^{-\frac{1}{2}}(K_{x_i}f_\rho^*(x_i) - T_{x_i}f_\lambda)$. Consider the subset $\Omega_1 = \{x \in \mathcal{X} : |f_\rho^*(x)| \leq t\}$ and $\Omega_2 = \mathcal{X}\backslash\Omega_1$, where $t$ will be chosen later. We have the following decomposition of equation 41:

$$\left\|\frac{1}{n}\sum_{i=1}^n \xi_i - \mathbb{E}\xi(x)\right\|_{\mathcal{H}} \leq \left\|\frac{1}{n}\sum_{i=1}^n \xi_i I_{x_i \in \Omega_1} - \mathbb{E}\xi(x)I_{x\in\Omega_1}\right\|_{\mathcal{H}} + \|\frac{1}{n}\sum_{i=1}^n \xi_i I_{x_i\in\Omega_2}\|_{\mathcal{H}} + \|\mathbb{E}\xi(x)I_{x\in\Omega_2}\|_{\mathcal{H}}$$

$$:= \text{I} + \text{II} + \text{III}. \quad (45)$$

Next we choose $t = n^{\frac{1-s}{2}+\epsilon_1}, q = \frac{2}{1-s} - \epsilon_2$ such that

$$\epsilon_1 < \epsilon; \quad \text{and} \quad \frac{1-s}{2} + \epsilon_1 > 1/\left(\frac{2}{1-s} - \epsilon_2\right), \quad (46)$$

where $\epsilon$ is given in equation 8. Then we can bound the three terms in equation 45 as follows:

(i) For the first term in equation 45, denoted as I, notice that

$$\left\|(f_\lambda - f_\rho^*)I_{x_i\in\Omega_1}\right\|_{L^\infty} \leq \|f_\lambda\|_{L^\infty} + n^{\frac{1-s}{2}+\epsilon_1}. \quad (47)$$

Imitating the third part in the proof of Lemma A.8, we have

$$\text{I} \lesssim 4\sqrt{2}\log\frac{2}{\delta}\left(\frac{\sqrt{\frac{\kappa^2}{\lambda}}(n^{\frac{1-s}{2}+\epsilon_1} + \|f_\lambda\|_{L^\infty})}{n} + \frac{\sqrt{\frac{\kappa^2}{\lambda}}\mathcal{R}_2(\lambda)^{1/2}}{\sqrt{n}}\right). \quad (48)$$

(ii) For the second term in equation 45, denoted as II. Since $q = \frac{2}{1-s} - \epsilon_2 < \frac{2}{1-s}$, Lemma A.28 shows that,

$$[\mathcal{H}]^s \hookrightarrow L^q(\mathcal{X}, \mu). \quad (49)$$

Hence, there exists $0 < C_q < \infty$ such that $\|f_\rho^*\|_{L^q(\mathcal{X},\mu)} \leq C_q$. Using the Markov inequality, we have

$$P(x \in \Omega_2) = P\left(|f_\rho^*(x)| > t\right) \leq \frac{\mathbb{E}|f_\rho^*(x)|^q}{t^q} \leq \frac{(C_q)^q}{t^q}.$$

Further, since equation 46 guarantees $t^q \gg n$, we have

$$\tau_n := P\left(\text{II} \geq \mathcal{R}_2(\lambda)^{\frac{1}{2}}\right) \leq P\left(\exists x_i \text{ s.t. } x_i \in \Omega_2,\right)$$

$$= 1 - P\left(x \notin \Omega_2\right)^n$$

$$\leq 1 - \left(1 - \frac{(C_q)^q}{t^q}\right)^n \to 0. \tag{50}$$

(iii)   For the third term in equation 45, denoted as III.   Since Lemma A.26 implies that $\|T_\lambda^{-\frac{1}{2}} k(x, \cdot)\|_{\mathcal{H}} \leq \sqrt{\frac{\kappa^2}{\lambda}}$, $\mu$-a.e. $x \in \mathcal{X}$, so

$$\text{III} \leq \mathbb{E}\|\xi(x) I_{x \in \Omega_2}\|_{\mathcal{H}} \leq \mathbb{E}\left[\|T_\lambda^{-\frac{1}{2}} k(x, \cdot)\|_{\mathcal{H}} \cdot \left|\left(f_\rho^* - f_\lambda(x)\right) I_{x \in \Omega_2}\right|\right]$$

$$\leq \sqrt{\frac{\kappa^2}{\lambda}} \mathbb{E}\left|\left(f_\rho^* - f_\lambda(x)\right) I_{x \in \Omega_2}\right|$$

$$\leq \sqrt{\frac{\kappa^2}{\lambda}} \|f_\rho^* - f_\lambda\|_{L^2}^{\frac{1}{2}} \cdot P\left(x \in \Omega_2\right)^{\frac{1}{2}}$$

$$\lesssim \sqrt{\frac{\kappa^2}{\lambda}} \mathcal{R}_2(\lambda)^{\frac{1}{2}} t^{-\frac{q}{2}}, \tag{51}$$

where we use Cauchy-Schwarz inequality for the third inequality and Lemma A.7 for the forth inequality. Recalling that the choices of $t, q$ satisfy $t^{-q} \ll n^{-1}$ and we have assumed $\frac{1}{n\lambda} = o(1)$, we have

$$\text{III} = o\left(\mathcal{R}_2(\lambda)^{\frac{1}{2}}\right). \tag{52}$$

Plugging equation 48, equation 50 and equation 52 into equation 45, we finish the proof.

## A.4   PROOF OF THEOREM 4.8

### A.4.1   FACTS OF THE EIGENVALUES OF LARGE DIMENSIONAL GAUSSIAN KERNEL

Consider the kernel satisfying Assumption 4.1. Recall the notations in Proposition 4.4. We have the following facts of the eigenvalues.

**Lemma A.10.** *For any fixed integer $p \geq 0$, there exist constants $\mathfrak{C}, \mathfrak{C}_1$ and $\mathfrak{C}_2$ only depending on $p$ such that for any $d \geq \mathfrak{C}$, we have*

$$\mathfrak{C}_1 d^{-k} \leq \mu_k \leq \mathfrak{C}_2 d^{-k}, \quad k = 0, 1, \cdots, p + 1. \tag{53}$$

*Proof.* Recall that $\mu_k = (2a/A)^{\frac{d}{2}} \cdot B^k$, for $(2a/A)^{\frac{1}{2}}$, we have:

$$\sqrt{2a/A} = \sqrt{\frac{2a}{a + b + \sqrt{a^2 + 2ab}}}$$

$$= \sqrt{\frac{2a}{2a + 2b + \sqrt{a^2 + 2ab} - a - b}}$$

$$= \sqrt{\frac{2a}{2a + 2b + \Theta(d^{-2})}} \tag{54}$$

$$= \sqrt{\frac{1}{1 + \frac{1}{2\ell^2 ad} + \Theta(d^{-2})}}$$

$$= \sqrt{1 - \frac{1}{2\ell^2 ad} + \Theta(d^{-2})},$$

where the notation $\Theta$ involves constants only depending on $p$. Hence, when $d$ is large enough,

$$(2a/A)^{\frac{d}{2}} = (1 - \frac{1}{2\ell^2 ad} + \Theta(d^{-2}))^{d/2} \in \left(e^{-\frac{1}{4\ell^2 a}}, 1\right) \tag{55}$$

For $B$, we have:

$$B = \frac{b}{A}$$

$$= \frac{b}{a + b + \sqrt{a^2 + 2ab}} \tag{56}$$

$$= \frac{b}{2a + 2b + \Theta(d^{-2})}$$

$$= \Theta(d^{-1})$$

Hence when $d$ is large enough, we have

$$\mathfrak{C}_1 d^{-k} \leq \mu_k \leq \mathfrak{C}_2 d^{-k}, \quad k = 0, 1, \cdots, p+1. \tag{57}$$

$\square$

**Lemma A.11.** *For any fixed integer $p \geq 0$, there exist constants $\mathfrak{C}_3, \mathfrak{C}_4$ and $\mathfrak{C}$ only depending on $p$, such that for any $d \geq \mathfrak{C}$, we have*

$$\mathfrak{C}_3 d^k \leq N(d, k) \leq \mathfrak{C}_4 d^k, \quad k = 0, 1, \cdots, p+1. \tag{58}$$

*Proof.* Recall that $N(d, k) = C_{k+d-1}^{d-1} = \frac{(k+d-1)!}{k!(d-1)!}$. By Stirling formula, we have

$$(k + d - 1)! = \sqrt{2\pi(k + d - 1)} \left(\frac{k + d - 1}{e}\right)^{k+d-1} (1 + \Theta(d^{-1}))$$

$$(d - 1)! = \sqrt{2\pi(d - 1)} \left(\frac{d - 1}{e}\right)^{d-1} (1 + \Theta(d^{-1}))$$

Hence,

$$N(d, k) = \frac{\sqrt{2\pi(k + d - 1)} \left(\frac{k+d-1}{e}\right)^{k+d-1} (1 + \Theta(d^{-1}))}{k!\sqrt{2\pi(d - 1)} \left(\frac{d-1}{e}\right)^{d-1} (1 + \Theta(d^{-1}))}$$

$$= \Theta\left(\frac{(k + d - 1)^{k+d-1}}{(d - 1)^{d-1}}\right) \tag{59}$$

$$= \Theta\left((1 + \frac{k}{d - 1})^{d-1}(k + d - 1)^k\right)$$

$$= \Theta(d^k),$$

where the notation $\Theta$ involves constants only depending on $p$. $\square$

**Lemma A.12.** *Let $k = k_d$ be a kernel satisfying Assumption 3.3,4.1. By choosing $\lambda = d^{-l}$ for some $l > 0$, if $p \leq l \leq p + 1$ for some $p \in \{0, 1, 2 \cdots\}$, we have:*

$$\mathcal{N}_1(\lambda) = O\left(\lambda^{-1}\right); \tag{60}$$

$$\mathcal{N}_2(\lambda) = \Theta\left(d^p + \lambda^{-2}d^{-(p+1)}\right). \tag{61}$$

*The notation $O, \Theta$ involves constants only depending on $p$.*

*Proof.* For $\mathcal{N}_1$, the following inequality holds:

$$\mathcal{N}_1 \leq \frac{1}{\lambda} \sup k(x, x) \leq \frac{\kappa^2}{\lambda} \tag{62}$$

For $\mathcal{N}_2$, the following inequality holds:

$$\mathcal{N}_2 = \sum_{k \in N} \left(\frac{\lambda_k}{\lambda + \lambda_k}\right)^2$$

$$\leq \sum_{k \leq p} N(d, k) + \lambda^{-2} \sum_{k > p} \mu_k^2 N(d, k)$$

$$\leq \sum_{k \leq p} N(d, k) + \lambda^{-2}\mu_{p+1} \sum_{k > p} \mu_k N(d, k) \tag{63}$$

$$\lesssim d^p + \lambda^{-2}d^{-(p+1)}$$

On the other hand, we have the lower bound:

$$\mathcal{N}_2 = \sum_{k \in N} \left( \frac{\lambda_k}{\lambda + \lambda_k} \right)^2$$

$$\geq \left( \frac{\mu_p}{\mu_p + \lambda} \right)^2 N(d, p) + (\mu_{p+1} + \lambda)^{-2} \mu_{p+1}^2 N(d, p+1) \tag{64}$$

$$\geq \frac{1}{4} N(d, p) + (2\lambda)^{-2} \mu_{p+1}^2 N(d, p+1)$$

$$\gtrsim d^p + \lambda^{-2} d^{-(p+1)}$$

$\square$

**Lemma A.13.** *Let $k = k_d$ be a sequence of kernels satisfying Assumption 3.3,4.1. Define $\tilde{s} = \min\{s, 2\}$. By choosing $\lambda = d^{-l}$ for some $l > 0$, if $p \leq l < p + 1$ for some $p \in \{0, 1, 2 \cdots\}$, we have:*

$$\mathcal{R}_2(\lambda) = \Theta(\lambda^2 d^{(2-\tilde{s})p} + d^{-(p+1)\tilde{s}}). \tag{65}$$

*The notation $\Theta$ involves constants only depending on $p$.*

*Proof.* Define $\mathcal{I}_{d,k}$ as the indicator set of all eigenfunctions of $\mu_k$. When $s \leq 2$,

$$\mathcal{R}_2(\lambda) = \lambda^2 \sum_{k=0}^{\infty} \frac{\mu_k^s}{(\mu_k + \lambda)^2} \sum_{i \in \mathcal{I}_{d,k}} \mu_k^{-s} f_i^2$$

$$\leq \lambda^2 \left( \sum_{k=0}^{p} \frac{\mu_k^s}{(\mu_k + \lambda)^2} R_1^2 + \sum_{k=p+1}^{\infty} \frac{\mu_k^s}{(\mu_k + \lambda)^2} \sum_{i \in \mathcal{I}_{d,k}} \mu_k^{-s} f_i^2 \right)$$

$$\leq \lambda^2 \left( \sum_{k=0}^{p} \mu_k^{s-2} R_1^2 + \lambda^{-2} \sum_{k=p+1}^{\infty} \mu_{p+1}^s \sum_{i \in \mathcal{I}_{d,k}} \mu_k^{-s} f_i^2 \right)$$

$$\leq \lambda^2 \left( p \mu_p^{s-2} R_1^2 + \lambda^{-2} \mu_{p+1}^s R_1^2 \right)$$

$$\lesssim \lambda^2 d^{(2-s)p} + d^{-(p+1)s}.$$

On the other hand,

$$\mathcal{R}_2(\lambda) \geq \lambda^2 \left( \frac{\mu_p^s}{(\mu_p + \lambda)^2} \sum_{i \in \mathcal{I}_{d,p}} \mu_p^{-s} f_i^2 + \frac{\mu_{p+1}^s}{(\mu_{p+1} + \lambda)^2} \sum_{i \in \mathcal{I}_{d,p+1}} \mu_{p+1}^{-s} f_i^2 \right)$$

$$\gtrsim \lambda^2 \left( \mu_p^{s-2} + \lambda^{-2} \mu_{p+1}^s \right)$$

$$\gtrsim \lambda^2 d^{(2-s)p} + d^{-(p+1)s}.$$

When $s > 2$, without loss of generality, we may assume that $\lambda_0 \leq 1, \lambda \leq 1$. Hence,

$$\mathcal{R}_2(\lambda) \geq \lambda^2 \sum_{i=0}^{\infty} \frac{f_i^2}{(\lambda_i + \lambda)^2} \geq \frac{1}{4} \lambda^2 \sum_{i=0}^{\infty} f_i^2 \gtrsim \lambda^2.$$

On the other hand,

$$\mathcal{R}_2(\lambda) = \lambda^2 \sum_{k=0}^{\infty} \left( \frac{\mu_k^s}{(\mu_k + \lambda)^2} \sum_{i \in \mathcal{I}_{d,k}} \mu_k^{-s} f_i^2 \right)$$

$$\leq \lambda^2 \left( \sup_{k \geq 0} \frac{\mu_k^s}{(\mu_k + \lambda)^2} \cdot \sum_{k=0}^{\infty} \sum_{i \in \mathcal{I}_{d,k}} \mu_k^{-s} f_i^2 \right)$$

$$\leq \lambda^2 \cdot \sup_{k \geq 0} \frac{\mu_k^s}{(\mu_k + \lambda)^2} \cdot R_1^2$$

$$\lesssim \lambda^2.$$

Hence, we get the desired result. $\square$

**Lemma A.14.** *Let $k = k_d$ be a sequence of kernels satisfying Assumption 3.3,4.1. When $0 < s < 1$, by choosing $\lambda = d^{-l}$ for some $l > 0$, if $p \le l < p + 1$ for some $p \in \{0, 1, 2 \cdots\}$, we have:*

$$\|f_\lambda\|_{L^\infty} = O\left(d^{\frac{(1-s)p}{2}} + \lambda^{-1} d^{-\frac{(1+s)(p+1)}{2}}\right). \tag{66}$$

*The notation $\Theta$ involves constants only depending on $p$.*

*Proof.* Notice that $f(x) = \langle k(x, \cdot), f\rangle_\mathcal{H} \le \|k(x, \cdot)\|_\mathcal{H} \|f\|_\mathcal{H} \le \kappa\|f\|_\mathcal{H}$. We shall now bound $f_\mathcal{H}$.

$$
\begin{aligned}
\|f_\lambda\|_\mathcal{H}^2 &= \left\|\sum_{i=0}^\infty \frac{\lambda_i}{\lambda_i + \lambda} f_i e_i\right\|_\mathcal{H}^2 \\
&= \sum_{i=0}^\infty \frac{\lambda_i^{1+s}}{(\lambda_i + \lambda)^2} \lambda_i^{-s} f_i^2 \\
&= \sum_{k=0}^\infty \frac{\mu_k^{1+s}}{(\mu_k + \lambda)^2} \sum_{i \in \mathcal{I}_{d,k}} \mu_k^{-s} f_i^2 \\
&\le \sum_{k=0}^p \frac{\mu_k^{1+s}}{(\mu_k + \lambda)^2} R_1^2 + \sum_{k=p+1}^\infty \frac{\mu_k^{1+s}}{(\mu_k + \lambda)^2} \sum_{i \in \mathcal{I}_{d,k}} \mu_k^{-s} f_i^2 \\
&\le \sum_{k=0}^p \mu_k^{s-1} R_1^2 + \lambda^{-2} \sum_{k=p+1}^\infty \mu_{p+1}^{s+1} \sum_{i \in \mathcal{I}_{d,k}} \mu_k^{-s} f_i^2 \\
&\le p\lambda_p^{s-1} R_1^2 + \lambda^{-2} \mu_{p+1}^{s+1} R_1^2 \\
&\lesssim d^{(1-s)p} + \lambda^{-2} d^{-(p+1)(s+1)}.
\end{aligned}
$$

Hence, we get the desired result. $\qquad\square$

### A.4.2 Proof of Theorem 4.8

Now we are able to provide the proof of Theorem 4.8.

*Proof of Theorem 4.8.* In order to prove the theorem, we will (i) locate the balancing regularization index $l_{\text{balance}}$ (so $\lambda_{\text{balance}} = d^{-l_{\text{balance}}}$), (ii) verify the four technical conditions in equation 5 for $l = l_{\text{balance}}$, and (iii) show that no other choice of $\lambda$ yields a strictly better rate (hence $\lambda_{\text{balance}}$ is optimal). Throughout we use Lemma A.12 and Lemma A.13 for asymptotic orders of $\mathcal{N}_1, \mathcal{N}_2, \mathcal{R}_2$.

**Step 1. Determine $l_{\text{balance}}$.** Assume $s \ge 1$, let $\tilde{s} = \min\{s, 2\}$ and write $\lambda = d^{-l}$ with $0 < l < \gamma$. Using Lemma A.12 and Lemma A.13 we separate three regimes for $l$ (indexed by the integer $p \ge 0$):

- If $l \in \left(p,\ p + \frac{1}{2}\right]$ then

$$\frac{\mathcal{N}_2(\lambda)}{n} \asymp d^{p-\gamma}, \qquad \mathcal{R}_2(\lambda) \asymp d^{-2l+(2-\tilde{s})p}.$$

Balancing variance and bias,

$$d^{p-\gamma} \asymp d^{-2l+(2-\tilde{s})p} \implies l_{\text{balance}} = \frac{\gamma + p - p\tilde{s}}{2},$$

and requiring $l_{\text{balance}} \in \left(p, p + \frac{1}{2}\right]$ yields

$$\gamma \in \left(p + p\tilde{s},\ p + p\tilde{s} + 1\right].$$

- If $l \in \left(p + \frac{1}{2},\ p + \frac{\tilde{s}}{2}\right]$ then

$$\frac{\mathcal{N}_2(\lambda)}{n} \asymp d^{2l-p-1-\gamma}, \qquad \mathcal{R}_2(\lambda) \asymp d^{-2l+(2-\tilde{s})p},$$

and balancing gives

$$d^{2l-p-1-\gamma} \asymp d^{-2l+(2-\tilde{s})p} \quad \Longrightarrow \quad l_{\text{balance}} = \frac{\gamma + 3p - p\tilde{s} + 1}{4},$$

with admissible $\gamma$ satisfying

$$\gamma \in \big(p + p\tilde{s} + 1, \ p + p\tilde{s} + 2\tilde{s} - 1\big].$$

- If $l \in \big(p + \frac{\tilde{s}}{2}, \ p + 1\big]$ then

$$\frac{\mathcal{N}_2(\lambda)}{n} \asymp d^{2l-p-1-\gamma}, \qquad \mathcal{R}_2(\lambda) \asymp d^{-(p+1)\tilde{s}},$$

and balancing yields

$$d^{2l-p-1-\gamma} \asymp d^{-(p+1)\tilde{s}} \quad \Longrightarrow \quad l_{\text{balance}} = \frac{\gamma + (p+1)(1-\tilde{s})}{2},$$

with admissible $\gamma$ satisfying

$$\gamma \in \big(p + p\tilde{s} + 2\tilde{s} - 1, \ (p+1) + (p+1)\tilde{s}\big].$$

From now on we fix $l_{\text{balance}}$ as above according to which $\gamma$-regime we are in.

**Step 2. Verification of the hypotheses of Theorem 3.4 (conditions equation 5).** To apply Theorem 3.4 we must check the four conditions

$$\frac{\mathcal{N}_1(\lambda)}{n} \ln n = o(1), \qquad \frac{1}{n\lambda} = o(1), \qquad \mathcal{N}_2(\lambda) = \Omega(1), \qquad \frac{\ln n}{n\lambda^2} = o\big(\mathcal{N}_2(\lambda)\big) \qquad (67)$$

for $\lambda = \lambda_{\text{balance}}$. We verify these case by case. (The orders of $\mathcal{N}_1, \mathcal{N}_2, \mathcal{R}_2$ used below are precisely those given by Lemmas A.12 and A.13.)

*Case A:* $\gamma \in \big(p + p\tilde{s}, \ p + p\tilde{s} + 1\big]$ (so $l_{\text{balance}} = (\gamma + p - p\tilde{s})/2 \in (p, p + \frac{1}{2}])$.
(1) $\frac{1}{n\lambda} = d^{l-\gamma} = o(1)$ is equivalent to $l < \gamma$. Substituting $l_{\text{balance}}$ we get

$$\frac{\gamma + p - p\tilde{s}}{2} < \gamma \iff \gamma > p - p\tilde{s},$$

which holds because $\gamma > p + p\tilde{s} \geq p - p\tilde{s}$. Thus the second condition in equation 67 holds.

(2) For $\frac{\mathcal{N}_1(\lambda)}{n} \ln n = o(1)$, by Lemma A.12 (case $l \in (p, p + \frac{1}{2}]$) we have

$$\frac{\mathcal{N}_1(\lambda)}{n} \ln n \lesssim \frac{1}{n\lambda} \ln n = d^{l-\gamma} \ln n = o(1).$$

Thus the first condition in equation 67 holds.

(3) For $\mathcal{N}_2(\lambda) = \Omega(1)$, from Lemma A.12 we have $\mathcal{N}_2(\lambda) \gtrsim d^p$, hence $\mathcal{N}_2(\lambda) = \Omega(1)$ obviously holds.

(4) For $(\ln n)/(n\lambda^2) = o(\mathcal{N}_2(\lambda))$, using $n \asymp d^\gamma$ and $\lambda = d^{-l_{\text{balance}}}$, this condition is equivalent to

$$d^{-\gamma} \cdot d^{2l_{\text{balance}}} \cdot \gamma \ln d \ll d^p.$$

Substituting $l_{\text{balance}} = (\gamma + p - p\tilde{s})/2$ and we find that the condition can be verified when $p > 0$. When $p = 0$, a simple calculation also implies that the condition holds.

*Case B:* $\gamma \in \big(p + p\tilde{s} + 1, \ p + p\tilde{s} + 2\tilde{s} - 1\big]$ (so $l_{\text{balance}} = (\gamma + 3p - p\tilde{s} + 1)/4 \in (p + \frac{1}{2}, p + \frac{\tilde{s}}{2}])$.

(1) $\frac{1}{n\lambda} = d^{l-\gamma} = o(1)$ reduces to

$$\frac{\gamma + 3p - p\tilde{s} + 1}{4} < \gamma \iff \gamma > p - \frac{p\tilde{s}}{3} + \frac{1}{3},$$

which holds on the stated $\gamma$-interval.

(2) By Lemma A.12 in this regime we have

$$\frac{\mathcal{N}_1(\lambda)}{n} = O\big(d^{\frac{\gamma+3p-p\tilde{s}+1}{2}-\gamma}\big),$$

hence

$$\frac{\mathcal{N}_1(\lambda)}{n} \ln n = O\big(d^{\frac{\gamma+3p-p\tilde{s}+1}{2}-\gamma} \cdot \gamma \ln d\big) = o(1).$$

(3) For $\mathcal{N}_2(\lambda) = \Omega(1)$, from Lemma A.12 we have $\mathcal{N}_2(\lambda) \gtrsim d^p$, hence $\mathcal{N}_2(\lambda) = \Omega(1)$ obviously holds.

(4) For $(\ln n)/(n\lambda^2) = o(\mathcal{N}_2(\lambda))$, we have

$$\frac{\ln n}{n\lambda^2} \asymp d^{2l-\gamma} \ln n = o(d^{2l-p-1}) = o(\mathcal{N}_2(\lambda)).$$

*Case C:* $\gamma \in \big(p+p\tilde{s}+2\tilde{s}-1,\ (p+1)+(p+1)\tilde{s}\big]$ (so $l_{\text{balance}} = (\gamma+(p+1)(1-\tilde{s}))/2 \in (p+\frac{\tilde{s}}{2}, p+1]$). Again the check is analogous:

(1) $1/(n\lambda) = o(1)$ is equivalent to

$$\frac{\gamma + (p+1)(1-\tilde{s})}{2} < \gamma \iff \gamma > p - p\tilde{s} + 1 - \tilde{s},$$

which is true for the stated $\gamma$-range.

(2) By Lemma A.12 in this regime we have

$$\frac{\mathcal{N}_1(\lambda)}{n} = O\big(d^{\frac{\gamma+(p+1)(1-\tilde{s})}{2}-\gamma}\big),$$

hence

$$\frac{\mathcal{N}_1(\lambda)}{n} \ln n = O\big(d^{\frac{\gamma+(p+1)(1-\tilde{s})}{2}-\gamma} \cdot \gamma \ln d\big) = o(1).$$

(3) For $\mathcal{N}_2(\lambda) = \Omega(1)$, from Lemma A.12 we have $\mathcal{N}_2(\lambda) \gtrsim d^p$, hence $\mathcal{N}_2(\lambda) = \Omega(1)$ obviously holds.

(4) For $(\ln n)/(n\lambda^2) = o(\mathcal{N}_2(\lambda))$, we have

$$\frac{\ln n}{n\lambda^2} \asymp d^{2l-\gamma} \ln n = o(d^{2l-p-1}) = o(\mathcal{N}_2(\lambda)).$$

Thus in each admissible $\gamma$-regime the four conditions in equation 67 are satisfied for $\lambda = \lambda_{\text{balance}}$. Moreover, by simple calculation, the conditions of Theorem 3.4 also hold for all $l \leq l_{\text{balance}}$.

**Step 3. Conclude the optimal rate and show $\lambda_{\text{balance}}$ is best.** By Theorem 3.4 (applied with the verified conditions equation 67) we have for $\lambda = \lambda_{\text{balance}}$

$$\mathbb{E}\left[\|\hat{f}_{\lambda_{\text{balance}}} - f_\rho^*\|_{L^2}^2 \mid \mathbf{X}\right] = \Theta_{\mathbb{P}}\Big(\frac{\sigma^2 \mathcal{N}_2(\lambda_{\text{balance}})}{n} + \mathcal{R}_2(\lambda_{\text{balance}})\Big), \tag{68}$$

and the computations in Step 1 (together with Lemma A.13) give the three displayed rates corresponding to the three $\gamma$-regimes.

It remains to show no other $\lambda$ yields a strictly smaller rate. For $\lambda \gtrsim \lambda_{\text{balance}}$ the bias term $\mathcal{R}_2(\lambda)$ cannot decrease below $\mathcal{R}_2(\lambda_{\text{balance}})$, while $\mathcal{R}_2(\lambda_{\text{balance}}) = \Theta_{\mathbb{P}}\Big(\frac{\sigma^2 \mathcal{N}_2(\lambda_{\text{balance}})}{n} + \mathcal{R}_2(\lambda_{\text{balance}})\Big)$ by the definition of $\lambda_{\text{balance}}$. Hence, the RHS of equation 68 gives a lower bound (up to constants) for all such $\lambda$. For $\lambda \lesssim \lambda_{\text{balance}}$, note the matrix inequality

$$(\mathbf{K} + \lambda_1)^{-2} \succeq (\mathbf{K} + \lambda_2)^{-2}, \qquad \text{whenever } \lambda_1 \leq \lambda_2,$$

(which implies $\mathbf{Var}(\lambda_1) \geq \mathbf{Var}(\lambda_2)$ by the proof in Lemma A.1). Hence for $\lambda \leq \lambda_{\text{balance}}$, we have

$$\mathbb{E}\left[\|\hat{f}_\lambda - f_\rho^*\|_{L^2}^2 \mid \mathbf{X}\right] \geq \mathbf{Var}(\lambda) \geq \mathbf{Var}(\lambda_{\text{balance}}) \asymp \mathbb{E}\left[\|\hat{f}_{\lambda_{\text{balance}}} - f_\rho^*\|_{L^2}^2 \mid \mathbf{X}\right].$$

Combining the two sides shows that $\lambda_{\text{balance}}$ attains (up to constants) the minimal possible conditional mean-squared error among all admissible regularization parameters, that is, it is the optimal choice and the rates given in the theorem are sharp.

This completes the proof. □

### A.4.3 PROOF OF THEOREM 4.10

*Proof of Theorem 4.10.* We follow the steps above: (i) locate the balancing index $l_{\text{balance}}$ (so $\lambda_{\text{balance}} = d^{-l_{\text{balance}}}$); (ii) verify all hypotheses of Theorem 4.10 at $l = l_{\text{balance}}$ (including the extra approximation condition involving the $L^\infty$-norm of $f_\lambda$).

We use Lemma A.12 and Lemma A.13 for the asymptotic orders of $\mathcal{N}_1, \mathcal{N}_2, \mathcal{R}_2$. The $L^\infty$-bound for $f_\lambda$ is provided by Lemma A.14.

**Step 1. Determination of $l_{\text{balance}}$.** Write $\lambda = d^{-l}$ with $0 < l < \gamma$. As in the main text, separate three regimes indexed by $p \in \{0, 1, 2, \dots\}$:

- If $l \in \left(p, \ p + \frac{s}{2}\right]$, then (Lemma A.12, Lemma A.13)

$$\frac{\mathcal{N}_2(\lambda)}{n} \asymp d^{p-\gamma}, \qquad \mathcal{R}_2(\lambda) \asymp d^{-2l+(2-s)p}.$$

  Balancing variance and bias gives

$$d^{p-\gamma} \asymp d^{-2l+(2-s)p} \implies l_{\text{balance}} = \frac{\gamma + p - ps}{2},$$

  and the admissible $\gamma$-range is $\gamma \in \left(p + ps, \ p + ps + s\right]$.

- If $l \in \left(p + \frac{s}{2}, \ p + \frac{1}{2}\right]$, then

$$\frac{\mathcal{N}_2(\lambda)}{n} \asymp d^{p-\gamma}, \qquad \mathcal{R}_2(\lambda) \asymp d^{-(p+1)s},$$

  which is balanced if and only if $\gamma = p + ps + s$.

- If $l \in \left(p + \frac{1}{2}, \ p + 1\right]$, then

$$\frac{\mathcal{N}_2(\lambda)}{n} \asymp d^{2l-p-1-\gamma}, \qquad \mathcal{R}_2(\lambda) \asymp d^{-(p+1)s},$$

  hence

$$l'_{\text{balance}} = \frac{\gamma + (p+1)(1-s)}{2}, \qquad \gamma \in \left(p + ps + s, \ (p+1) + (p+1)s\right].$$

Notice that the second circumstance is covered by the first one, hence there are only two intervals of $\gamma$. When $\gamma \in \left(p + ps + s, \ (p+1) + (p+1)s\right]$, by choosing $l = p + \frac{s}{2}$, the convergence rate remains the same. Hence, we substitute $l'_{\text{balance}} = \frac{\gamma + (p+1)(1-s)}{2}$ with $l_{\text{balance}} = p + \frac{s}{2}$. From now on $l_{\text{balance}}$ denotes the formula appropriate to the $\gamma$-regime above.

**Step 2. Verification of the hypotheses of Theorem 3.5.** Theorem 3.5 requires, for the chosen $\lambda$, the four standard conditions equation 7

$$\frac{\mathcal{N}_1(\lambda)}{n} \ln n = o(1), \qquad \frac{1}{n\lambda} = o(1), \qquad \mathcal{N}_2(\lambda) = \Omega(1), \qquad \frac{\ln n}{n\lambda^2} = o\big(\mathcal{N}_2(\lambda)\big), \quad (69)$$

together with the extra approximation condition equation 8: there exists $\varepsilon > 0$ such that

$$\frac{\sqrt{\frac{1}{\lambda}}(n^{\frac{1-s}{2}+\epsilon} + \|f_\lambda\|_{L^\infty})}{n} = o\big(\frac{1}{\sqrt{n}}\mathcal{N}_2(\lambda)^{\frac{1}{2}} + \mathcal{R}_2(\lambda)^{\frac{1}{2}}\big) \quad (70)$$

We check equation 69 and equation 70 at $\lambda = \lambda_{\text{balance}}$. (Because each of the left-hand quantities in equation 69 and the left-hand side of equation 70 is non-decreasing in $l$ when $\lambda = d^{-l}$, verification at $l_{\text{balance}}$ implies the conditions hold for all $l \leq l_{\text{balance}}$.)

The verification of the four relations in equation 69 is identical to the checks carried out in the proof of Theorem 4.8 (replace $\tilde{s}$ by $s$ here). It remains to verify the extra condition equation 70. By Lemma C.5 we have the explicit bound

$$\|f_\lambda\|_{L^\infty} = O\Big(d^{\frac{(1-s)p}{2}} + \lambda^{-1}d^{-\frac{(1+s)(p+1)}{2}}\Big), \qquad \text{whenever } p \leq l < p+1, \quad (71)$$

Set $n \asymp d^\gamma$, $\lambda = d^{-l_{\text{balance}}}$, using equation 71 and noticing that $\gamma \geq p$, we have

$$LHS \lesssim d^{l_{\text{balance}}/2 - \gamma}(d^{(\frac{1-s}{2} + \epsilon)\gamma} + d^{l_{\text{balance}} - \frac{(1+s)(p+1)}{2}}) \tag{72}$$

On the right-hand side of equation 70 we have

$$\frac{1}{\sqrt{n}}\mathcal{N}_2(\lambda)^{1/2} \asymp d^{(p-\gamma)/2} + d^{(2l_{\text{balance}} - p - 1 - \gamma)/2}, \qquad \mathcal{R}_2(\lambda)^{1/2} \asymp d^{-l_{\text{balance}} + \frac{(2-s)p}{2}},$$

so

$$\text{RHS} \asymp \max\{d^{(p-\gamma)/2}, d^{(2l_{\text{balance}} - p - 1 - \gamma)/2}, d^{-l_{\text{balance}} + \frac{(2-s)p}{2}}\}.$$

*Case A:* $l_{\text{balance}} = \dfrac{\gamma + p - ps}{2}$ *(so $l \in (p, p + \frac{1}{2}]$, $\gamma \in (p + ps, p + ps + s]$).* By simple calculation, we may find that condition equation 8 is equivalent to

$$(1 - 2s)\gamma < p(s + 1),$$

which naturally holds when $s > \frac{1}{2}$. When $s \leq \frac{1}{2}$, $(1 - 2s)\gamma < p(s + 1)$ only holds for $p > 0$. Hence, we are not able to give the convergence rate when $\gamma < \frac{1}{2}$.

*Case B:* $l_{\text{balance}} = p + \frac{s}{2}$ *(so $\gamma \in (p + ps + s, (p + 1) + (p + 1)s])$.*

By simple calculation, we may find that condition equation 8 is equivalent to

$$\gamma > \frac{2p + 3s + 2ps}{2(s + 1)}.$$

When $s > \frac{1}{2}$, we may find that the condition holds naturally for $p > 0$. When $p = 0$, the extra condition equation 70 can also be verified.

When $s \leq \frac{1}{2}$, we may find that the condition only holds for $p > 0$. Hence, we are not able to give the convergence rate when $\gamma \leq \frac{3s}{2(s+1)}$.

In summary, by the explicit bound of Lemma C.5 for $\|f_\lambda\|_{L^\infty}$ (used in equation 71) and the exponent comparisons above, the extra approximation condition equation 70 required by Theorem 4.10 is satisfied at $\lambda = \lambda_{\text{balance}}$ in every admissible $\gamma$-regime. Together with the four conditions equation 69 this verifies all hypotheses of Theorem 4.10 at $l = l_{\text{balance}}$.

This completes the proof. $\qquad\square$

### A.5 PROOF OF THEOREM 4.13

#### A.5.1 MORE PRELIMINARIES ABOUT MINIMAX LOWER BOUND

We shall first introduce several concepts about minimax lower bound which can be frequently found in related literature Yang & Barron (1999); Lu et al. (2023), etc..

Suppose that $(\mathbf{Z}, d)$ is a topological space with a compatible loss function $d$, which are mappings from $\mathbf{Z} \times \mathbf{Z}$ to $\mathbb{R}_{\geq 0}$ with $d(f, f) = 0$ and $d(f, f') > 0$ for $f \neq f'$. We call such a loss function a *distance*. We introduce the packing entropy and covering entropy below:

**Definition A.15** (Packing entropy). A finite set $N_\epsilon \subset \mathbf{Z}$ is said to be an $\epsilon$-packing set in $\mathbf{Z}$ with separation $\epsilon > 0$, if for any $f, f' \in N_\epsilon, f \neq f'$, we have $d(f, f') > \epsilon$. The logarithm of the maximum cardinality of $\epsilon$-packing set is called the $\epsilon$-packing entropy of $\mathbf{Z}$ with distance $d$ and is denoted by $M_d(\epsilon, \mathbf{Z})$.

**Definition A.16** (Covering entropy). A set $G_\epsilon \subset \mathbf{Z}$ is said to be an $\epsilon$-net for $\mathbf{Z}$ if for any $\tilde{f} \in \mathbf{Z}$, there exists an $f_0 \in G_\epsilon$ such that $d(\tilde{f}, f_0) \leq \epsilon$. The logarithm of the minimum cardinality of $\epsilon$-net is called the $\epsilon$-covering entropy of $\mathbf{Z}$ with distance $d$ and is denoted by $V_d(\epsilon, \mathbf{Z})$.

Let $\mathcal{B} = \{f \in \mathcal{H}, \|f\|_{\mathcal{H}} \leq R\}$, where $R$ is the constant from Assumption 3.3. Without loss of generality, we can consider $\mathcal{B}$ be the unit ball in $\mathcal{H}$. Let $M_2(\epsilon, \mathcal{B})$ be the $\epsilon$-packing entropy of

$(\mathcal{B}, d^2 = \|\cdot\|_{L^2}^2)$ and $V_2(\epsilon, \mathcal{B})$ be the $\epsilon$-covering entropy of $(\mathcal{B}, d^2 = \|\cdot\|_{L^2}^2)$. Recalling that $\mu$ is the marginal distribution on $\mathcal{X}$, we further define

$$\mathcal{D} = \left\{ \rho_f \;\middle|\; \text{joint distribution of } (y, x) \text{ where } x \sim \mu, y = f(x) + \epsilon, \epsilon \sim N(0, \sigma_\epsilon^2), f \in \mathcal{B} \right\},$$

and let $V_K(\epsilon, \mathcal{D})$ be the $\epsilon$-covering entropy of $(\mathcal{D}, d^2 = \text{ KL divergence })$. It is easy to see that $\mathcal{D}$ is an subset of $\mathcal{P}$ which is defined in Theorem 4.13, i.e., $\mathcal{D} \subset \mathcal{P}$.

The following lemmas give useful characterizations of $M_2(\epsilon, \mathcal{B})$, $V_2(\epsilon, \mathcal{B})$ and $V_K(\epsilon, \mathcal{D})$. We refer to Lemma A.5, Lemma A.7 and Lemma A.8 in Lu et al. (2023) for their proofs.

**Lemma A.17.** *For any $\epsilon > 0$, we have $M_2(2\epsilon, \mathcal{B}) \leq V_2(\epsilon, \mathcal{B}) \leq M_2(\epsilon, \mathcal{B})$.*

**Lemma A.18.** $V_2(\epsilon, \mathcal{B}) = V_K\left(\frac{\epsilon}{\sqrt{2}\sigma_\epsilon}, \mathcal{D}\right).$

**Lemma A.19.** *Let $\{\lambda_j\}_{j=1}^\infty$ be the eigenvalues of $\mathcal{H}$. For any $\epsilon > 0$, let $K(\epsilon) = \frac{1}{2} \sum\limits_{j:\lambda_j > \epsilon^2} \ln\left(\lambda_j / \epsilon^2\right)$. We have*

$$V_2(6\epsilon, \mathcal{B}) \leq K(\epsilon) \leq V_2(\epsilon, \mathcal{B}). \tag{73}$$

The following important lemma is a modification of Theorem 1 and Corollary 1 in Yang & Barron (1999). We refer to Lemma 4.1 in Lu et al. (2023) for the proof.

**Lemma A.20.** *Let $\mathfrak{c} \in (0, 1)$ be a constant only depending on $c_1$, $c_2$, and $\gamma$, where $c_1, c_2$ are the constants given in Theorem 4.13. For any $0 < \tilde{\epsilon}_1, \tilde{\epsilon}_2 < \infty$ only depending on $n$, $d$, $\{\lambda_j\}$, $c_1$, $c_2$, and $\gamma$ and satisfying*

$$\frac{V_K(\tilde{\epsilon}_2, \mathcal{D}) + n\tilde{\epsilon}_2^2 + \ln 2}{V_2(\tilde{\epsilon}_1, \mathcal{B})} \leq \mathfrak{c}, \tag{74}$$

*we have*

$$\min_{\hat{f}} \max_{\rho_{f^*} \in \mathcal{D}} \mathbb{E}_{(\mathcal{X}, \mathbf{y}) \sim \rho_{f^*}^{\otimes n}} \left\| \hat{f} - f^* \right\|_{L^2}^2 \geq \frac{1 - \mathfrak{c}}{4} \tilde{\epsilon}_1^2. \tag{75}$$

### A.5.2 Proof of Theorem 4.13

Now we are ready to use the lemmas in the last subsection to prove Theorem 4.13. The proof is divided into two parts, dealing with the two cases of the interval in which $\gamma$ falls into. Notice that the proof is adapted from Zhang et al. (2024)

*Proof of Theorem 4.13 (i).* In this case, we assume $\gamma \in (p + ps, p + ps + s]$ for some integer $p \geq 0$. Let $\tilde{\epsilon}_2^2 = C_2 d^{-(\gamma - p)}$, in which we will choose the constant $C_2$ later. Note that $\gamma - p \in (ps, (p+1)s]$. Lemma A.10 implies that there exists $C_2$ only depending on $p$ such that for any sufficiently large $d$, we have

$$\mu_{p+1}^s < \tilde{\epsilon}_2^2 < \mu_p^s. \tag{76}$$

Next we choose $\tilde{\epsilon}_1^2 = d^{-(\gamma - p + \epsilon)}, \epsilon > 0$. Since $\gamma - p + \epsilon > ps$, when $d \geq \mathfrak{C}$, we have

$$\tilde{\epsilon}_1^2 < \mu_p^s. \tag{77}$$

Hence, using Lemma A.19 and Lemma A.10, for any $d \geq \mathfrak{C}$, we have

$$V_2(\tilde{\epsilon}_1, \mathcal{B}) \geq K(\tilde{\epsilon}_1) \tag{78}$$

$$\geq \frac{1}{2} N(d, p) \ln\left(\frac{\mu_p^s}{\tilde{\epsilon}_1^2}\right)$$

$$\geq \frac{1}{2} N(d, p) \ln\left(\frac{\mathfrak{C}_1 d^{-ps}}{d^{-(\gamma - p + \epsilon)}}\right)$$

$$= \frac{1}{2} N(d, p) \left(\ln \mathfrak{C}_1 + (\gamma - ps - p + \epsilon) \ln d\right). \tag{79}$$

In addition, using Lemma A.10, we have the following claim.

*Claim* A.21. Suppose that $\gamma \in (p + ps, p + ps + s]$ for some integer $p \geq 0$. Let $\tilde{\epsilon}_2^2$ be defined as above. For any $\epsilon_0 > 0$, there exists a sufficiently large constant $\mathfrak{C}$ such that for any $d \geq \mathfrak{C}$, we have

$$K\left(\sqrt{2}\sigma\tilde{\epsilon}_2/6\right) \leq (1 + \epsilon_0)\frac{1}{2}N(d, p)\ln\left(\frac{18\mu_p^s}{\tilde{\epsilon}_2^2\sigma^2}\right).$$

Therefore, for any $d \geq \mathfrak{C}$, we have

$$V_K(\tilde{\epsilon}_2, \mathcal{D}) = V_2\left(\sqrt{2}\sigma\tilde{\epsilon}_2, \mathcal{B}\right) \tag{80}$$

$$\leq K\left(\frac{\sqrt{2}\sigma\tilde{\epsilon}_2}{6}\right)$$

$$\leq (1 + \epsilon_0)\frac{1}{2}N(d, p)\ln\left(\frac{18\mu_p^s}{\tilde{\epsilon}_2^2\sigma^2}\right)$$

$$\leq (1 + \epsilon_0)\frac{1}{2}N(d, p)\ln\left(\frac{18\mathfrak{C}_2 d^{-ps}}{C_2\sigma^2 d^{-(\gamma-p)}}\right)$$

$$= (1 + \epsilon_0)\frac{1}{2}N(d, p)\left(\ln\frac{18\mathfrak{C}_2}{C_2\sigma^2} + (\gamma - ps - p)\ln d\right), \tag{81}$$

where we use Lemma A.18 and Lemma A.19 for the first line and use Lemma A.10 for the third line.

Using equation 78 and equation 80, recalling that $c_1 d^\gamma \leq n \leq c_2 d^\gamma$, we have

$$\frac{V_K(\tilde{\epsilon}_2, \mathcal{D}) + n\tilde{\epsilon}_2^2 + \ln 2}{V_2(\tilde{\epsilon}_1, \mathcal{B})} \leq \frac{(1 + \epsilon_0)\frac{1}{2}N(d, p)\left(\ln\frac{18\mathfrak{C}_2}{C_2\sigma^2} + (\gamma - ps - p)\ln d\right) + c_2 d^\gamma \cdot C_2 d^{-(\gamma-p)} + \ln 2}{\frac{1}{2}N(d, p)(\ln\mathfrak{C}_1 + (\gamma - ps - p + \epsilon)\ln d)}. \tag{82}$$

The dominant terms in equation 82 are:

$$\frac{\frac{1}{2}(1 + \epsilon_0)(\gamma - ps - p)N(d, p)\ln d}{\frac{1}{2}(\gamma - p + \epsilon - ps)N(d, p)\ln d}. \tag{83}$$

Hence, for any $\epsilon > 0$, we can choose $\epsilon_0$ small enough such that

$$\frac{V_K(\tilde{\epsilon}_2, \mathcal{D}) + n\tilde{\epsilon}_2^2 + \ln 2}{V_2(\tilde{\epsilon}_1, \mathcal{B})} \leq 82 := \mathfrak{c} < 1. \tag{84}$$

Then using Lemma A.20, we have

$$\min_{\hat{f}}\max_{\rho_{f^*}\in\mathcal{D}}\mathbb{E}_{(x,\mathbf{y})\sim\rho_{f^*}^{\otimes n}}\left\|\hat{f} - f^*\right\|_{L^2}^2 \geq \frac{1 - \mathfrak{c}}{4}\tilde{\epsilon}_1^2 = \frac{1 - \mathfrak{c}}{4}d^{-(\gamma-p-\epsilon)}.$$

Further recalling that $\mathcal{D} \subset \mathcal{P}$, we have

$$\min_{\hat{f}}\max_{\rho\in\mathcal{P}}\mathbb{E}_{(x,\mathbf{y})\sim\rho^{\otimes n}}\left\|\hat{f} - f_\rho^*\right\|_{L^2}^2 \geq \min_{\hat{f}}\max_{\rho_{f^*}\in\mathcal{D}}\mathbb{E}_{(x,\mathbf{y})\sim\rho_{f^*}^{\otimes n}}\left\|\hat{f} - f^*\right\|_{L^2}^2 \geq \frac{1 - \mathfrak{c}}{4}d^{-(\gamma-p-\epsilon)}. \tag{85}$$

We finish the proof of Theorem 4.13 (i).

*Proof of Theorem 4.13 (ii).* We assume $\gamma \in (p + ps + s, (p + 1) + (p + 1)s]$ for some integer $p \geq 0$. Let $\tilde{\epsilon}_2^2 = C_2 d^{-(p+1)s}\ln d$, where we shall choose the constant $C_2$ later. Lemma A.10 implies that there exists a constant $\mathfrak{C}$ such that for any $d \geq \mathfrak{C}$, we have

$$\mu_{p+1}^s < \tilde{\epsilon}_2^2 < \mu_p^s. \tag{86}$$

Let $\tilde{\epsilon}_1^2 = C_1 d^{-(p+1)s}$. Using Lemma A.10, we can choose $C_1 < \mathfrak{C}_1^s$, where $\mathfrak{C}_1$ is the constant in Lemma A.10, such that for any $d \geq \mathfrak{C}$, where $\mathfrak{C}$ is a constant only depending on $s$ and $p$, we have

$$\tilde{\epsilon}_1^2 < \mu_{p+1}^s. \tag{87}$$

Therefore, by Lemma A.19 and Lemma A.10, for any $d \geq \mathfrak{C}$, we have

$$V_2\left(\tilde{\epsilon}_1, \mathcal{B}\right) \geq K(\tilde{\epsilon}_1) \tag{88}$$

$$\geq \frac{1}{2}N(d, p+1)\ln\left(\frac{\mu_{p+1}^s}{\tilde{\epsilon}_1^2}\right)$$

$$\geq \frac{1}{2}N(d, p+1)\ln\left(\frac{\mathfrak{C}_1 d^{-(p+1)s}}{C_1 d^{-(p+1)s}}\right)$$

$$= \frac{1}{2}N(d, p+1)\ln\frac{\mathfrak{C}_1}{C_1}. \tag{89}$$

In addition, using Lemma A.10, we have the following claim.

*Claim* A.22. Suppose that $\gamma \in (p + ps + s, (p+1) + (p+1)s]$ for some integer $p \geq 0$. Let $\tilde{\epsilon}_2^2$ be defined as above. For any $\epsilon_0 > 0$, there exists a sufficiently large constant $\mathfrak{C}$ only depending on $s, p$ and $\epsilon_0$, such that for any $d \geq \mathfrak{C}$, we have

$$K\left(\sqrt{2}\sigma\tilde{\epsilon}_2/6\right) \leq (1 + \epsilon_0)\frac{1}{2}N(d, p)\ln\left(\frac{18\mu_p^s}{\tilde{\epsilon}_2^2\sigma^2}\right).$$

Therefore, for any $d \geq \mathfrak{C}$, where $\mathfrak{C}$ is a constant only depending on $s, p$ and $\{a_j\}_{j \leq p+1}$, we have

$$V_K\left(\tilde{\epsilon}_2, \mathcal{D}\right) = V_2\left(\sqrt{2}\sigma\tilde{\epsilon}_2, \mathcal{B}\right) \tag{90}$$

$$\leq K\left(\frac{\sqrt{2}\sigma\tilde{\epsilon}_2}{6}\right)$$

$$\leq (1 + \epsilon_0)\frac{1}{2}N(d, p)\ln\left(\frac{18\mu_p^s}{\tilde{\epsilon}_2^2\sigma^2}\right)$$

$$\leq (1 + \epsilon_0)\frac{1}{2}N(d, p)\ln\left(\frac{18\mathfrak{C}_2 d^{-ps}}{C_2\sigma^2 d^{-(p+1)s}}\right)$$

$$= (1 + \epsilon_0)\frac{1}{2}N(d, p)\left(\ln\frac{18\mathfrak{C}_2}{C_2\sigma^2} + s\ln d\right), \tag{91}$$

where we use Lemma A.18 and Lemma A.19 for the first line and Lemma A.10 for the third line.

Using equation 78 and equation 80, also recalling that we assume $c_1 d^\gamma \leq n \leq c_2 d^\gamma$, we have

$$\frac{V_K(\tilde{\epsilon}_2, \mathcal{D}) + n\tilde{\epsilon}_2^2 + \ln 2}{V_2(\tilde{\epsilon}_1, \mathcal{B})} \leq \frac{(1 + \epsilon_0)\frac{1}{2}N(d, p)\left(\ln\frac{18\mathfrak{C}_2}{C_2\sigma^2} + s\ln d\right) + c_2 d^\gamma \cdot C_2 d^{-(p+1)s} + \ln 2}{\frac{1}{2}N(d, p+1)\ln\frac{\mathfrak{C}_1}{C_1}}. \tag{92}$$

The dominant terms in equation 92 are:

$$\frac{c_2 C_2 d^{\gamma - (p+1)s}}{\frac{1}{2}\ln\frac{\mathfrak{C}_1}{C_1}N(d, p+1)\ln d}. \tag{93}$$

Further noticing that $\gamma - (p+1)s \leq p + 1$ for any $\gamma \in (p + ps + s, (p+1) + (p+1)s]$, so we can choose $C_2$ small enough and only depending on $s, \sigma, \gamma, \kappa, c_1, c_2$, such that

$$\frac{V_K(\tilde{\epsilon}_2, \mathcal{D}) + n\tilde{\epsilon}_2^2 + \ln 2}{V_2(\tilde{\epsilon}_1, \mathcal{B})} \leq 92 := \mathfrak{c} < 1. \tag{94}$$

Then using Lemma A.20 again, we have

$$\min_{\hat{f}} \max_{\rho_{f^*} \in \mathcal{D}} \mathbb{E}_{(x,\mathbf{y}) \sim \rho_{f^*}^{\otimes n}} \left\|\hat{f} - f^*\right\|_{L^2}^2 \geq \frac{1 - \mathfrak{c}}{4}\tilde{\epsilon}_1^2 = \frac{1 - \mathfrak{c}}{4}C_1 d^{-(p+1)s}.$$

Further recalling that $\mathcal{D} \subset \mathcal{P}$, we have

$$\min_{\hat{f}} \max_{\rho \in \mathcal{P}} \mathbb{E}_{(x,\mathbf{y}) \sim \rho^{\otimes n}} \left\|\hat{f} - f_\rho^*\right\|_{L^2}^2 \geq \min_{\hat{f}} \max_{\rho_{f^*} \in \mathcal{D}} \mathbb{E}_{(x,\mathbf{y}) \sim \rho_{f^*}^{\otimes n}} \left\|\hat{f} - f^*\right\|_{L^2}^2 \geq \frac{1 - \mathfrak{c}}{4}C_1 d^{-(p+1)s}. \tag{95}$$

We finish the proof of Theorem 4.13 (ii).

## A.6 EXPERIMENTS OF LARGE DIMENSIONAL GAUSSIAN KERNEL

We consider the distribution as $N(0, I_d)$, $k_d(x, x') = \exp(-\|x - x'\|_2^2/(2\ell^2 d))$ for some positive constant $\ell$ and the following two cases of true functions $f_\rho^*$:

(i) $f_\rho^* \in \mathcal{H}$. Following the setting of Lu et al. (2023), we choose $f_\rho^*(x) = k_d(x, x_1) + k_d(x, x_2) + k_d(x, x_3)$, where $x_1, x_2, x_3$ are chosen randomly from $N(0, I_d)$.

(ii) $f_\rho^* \in [\mathcal{H}]^s$, $s$ can be arbitrarily large. We choose $f_\rho^*(x)$ as the first eigenfunction of the large dimensional Gaussian kernel. We may denote $f_\rho^* \in [\mathcal{H}]^\infty$ without causing ambiguity.

We consider the following regression model $y = f_\rho^*(x) + \epsilon$, where $\epsilon \sim N(0, \sigma^2 I_d)$, $\sigma = 0.1$. We draw $n = \lceil d^{1.5} \rceil$ samples from the model. The results are as follows.

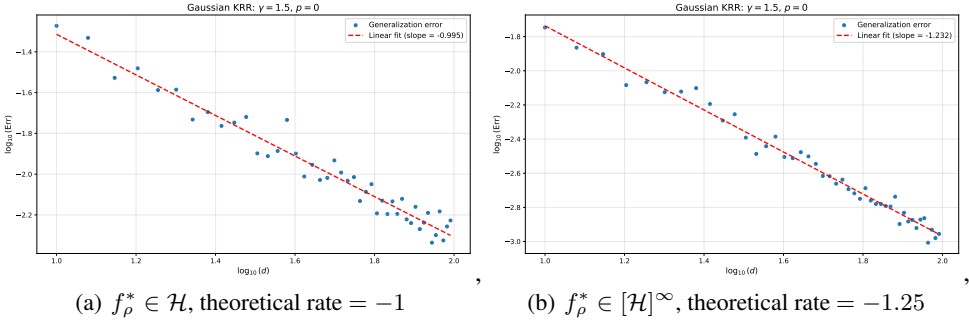

(a) $f_\rho^* \in \mathcal{H}$, theoretical rate $= -1$     (b) $f_\rho^* \in [\mathcal{H}]^\infty$, theoretical rate $= -1.25$

Figure 3: Generalization error of KRR under large dimensional Gaussian kernel. Each data point of generalization error is the average of 30 trials.

As is shown in Figure 3, the empirical convergence rates align closely with the theoretical predictions in both cases. It can be observed that the convergence in Figure 3(a) is slower than that in Figure 3(b), indicating that improved smoothness indeed enhances the rate of convergence. Moreover, the empirical rate in Figure 3(b) more closely approximates the theoretical exact rate of $-1.25$, rather than the minimax rate of $-1.5$, which suggests the presence of a saturation effect under this setting.

## A.7 ANOTHER APPLICABLE KERNEL

In this subsection, we shall provide another kernel to which our results are applicable.

Let $x, y \in \mathbb{R}^d$. Assume

$$x, y \sim N(0, \Sigma), \qquad \Sigma = \operatorname{diag}(\sigma_1^2, \ldots, \sigma_d^2), \quad \sigma_i > 0.$$

Define the anisotropic kernel with parameter $r$:

$$K_r(x, y) = \frac{1}{(1 - (\frac{r}{d})^2)^{d/2}} \exp\left( \frac{\sum_{i=1}^d \left( \frac{r}{d} \frac{x_i y_i}{\sigma_i^2} - \frac{1}{2}(\frac{r}{d})^2 \frac{x_i^2 + y_i^2}{\sigma_i^2} \right)}{1 - (\frac{r}{d})^2} \right).$$

First, we provide the following orthonormal basis:

Let the probabilists' Hermite polynomials be defined by

$$H_n(x) = (-1)^n e^{x^2/2} \frac{d^n}{dx^n} e^{-x^2/2}.$$

For a multi-index $\alpha = (\alpha_1, \ldots, \alpha_d) \in \mathbb{N}^d$, define

$$H_\alpha(x) = \prod_{i=1}^d H_{\alpha_i}\left(\frac{x_i}{\sigma_i}\right), \qquad \alpha! = \prod_{i=1}^d \alpha_i!, \qquad |\alpha| = \sum_{i=1}^d \alpha_i.$$

Then the orthonormal basis is

$$\psi_\alpha(x) = \frac{H_\alpha(x)}{\sqrt{\alpha!}} = \prod_{i=1}^{d} \frac{H_{\alpha_i}(x_i/\sigma_i)}{\sqrt{\alpha_i!}}.$$

We have the following Mercer's decomposition:

$$K_r(x,y) = \sum_{k=0}^{\infty} (\frac{r}{d})^k \sum_{|\alpha|=k} \psi_\alpha(x)\psi_\alpha(y),$$

which can be simply derived from one-dimensional Mehler's formula, and the multiplicity of eigenvalue $(\frac{r}{d})^k$ is

$$N(d,k) = \binom{d+k-1}{k}.$$

We may notice that the asymptotic properties of the eigenvalues of the above kernel are similar to the large dimensional Gaussian kernel. Since our main theorems (Theorem 3.4 and Theorem 3.5) only depends on eigenvalues, we can similarly apply our theorem to the above kernel. As a result, we may derive the similar exact rates and report the same saturation effects and multiple descent behavior. Also, by producing an orthogonal matrix to $x$, we actually provide a kind of kernel which can be applied to almost all general Gaussian distributions $N(\mu, \Sigma')$, where $\Sigma'$ need not be a diagonal matrix.

## A.8 AUXILIARY RESULTS

The following proposition about estimating the $L^2$ norm with empirical norm is from Li et al. (2023a, Proposition C.9), which dates back to Caponnetto & Yao (2010).

**Proposition A.23.** *Let $\mu$ be a probability measure on $\mathcal{X}$, $f \in L^2(\mathcal{X}, \mu)$ and $\|f\|_{L^\infty} \leq M$. Suppose we have $x_1, \ldots, x_n$ sampled i.i.d. from $\mu$. Then for $\delta \in (0,1)$, the following holds with probability at least $1 - \delta$:*

$$\frac{1}{2}\|f\|_{L^2}^2 - \frac{5M^2}{3n}\ln\frac{2}{\delta} \leq \|f\|_{L^2,n}^2 \leq \frac{3}{2}\|f\|_{L^2}^2 + \frac{5M^2}{3n}\ln\frac{2}{\delta}. \tag{96}$$

The following concentration inequality about self-adjoint Hilbert-Schmidt operator valued random variables is frequently used in related literature, e.g., Fischer & Steinwart (2020, Theorem 27) and Lin & Cevher (2020, Lemma 26).

**Lemma A.24.** *Let $(\Omega, \mathcal{B}, P)$ be a probability space, $\mathcal{H}$ be a separable Hilbert space. Suppose that $A_1, \cdots, A_n$ are i.i.d. random variables with values in the set of self-adjoint Hilbert-Schmidt operators. If $\mathbb{E}A_i = 0$, and the operator norm $\|A_i\| \leq L$, $P$-a.e., and there exists a self-adjoint positive semi-definite trace class operator $V$ with $\mathbb{E}A_i^2 \preceq V$. Then for $\delta \in (0,1)$, with probability at least $1 - \delta$, we have*

$$\left\|\frac{1}{n}\sum_{i=1}^{n} A_i\right\| \leq \frac{2L\beta}{3n} + \sqrt{\frac{2\|V\|\beta}{n}}, \quad \beta = \ln\frac{4\mathrm{tr}V}{\delta\|V\|}.$$

The following Bernstein inequality about vector-valued random variables is frequently used, e.g., Caponnetto & De Vito (2007, Proposition 2) and Fischer & Steinwart (2020, Theorem 26).

**Lemma A.25** (Bernstein inequality). *Let $(\Omega, \mathcal{B}, P)$ be a probability space, $H$ be a separable Hilbert space, and $\xi : \Omega \to H$ be a random variable with*

$$\mathbb{E}\|\xi\|_H^m \leq \frac{1}{2}m!\sigma^2 L^{m-2},$$

*for all $m > 2$. Then for $\delta \in (0,1)$, $\xi_i$ are i.i.d. random variables, with probability at least $1 - \delta$, we have*

$$\left\|\frac{1}{n}\sum_{i=1}^{n}\xi_i - \mathbb{E}\xi\right\|_H \leq 4\sqrt{2}\ln\frac{2}{\delta}\left(\frac{L}{n} + \frac{\sigma}{\sqrt{n}}\right).$$

**Lemma A.26.** *We have*

$$\|T_\lambda^{-\frac{1}{2}}k(x,\cdot)\|_\mathcal{H}^2 \le \frac{\kappa^2}{\lambda}, \quad \mu\text{-a.e. } x \in \mathcal{X}. \tag{97}$$

*Proof.*

$$\|T_\lambda^{-\frac{1}{2}}k(x,\cdot)\|_\mathcal{H}^2 = \Big\| \sum_{i \in N} (\frac{1}{\lambda_i + \lambda})^{\frac{1}{2}} \lambda_i e_i(x) e_i(\cdot) \Big\|_\mathcal{H}^2$$

$$= \sum_{i \in N} \frac{\lambda_i}{\lambda_i + \lambda} e_i^2(x)$$

$$\le \sum_{i \in N} \frac{1}{\lambda} \lambda_i e_i^2(x)$$

$$\le \frac{\kappa^2}{\lambda}, \quad \mu\text{-a.e. } x \in \mathcal{X}.$$

The last inequality is due to the fact that $\sum_{i \in N} \lambda_i e_i^2(x) = k(x,x) \le \sup_{x \in \mathcal{X}} k(x,x) \le \kappa^2$. $\qquad\square$

Lemma A.26 has a direct corollary.

**Lemma A.27.** *Given the definition of $\mathcal{N}_1(\lambda)$ as in equation 4. We have*

$$\|T_\lambda^{-\frac{1}{2}} T_x T_\lambda^{-\frac{1}{2}}\| \le \frac{\kappa^2}{\lambda}, \quad \mu\text{-a.e. } x \in \mathcal{X}.$$

*Proof.* Note that for any $f \in \mathcal{H}$,

$$T_\lambda^{-\frac{1}{2}} T_x T_\lambda^{-\frac{1}{2}} f = T_\lambda^{-\frac{1}{2}} K_x K_x^* T_\lambda^{-\frac{1}{2}} f$$

$$= T_\lambda^{-\frac{1}{2}} K_x \langle k(x,\cdot), T_\lambda^{-\frac{1}{2}} f \rangle_\mathcal{H}$$

$$= T_\lambda^{-\frac{1}{2}} K_x \langle T_\lambda^{-\frac{1}{2}} k(x,\cdot), f \rangle_\mathcal{H}$$

$$= \langle T_\lambda^{-\frac{1}{2}} k(x,\cdot), f \rangle_\mathcal{H} \cdot T_\lambda^{-\frac{1}{2}} k(x,\cdot).$$

So $\|T_\lambda^{-\frac{1}{2}} T_x T_\lambda^{-\frac{1}{2}}\| = \sup_{\|f\|_\mathcal{H}=1} \|T_\lambda^{-\frac{1}{2}} T_x T_\lambda^{-\frac{1}{2}} f\|_\mathcal{H} = \sup_{\|f\|_\mathcal{H}=1} \langle T_\lambda^{-\frac{1}{2}} k(x,\cdot), f \rangle_\mathcal{H} \cdot \|T_\lambda^{-\frac{1}{2}} k(x,\cdot)\|_\mathcal{H} = \|T_\lambda^{-\frac{1}{2}} k(x,\cdot)\|_\mathcal{H}^2$. Using Lemma A.26, we finish the proof. $\qquad\square$

The following theorem is borrowed from Zhang et al. (2024, Theorem 42). While they provided the proof of the theorem on compact set $\mathcal{X}$, we check their proof carefully and find that their proof can be easily extended to unbounded domains.

**Theorem A.28** ($L^q$-embedding property). *Suppose that $\mathcal{H}$ is the RKHS associated with a continuous, positive-definite and symmetric kernel $k$ on $\mathcal{X} \subset \mathbb{R}^d$ and the probability distribution on $\mathcal{X}$ is $\mu$. Further suppose that $\sup_{x \in \mathcal{X}} |k(x,x)| \le \kappa^2$, where $\kappa$ is an absolute constant. Then for any $0 < s < 1$, we have*

$$[\mathcal{H}]^s \hookrightarrow L^{q_s}(\mathcal{X}, \mu), \quad \forall q_s < \frac{2}{1-s}, \tag{98}$$

*and there exists a constant $C_{s,\kappa}$ only depending on $s$ and $\kappa$, such that the operator norm of the embedding operator satisfies*

$$\|[\mathcal{H}]^s \hookrightarrow L^{q_s}(\mathcal{X}, \mu)\| \le C_{s,\kappa}. \tag{99}$$

**Lemma A.29.** *Assume that $X \sim N(0, \sigma^2 I_d)$, when $d$ diverges to $\infty$, we have*

$$\mathbb{E}\left[\left\|\frac{X}{\sigma\sqrt{d}}\right\|_2^{d^\gamma}\right] \to \begin{cases} 1, & 0 < \gamma < \frac{1}{2} \\ e^{\frac{1}{4}}, & \gamma = \frac{1}{2} \\ \infty, & \gamma > \frac{1}{2} \end{cases},$$

*Proof.* Notice that $\|X\|_2^2 \sim \sigma^2 \mathcal{X}(d)$, hence we can derive that for all $k > 0$

$$\mathbb{E}\left[\|X\|_2^{2k}\right] = \sigma^{2k} 2^k \frac{\Gamma(d/2 + k)}{\Gamma(d/2)}.$$

Hence, we have

$$\mathbb{E}\left[\left\|\frac{X}{\sigma\sqrt{d}}\right\|_2^{d^\gamma}\right] = d^{-d^\gamma/2} 2^{d^\gamma/2} \frac{\Gamma(d/2 + d^\gamma/2)}{\Gamma(d/2)}$$

$$\asymp d^{-d^\gamma/2} 2^{d^\gamma/2} \frac{\sqrt{\frac{2\pi}{d/2+d^\gamma/2}} \left(\frac{d/2+d^\gamma/2}{e}\right)^{d/2+d^\gamma/2}}{\sqrt{\frac{2\pi}{d/2}} \left(\frac{d/2}{e}\right)^{d/2}} \qquad \text{(by Stirling formula)}$$

$$\asymp e^{-d^\gamma/2} (1 + d^{\gamma-1})^{d/2+d^\gamma/2-1/2}.$$

When $\gamma > 1$ or $\gamma = 1$, in both cases we have $\mathbb{E}\left[\left\|\frac{X}{\sigma\sqrt{d}}\right\|_2^{d^\gamma}\right] \to \infty$. When $\gamma < 1$, we have

$$\log\left(\mathbb{E}\left[\left\|\frac{X}{\sigma\sqrt{d}}\right\|_2^{d^\gamma}\right]\right) \asymp -d^\gamma/2 + (d/2 + d^\gamma/2 - 1/2)\log(1 + d^{\gamma-1})$$

$$\asymp -d^\gamma/2 + (d/2 + d^\gamma/2 - 1/2)(d^{\gamma-1} - \frac{1}{2}d^{2\gamma-2} + O(d^{3\gamma-3}))$$

$$\asymp d^{(2\gamma-1)}/4$$

Hence, we conclude that

$$\mathbb{E}\left[\left\|\frac{X}{\sigma\sqrt{d}}\right\|_2^{d^\gamma}\right] \to \begin{cases} 1, & 0 < \gamma < \frac{1}{2} \\ e^{\frac{1}{4}}, & \gamma = \frac{1}{2} \\ \infty, & \gamma > \frac{1}{2} \end{cases},$$

$\square$

