# OpenReview forum: "Exact Rates and Saturation Effect of Kernel Ridge Regression over Unbounded Input Space in Large Dimensions"
_ICLR.cc/2026/Conference — Submitted to ICLR 2026_

### Official Review · Reviewer_rgNk · 2025-10-17

**Soundness:** 3
**Presentation:** 4
**Contribution:** 2
**Rating:** 4
**Confidence:** 3

**Summary:**

The authors investigate the error rates of kernel ridge regression in the large dimensional, polynomial regime $n\sim d^{-\gamma}$, under a source condition on the target function. Similarly to previous works, they derive the exact convergence rates, and the minimax rates, and evidence a multiple descent behavior. The central technical contribution lies in the fact that the present work succeeds in prescinding the need of assuming bounded (typically spherical) data distribution.

**Strengths:**

The paper is very clearly written, and addresses a problem of interest. The technical advance allows in principle to lift a rather restrictive data assumption common to many prior studies, e.g. Xiao et al, 2022, Zhang et al, 2024, and could prove of interest to the community. I have however not carefully read the proofs, and have limited familiarity with the technical tools the paper uses.

**Weaknesses:**

My main concern is the close proximity of the paper with the reference Zhang et al, 2024. While the authors carefully explain the difference between the current theorem 3.4 and theorem 1 in the latter, such detailed comparison is lacking in the discussion of subsequent results. For instance, the exact rates (Theorems 4.8, 4.10) and minimax lower bounds (Theorem 4.13) all appear, to the best of my understanding, in  Zhang et al, 2024, and the expressions are identical. A discussion of why this is the case, and high-level intuition, would be helpful. By the same token, the phenomenology (plateaus and multiple descent) discussed from line 414 seem to be unaltered from the spherical data case of Zhang et al, 2024. Thus, while the technical contribution may be interesting, there are limited new results (both in terms of rate and phenomenology) compared to the spherical case. I believe more comparison should at least be included, and would be willing to increase my score if the authors include further detailed discussion on the differences to the spherical case.

**Questions:**

- The similarity of the phenomenology to the spherical case could stem from the assumptions of isotropic Gaussian data, which while unbounded, is intuitively very close to a spherical distribution in high dimensions. Would it be possible to illustrate the results on another distribution, or would this be technically out of reach ?
- To the best of my understanding, (Pandit et al, Universality of Kernel Random Matrices and Kernel Regression in the Quadratic Regime, 2024) already consider potentially unbounded data (e.g. Gaussian), for the quadratic regime $n\sim d^2$.
- "their applicability to the most commonly used kernel, the Gaussian kernel, remains unverified" (l.70). I think the phrasing is slightly confusing, as one could very well apply a Gaussian kernel on spherical data.
- Is the setting of section 4 not satisfying the conditions of applicability of Theorem 1 in Zhang et al, 2024? Despite the discussion around l. 224, I do not believe this question is addressed. I am curious to know whether something fundamentally breaks in the unbounded case, warranting Theorem 3.4, or if the latter is just easier to verify.

---

> ### Author Response · Authors · 2025-11-19
>
> **W1**: Thank you for your request for a detailed discussion between our results and the results of Zhang et al, 2024.
>
> + For your first question on the similarity of behaviors of the two kernels, we want to point out that the Mehler kernels on anisotropic Gaussian distribution $x\sim N(0,\Sigma)$ defined in Q1 also share the same saturation effects, periodic plateau phenomenon and multiple descent behavior. Actually, such phenomena can be reported for all kernels whose eigenvalues possess a **staircase descent** property (the eigenvalues $\mu _ k\asymp d^{-k}$, with $\mu _ k$'s multiplicity as $N(d,k)\asymp d^k$, which is the case of the above three kinds of kernels) since our Theorem 3.4, Theorem 3.5 only relies on the eigenvalues. We believe such phenomena are widespread in the large dimensional KRR, and our work may contribute to the further work on the universality of such phenomena.
>
> + For your request on a more detailed discussion about the difference between large dimensional Gaussian kernel and inner product kernels on sphere. Besides the differences already exhibited in Appendix A.1, we also provide the following structural properties unique to the RKHS of Gaussian kernels at the end of question (also added blue in Appendix A.1), a nested structure which can be useful when considering large dimensional sparsity problem. Recently, Li, et al. 2024, "Improving Adaptivity via Over-Parameterization in Sequence Models" asserts that **overparametrization leads to adaptivity to low dimensional structure** in the fixed dimensional setting. We believe the nested structure of the Gaussian RKHS provides a natural functional-analytic foundation for generalizing their work to the large dimensional setting.
>
>
>
> *Structural Differences Between the RKHSs*
>
> Another important difference lies in the structural properties of the RKHS. The RKHS of Gaussian kernel possesses a nested property, which can be described as follows. Let $\{i _ 1, \cdots, i _ p\}$ be a subset of $\{1, 2, \cdots, d\}$, and for any $u = (u _ 1, \cdots, u _ d)$, define its projection as $u' = (u _ {i _ 1}, \cdots, u _ {i _ p})$. Consider a  Gaussian kernel
> $$k(u,v)=\exp \left(-\frac{||u-v|| _ 2^2}{2\ell^2d}\right).$$
> By Proposition 4.4, if we set $u _ i = 0$ for all $i \notin \{i _ 1, i _ 2, \dots, i _ p\}$ for every element in the RKHS of $k$, the resulting RKHS naturally coincides with that of a lower-dimensional Gaussian kernel
> $$k(u',v')=\exp \left(-\frac{||u'-v'|| _ 2^2}{2\ell'^2p}\right)$$
> with an appropriate $\ell'$.
>
> This nested structure implies that if the target function  $f _ \rho^{\star}$ possesses a low dimensional structure, i.e., $f _ \rho^{\star}(u _ 1,\cdots,u _ d)=g(u _ {i _ 1},\cdots,u _ {i _ p})$, a low dimensional Gaussian kernel can be used to effectively approximate $f _ \rho^{\star}$. In contrast, the RKHS of inner-product kernels on the sphere generally lacks such nested RKHS properties. Therefore, a thorough understanding of large-dimensional Gaussian kernels is essential for analyzing such problems involving low-dimensional structure and sparsity.
>
>
> We hope the above discussion shall address your questions regarding the distinctions between Gaussian kernel and inner product kernels on sphere and the similarities of behavior. Please let us know if you have further questions, and it would be our great honor if you could consider raising your score should our response have addressed your concerns.

---

> ### Author Response · Authors · 2025-11-19
>
> **Q1**: Thank you for your concern whether our results can be applied to kernels on a broader class of distributions. Generally speaking, our results can be applied to all kernels with explicit eigenvalues, without requiring additional properties on the eigenfunctions. As an example, we shall provider the following kernel, whose distribution is anisotropic Gaussian distribution, and can be further extended to almost all general Gaussian distributions.
>
>
> Let $x,y\in\mathbb{R}^d$. Assume
> $$
> x,y\sim N(0,\Sigma),\qquad
> \Sigma=\operatorname{diag}(\sigma _ 1^2,\dots,\sigma _ d^2),\quad \sigma _ i>0.
> $$
> Define the anisotropic kernel with parameter $r$:
> $$
> K _ r(x,y)
> = \frac{1}{(1-(\frac{r}{d})^2)^{d/2}}
> \exp\left(
> \frac{\displaystyle \sum _ {i=1}^d \big( \frac{r}{d}\frac{x _ i y _ i}{\sigma _ i^2}
> -\tfrac{1}{2}(\frac{r}{d})^2\frac{x _ i^2+y _ i^2}{\sigma _ i^2}\big)}
> {1-(\frac{r}{d})^2}
> \right).
> $$
>
> First, we provide the following orthonormal basis:
>
> Let the probabilists' Hermite polynomials be defined by
> $$
> H _ n(x) = (-1)^n e^{x^2/2}\frac{d^n}{dx^n} e^{-x^2/2}.
> $$
> For a multi-index
> $\alpha=(\alpha _ 1,\dots,\alpha _ d)\in\mathbb{N}^d$, define
> $$
> H _ \alpha(x)=\prod _ {i=1}^d H _ {\alpha _ i}\Big(\frac{x _ i}{\sigma _ i}\Big),\qquad
> \alpha! = \prod _ {i=1}^d \alpha _ i!,\qquad |\alpha|=\sum _ {i=1}^d\alpha _ i.
> $$
> Then the orthonormal basis is
> $$
> \psi _ \alpha(x)=\frac{H _ \alpha(x)}{\sqrt{\alpha!}}
> = \prod _ {i=1}^d \frac{H _ {\alpha _ i}(x _ i/\sigma _ i)}{\sqrt{\alpha _ i!}}.
> $$
> We have the following Mercer's decomposition:
> $$
> K _ r(x,y)=\sum _ {k=0}^\infty (\frac{r}{d})^k
> \sum _ {|\alpha|=k}\psi _ \alpha(x)\psi _ \alpha(y),
> $$
> which can be simply derived from one-dimensional Mehler's formula, and the multiplicity of eigenvalue $(\frac{r}{d})^k$ is
> $$
> N(d,k)=\binom{d+k-1}{k}.
> $$
> We may notice that the asymptotic properties of the eigenvalues of the above kernel are similar to the large dimensional Gaussian kernel. Since our main theorems (Theorem 3.4 and Theorem 3.5) only depend on eigenvalues, we can similarly apply our theorem to the above kernel. As a result, we may derive the similar exact rates and report the same saturation effects, periodic plateau phenomenon and multiple descent behavior, which (we believe) are universal phenomena for large dimensional KRR. Also, by producing an orthogonal matrix to $x$, we actually provide a kind of kernel which can be applied to almost all general Gaussian distributions $N(\mu,\Sigma')$, where $\Sigma'$ need not be a diagonal matrix. We shall add the discussion of the above kernel in the appendix.
>
> **Q2**: Thank you for your requirement for a comparison of Pandit et al, Universality of Kernel Random Matrices and Kernel Regression in the Quadratic Regime, 2024. They considered the setting $n\asymp d^2$, and proved that inner product kernels can be approximated by quadratic kernels. We believe the work is insightful, and have properly cited the work in our discussion (in blue). Also, we believe that expanding the spherical data to potentially unbounded data, and ultimately deriving the universality of behaviors and properties of KRR (such as the approximation in Pandit et al. and the saturation effects, periodic plateau phenomenon and multiple descent behavior in our paper) is of great importance.
>
> **Q3**: Thank you for pointing out the phrasing problem in line 70. We have changed it into "their applicability to the most commonly used kernel, the Gaussian kernel under Gaussian measure, remains unverified".

---

> ### Author Response · Authors · 2025-11-19
>
> **Q4**: Thank you for your request for a more detailed discussion on the difficulty of applying Theorem 1 in Zhang et al, 2024 to the large dimensional Gaussian kernel. We shall provide the discussion as follows.
>
> + In Zhang et al, 2024, their proof strongly relies on the **properties of spherical harmonic functions.** Such properties are used to verify Assumption 3 in their work, which is $\underset{\boldsymbol{x} \in \mathcal{X}}{\operatorname{ess} \sup } \sum _ {i=1}^{\infty}\left(\frac{\lambda _ i}{\lambda _ i+\lambda}\right)^2 e _ i^2(\boldsymbol{x}) \leq \mathcal{N} _ 2(\lambda)$ and $\underset{\boldsymbol{x} \in \mathcal{X}}{\operatorname{ess} \sup } \sum _ {i=1}^{\infty} \frac{\lambda _ i}{\lambda _ i+\lambda} e _ i^2(\boldsymbol{x}) \leq \mathcal{N} _ 1(\lambda)$ ( see Lemma 20 in their work for details).
> However, such properties do not generally hold for other eigenfunctions. Especially, when we consider kernels with unbounded domains, a uniform upper bound for the eigenfunctions may not even exist. Therefore, the requirement in Assumption 3 that the weighted sum of eigenfunctions must be controlled by $\mathcal{N} _ 1(\lambda)$ and $\mathcal{N} _ 2(\lambda)$ becomes unreasonable.
>
> + In contrast, our results **only depend on eigenvalues**. This key difference allows our theorem to not only encompass inner product kernels on the sphere but also be applied to the large dimensional Gaussian kernel. Furthermore, our results can also be applied to the Mehler kernels on anisotropic Gaussian data in Q1, which can also not be easily fitted into the setting of Theorem 1 in Zhang et al, 2024.
> Also, since our results reveal that the performance of large dimensional KRR is only dependent to the eigenvalues, behaviors such as the saturation effects, periodic plateau phenomenon and multiple descent behavior exist for all kernels whose eigenvalues possess a **staircase descent** property (the eigenvalues $\mu _ k\asymp d^{-k}$, with $\mu _ k$'s multiplicity as $N(d,k)\asymp d^k$). We believe our approach can be extended to establish the universality of these phenomena in large dimensional KRR, thereby underscoring the broader significance of our work.

---

> > ### Comment · Reviewer_rgNk · 2025-11-24
> > **Acknowledgement of rebuttal**
> >
> > I thank the authors for the clarifications, notably the comparisons to related works.
> >
> > Regarding the additional example on an anisotropic Gaussian distribution, the kernel considered seems very ad-hoc and tailored to the anisotropy, rescaling the individual components of the samples by the corresponding eigenvalue, therefore largely reducing it back to the isotropic case.
> >
> > I would like to maintain my score and initial evaluation, but please consider it as a high 4/5, but with low confidence.

---

> > > ### Author Response · Authors · 2025-11-24
> > >
> > > Thank you for changing your assessment of our work. We fully agree that the examples we provided here are a little bit ad hoc, even though we could extend it to slightly more general setting $k_{\Sigma},   N(0,\Sigma)$ where $\Sigma$ is a general covariance matrix rather than diagonal matrix. We wish to emphasize, however, that the distributions used in these examples are distinctly different from the uniform distribution on the sphere. Consequently, the behaviors of the kernel we presented cannot be directly inferred from results applicable to inner product kernels on the sphere.
> > >
> > > We would like to point out, the field of large dimensional kernel regression lacks concrete, verifiable examples. Many theoretical works present seemingly general results that are difficult to apply to practical settings. For example, Assumption 1 in Misiakiewicz and Saeed "A non-asymptotic theory of Kernel Ridge Regression: deterministic equivalents, test error, and GCV estimator" can be challenging to verify even for inner product kernels on sphere without additional conditions.
> > >
> > > To advance the field of large dimensional kernel regression, we believe it is in the community's interest to identify more concrete Mercer's decompositions within high-dimensional settings. Such foundational work would facilitate the exploration of richer phenomena. For instance, by leveraging families of kernels and distributions like $(K_{\Sigma), N(0,\Sigma)}$, we hypothesize that a form of universality may exist in large dimensional regression for the periodic plateau phenomenon, multiple descent behavior, or saturation effects across a broad class of kernels.
> > >
> > > We thank again for your quick response and constructive suggestions.

---

> > > ### Author Response · Authors · 2025-11-25
> > > **Another example kernel on $(0,\infty)^d$**
> > >
> > > We provide the following kernel, whose domain lies on $(0,\infty) ^ d$ rather than $R ^ d$, to which our theorem can also be applied.
> > >
> > > **Theorem** Let $x,y\in(0,\infty)^d$. Assume that each coordinate is independently distributed as $x _ i,y _ i \sim \Gamma(\alpha _ i+1,1), \alpha _ i>-1,  i=1,\dots,d$ with density $\mathrm{d}\mu _ {\alpha _ i}(t)= \frac{1}{\Gamma(\alpha _ i+1)} t^{\alpha _ i} e^{- t}dt.$ Define the anisotropic Laguerre kernel with parameter $r\in(0,1)$:
> > > $$
> > > K _ r (x,y) = \prod _ {i=1}^d G _ {r,i}(x _ i,y _ i),
> > > $$
> > > where
> > > $$
> > > \begin{aligned}
> > > G _ {r,i}(x _ i,y _ i)
> > > &=
> > > \frac{\Gamma(\alpha _ i+1)}{1- \frac{r}{d}}
> > > \exp\left(-\frac{ \frac{r}{d}(x _ i+y _ i)}{1- \frac{r}{d}}\right)
> > > ( \frac{r}{d}\cdot x _ iy _ i)^{-\alpha _ i/2}
> > > I _ {\alpha _ i}\left(\frac{2\sqrt{ \frac{r}{d}x _ iy _ i}}{1- \frac{r}{d}}\right).
> > > \end{aligned}
> > > $$
> > > Here the modified Bessel function is defined by
> > > $$
> > > I _ {\alpha _ i}(z)=\sum _ {m=0}^{\infty}\frac{1}{m!\Gamma(m+\alpha _ i+1)}\left(\frac{z}{2}\right)^{2m+\alpha _ i}.
> > > $$
> > >
> > > Then the Mercer decomposition is
> > > $$
> > > K _ r (x,y)=\sum _ {k=0}^ \infty  \left(\frac{r}{d}\right)^{k}
> > > \sum _ {|\beta|=k}
> > > \widetilde{L} _ {\beta}^{(\boldsymbol{\alpha})}(x)
> > > \widetilde{L} _ {\beta}^{(\boldsymbol{\alpha})}(y),
> > > $$
> > > where
> > > $$
> > > \widetilde{L} _ {\beta}^{(\boldsymbol{\alpha})}(x)
> > > = \prod _ {i=1}^d \widetilde{L} _ {\beta _ i}^{(\alpha _ i)}(x _ i),
> > > \qquad
> > > \beta! = \prod _ {i=1}^d \beta _ i!,
> > > \qquad
> > > |\beta| = \sum _ {i=1}^d \beta _ i
> > > $$
> > > for  multi-index $\boldsymbol{\alpha}=(\alpha_1,\cdots,\alpha_d)$ and $\beta=(\beta _ 1,\dots,\beta _ d)\in\mathbb{N}^d$, and $\widetilde{L} _ {n}^{(\alpha _ i)}$ denote the orthonormal Laguerre polynomial in
> > > $L^2((0,\infty),\mu _ {\alpha _ i})$.
> > >
> > > The Mercer decomposition of the above kernel can be derived from the Laguerre formula (in the end of this comment). Notice that the multiplicity of $\left(\frac{r}{d}\right)^{k}$ is $N(d,k)=\binom{d+k-1}{k}.$ We can also derive the same phenomena as the Gaussian kernel and anisotropic Mehler kernel.
> > >
> > > We hope the above kernel defined on $(0,\infty)^d$ rather than $R^d$ may further eliminate your concern on the isotropy of the data. Please let us know if you have further questions.
> > >
> > > **Laguerre formula**
> > > $$
> > > \sum _ {n=0}^{\infty} r^n
> > > \widetilde{L} _ {n}^{(\alpha _ i)}(x)
> > > \widetilde{L} _ {n}^{(\alpha _ i)}(y)=\frac{\Gamma(\alpha _ i+1)}{1-r}
> > > \exp\left(-\frac{r(x+y)}{1-r}\right)
> > > (rxy)^{-\alpha _ i/2}
> > > I _ {\alpha _ i}\left(\frac{2\sqrt{rxy}}{1-r}\right).
> > > $$

---

### Official Review · Reviewer_TaT4 · 2025-10-21

**Soundness:** 4
**Presentation:** 3
**Contribution:** 3
**Rating:** 6
**Confidence:** 3

**Summary:**

The paper studies the generalization error of kernel ridge regression in high dimensions for general data distributions on unbounded domains and without assumptions of the eigenfunctions of the kernel. This is an improvement over prior works that consider restricted data distributions, such as the sphere or hypercube, and/or inner product and hypercontractivity conditions on the kernel function. They apply their general result to the specific case of the Gaussian kernel and Gaussian data. As a result, the authors show the Gaussian kernel achieves minimax optimality when the target is sharper than the RKHS. The authors also show a saturation effect where KRR fails to achieve the minimax rate and does not see additional improvements in generalization error with smoothness above that of the RKHS.

**Strengths:**

-	The paper studies fundamental theoretical questions in machine learning: (1) How do non-parametric estimators generalize in high dimensions? What is the interaction of the smoothness of the target function with that of the function class used to estimate it?
-	The paper establishes satisfying results in these questions that seem to contribute substantially over prior works.

**Weaknesses:**

-	The main weakness is I would like to see some experimental evidence of the saturation effect for the setting considered in this work – Gaussian kernel and Gaussian data. In particular, it could be quite nice to have some plots illustrating why additional smoothness in the target function does not benefit the target estimate.

**Questions:**

-	Do we see saturation effects for non-Gaussian kernels? I suspect this may be a general phenomenon for kernel ridge regression.
-	Is there some intuition for why the saturation effect occurs? My feeling is if you know the target function is smoother than the minimum allowed by your function class, your function space is ‘too wide’ and you can truncate it without losing generalization error. It could be nice to see experimentally that trimming your function space improves error.

---

> ### Author Response · Authors · 2025-11-19
>
> **W1**: Thank you for your request of adding some experiments illustrating the saturation effect. We have added an experiment part in Section A.6 (in blue for your convenience). By considering the first eigenfunction of the Gaussian kernel (whose smoothness $s$ can be arbitrarily large), we derive a convergence rate of -1.231 under the case $n=\lceil d^{1.5}\rceil$, which is closer to the theoretical convergence rate -1.25 rather than the minimax rate -1.5 (see Figure 3(b) in the revised manuscript for details). We believe the above experiment is able to verify the saturation effect of KRR in the large dimensional setting.
>
> **Q1**: We strongly agree that the saturation effect is a general phenomenon for KRR. In fixed dimensional setting, saturation effects have been reported for polynomially eigenvalue decaying kernels ($\lambda _ i\asymp i^{-\beta}, \beta>1$) when $s>2$. While under the large dimensional setting, since our results reveal that the performance of KRR is only dependent to the eigenvalues, saturation effects exist for all kernels whose eigenvalues possess a **staircase descent** property (the eigenvalues $\mu _ k\asymp d^{-k}$, with $\mu _ k$'s multiplicity as $N(d,k)\asymp d^k$). Such properties hold for a large group of kernels, such as inner product kernels on sphere, large dimensional Gaussian kernels and a group of Mehler kernels on anisotropic Gaussian data which will be provided at the end of this question. We believe our approach can be extended to establish the universality of these phenomena in large dimensional KRR, thereby underscoring the broader significance of our work.
>
> We now provide the following kernel on anisotropic Gaussian data:
>
> Let $x,y\in\mathbb{R}^d$. Assume
> $$
> x,y\sim N(0,\Sigma),\qquad
> \Sigma=\operatorname{diag}(\sigma _ 1^2,\dots,\sigma _ d^2),\quad \sigma _ i>0.
> $$
> Define the anisotropic kernel with parameter $r$:
> $$
> K _ r(x,y)
> = \frac{1}{(1-(\frac{r}{d})^2)^{d/2}}
> \exp\left(
> \frac{\displaystyle \sum _ {i=1}^d \big( \frac{r}{d}\frac{x _ i y _ i}{\sigma _ i^2}
> -\tfrac{1}{2}(\frac{r}{d})^2\frac{x _ i^2+y _ i^2}{\sigma _ i^2}\big)}
> {1-(\frac{r}{d})^2}
> \right).
> $$
>
> First, we provide the following orthonormal basis:
>
> Let the probabilists' Hermite polynomials be defined by
> $$
> H _ n(x) = (-1)^n e^{x^2/2}\frac{d^n}{dx^n} e^{-x^2/2}.
> $$
> For a multi-index
> $\alpha=(\alpha _ 1,\dots,\alpha _ d)\in\mathbb{N}^d$, define
> $$
> H _ \alpha(x)=\prod _ {i=1}^d H _ {\alpha _ i}\Big(\frac{x _ i}{\sigma _ i}\Big),\qquad
> \alpha! = \prod _ {i=1}^d \alpha _ i!,\qquad |\alpha|=\sum _ {i=1}^d\alpha _ i.
> $$
> Then the orthonormal basis is
> $$
> \psi _ \alpha(x)=\frac{H _ \alpha(x)}{\sqrt{\alpha!}}
> = \prod _ {i=1}^d \frac{H _ {\alpha _ i}(x _ i/\sigma _ i)}{\sqrt{\alpha _ i!}}.
> $$
> We have the following Mercer's decomposition:
> $$
> K _ r(x,y)=\sum _ {k=0}^\infty (\frac{r}{d})^k
> \sum _ {|\alpha|=k}\psi _ \alpha(x)\psi _ \alpha(y),
> $$
> which can be simply derived from one-dimensional Mehler's formula, and the multiplicity of eigenvalue $(\frac{r}{d})^k$ is
> $$
> N(d,k)=\binom{d+k-1}{k}.
> $$
> We may notice that the asymptotic properties of the eigenvalues of the above kernel are similar to the large dimensional Gaussian kernel. Since our main theorems (Theorem 3.4 and Theorem 3.5) only depend on eigenvalues, we can similarly apply our theorem to the above kernel. As a result, we may derive the similar exact rates and report the same saturation effects and multiple descent behavior, which (we believe) are universal phenomena for large dimensional KRR. Also, by producing an orthogonal matrix to $x$, **we actually provide a kind of kernel which can be applied to almost all general Gaussian distributions $N(\mu,\Sigma')$, where $\Sigma'$ need not be a diagonal matrix.** We shall add the discussion of the above kernel in the appendix.
>
> **Mehler's formula**
> $$
> \sum _ {n=0}^{\infty} r^n \frac{H _ n(x/\sigma)H _ n(y/\sigma)}{n!}
> = \frac{1}{\sqrt{1-r^2}}
> \exp\left(
> \frac{rxy/\sigma^2 - \tfrac{1}{2}r^2(x^2+y^2)/\sigma^2}{1-r^2}
> \right),
> \quad |r|<1.
> $$

---

> ### Author Response · Authors · 2025-11-19
>
> **Q2**: We strongly agree with you that there should be some underlying reasons for the saturation effects.
>
> + In the **fixed dimensional** setting, saturation effects occur when $s > 2$. In this regime, the generalization error of KRR decays as $n^{-\frac{2\beta}{2\beta+1}}$, which is slower than the minimax optimal rate of $n^{-\frac{s\beta}{s\beta+1}}$ (where $\beta$ denotes the eigenvalue decay rate, $\lambda _ i \asymp i^{-\beta}$). This phenomenon occurs because the bias term remains $\Omega(\lambda^2)$, regardless of the smoothness $s$ of the target function when $s > 2$. The inability of the estimator to leverage additional smoothness beyond this point is the essence of the saturation effect.
>
> + In the **large dimensional** setting, it is more complicated. As is shown in Proposition 4.7, $\mathcal{R} _ 2(\lambda)$ (which represents the bias term) satisfies $\mathcal{R} _ {2}(\lambda) = \Theta(\lambda^2d^{(2-\tilde{s})p}+d^{-(p+1)\tilde{s}})$. As a result, due to the bias-variance tradeoff, the saturation effect arises when $s>1$. A particularly tractable case is $s > 2$, where $\mathcal{R} _ 2(\lambda)\asymp  \lambda^2$, implying that bias term remains the same no matter how smooth the target function is, which consequently leads to the saturation effect.
>
>
> For your second question, we are not sure if we have understood you clearly. To our understanding, you want some experimental results to illustrate that added target function smoothness improves the generalization error when not encountering saturation effects. As a result, we provide a target function whose smoothness is $s=1$. Our Figure 3(a) shows that the convergence rate is -0.995, which is (i) close to the theoretical result, (ii) slower than the convergence rate in Figure 3(b).

---

> ### Author Response · Authors · 2025-11-25
> **Another example kernel on $(0,\infty)^d$**
>
> We provide the following kernel, whose domain lies on $(0,\infty) ^ d$ rather than $R ^ d$, to which our theorem can also be applied.
>
> **Theorem** Let $x,y\in(0,\infty)^d$. Assume that each coordinate is independently distributed as $x _ i,y _ i \sim \Gamma(\alpha _ i+1,1), \alpha _ i>-1,  i=1,\dots,d$ with density $\mathrm{d}\mu _ {\alpha _ i}(t)= \frac{1}{\Gamma(\alpha _ i+1)} t^{\alpha _ i} e^{- t}dt.$ Define the anisotropic Laguerre kernel with parameter $r\in(0,1)$:
> $$
> K _ r (x,y) = \prod _ {i=1}^d G _ {r,i}(x _ i,y _ i),
> $$
> where
> $$
> \begin{aligned}
> G _ {r,i}(x _ i,y _ i)
> &=
> \frac{\Gamma(\alpha _ i+1)}{1- \frac{r}{d}}
> \exp\left(-\frac{ \frac{r}{d}(x _ i+y _ i)}{1- \frac{r}{d}}\right)
> ( \frac{r}{d}\cdot x _ iy _ i)^{-\alpha _ i/2}
> I _ {\alpha _ i}\left(\frac{2\sqrt{ \frac{r}{d}x _ iy _ i}}{1- \frac{r}{d}}\right).
> \end{aligned}
> $$
> Here the modified Bessel function is defined by
> $$
> I _ {\alpha _ i}(z)=\sum _ {m=0}^{\infty}\frac{1}{m!\Gamma(m+\alpha _ i+1)}\left(\frac{z}{2}\right)^{2m+\alpha _ i}.
> $$
>
> Then the Mercer decomposition is
> $$
> K _ r (x,y)=\sum _ {k=0}^ \infty  \left(\frac{r}{d}\right)^{k}
> \sum _ {|\beta|=k}
> \widetilde{L} _ {\beta}^{(\boldsymbol{\alpha})}(x)
> \widetilde{L} _ {\beta}^{(\boldsymbol{\alpha})}(y),
> $$
> where
> $$
> \widetilde{L} _ {\beta}^{(\boldsymbol{\alpha})}(x)
> = \prod _ {i=1}^d \widetilde{L} _ {\beta _ i}^{(\alpha _ i)}(x _ i),
> \qquad
> \beta! = \prod _ {i=1}^d \beta _ i!,
> \qquad
> |\beta| = \sum _ {i=1}^d \beta _ i
> $$
> for  multi-index $\boldsymbol{\alpha}=(\alpha_1,\cdots,\alpha_d)$ and $\beta=(\beta _ 1,\dots,\beta _ d)\in\mathbb{N}^d$, and $\widetilde{L} _ {n}^{(\alpha _ i)}$ denote the orthonormal Laguerre polynomial in
> $L^2((0,\infty),\mu _ {\alpha _ i})$.
>
> The Mercer decomposition of the above kernel can be derived from the Laguerre formula (in the end of this comment). Notice that the multiplicity of $\left(\frac{r}{d}\right)^{k}$ is $N(d,k)=\binom{d+k-1}{k}.$ We can also derive the same phenomena as the Gaussian kernel and anisotropic Mehler kernel.
>
> **Laguerre formula**
> $$
> \sum _ {n=0}^{\infty} r^n
> \widetilde{L} _ {n}^{(\alpha _ i)}(x)
> \widetilde{L} _ {n}^{(\alpha _ i)}(y)=\frac{\Gamma(\alpha _ i+1)}{1-r}
> \exp\left(-\frac{r(x+y)}{1-r}\right)
> (rxy)^{-\alpha _ i/2}
> I _ {\alpha _ i}\left(\frac{2\sqrt{rxy}}{1-r}\right).
> $$

---

### Official Review · Reviewer_SvDk · 2025-10-23

**Soundness:** 4
**Presentation:** 4
**Contribution:** 3
**Rating:** 8
**Confidence:** 4

**Summary:**

The paper first derives excess risk upper bounds for KRR on general domains with bounded kernels, expressed cleanly in terms of the kernel spectrum and source conditions. It then specializes to the high-dimensional Gaussian kernel, where it characterizes the saturation effect and documents the periodic plateau behavior of the convergence rates.

**Strengths:**

This paper presents a clear exposition with rigorous, carefully organized proofs. It extends prior high-dimensional KRR results from spherical inner-product kernels to kernels on general domains—a theoretically significant contribution that broadens applicability.

**Weaknesses:**

The only weakness of this paper is that it provides concrete high-dimensional rate calculations only for the Gaussian kernel.

**Questions:**

If the kernel is not Gaussian, will the periodic plateau rate curve and the source condition threshold for the saturation effect change?

---

> ### Author Response · Authors · 2025-11-19
>
> **W1**: Thank you for your approval of the importance of our paper. For your request for more kernels to which our results are applicable, we shall provide the following kernel:
>
> Let $x,y\in\mathbb{R}^d$. Assume
> $$
> x,y\sim N(0,\Sigma),\qquad
> \Sigma=\operatorname{diag}(\sigma _ 1^2,\dots,\sigma _ d^2),\quad \sigma _ i>0.
> $$
> Define the anisotropic kernel with parameter $r$:
> $$
> K _ r(x,y)
> = \frac{1}{(1-(\frac{r}{d})^2)^{d/2}}
> \exp\left(
> \frac{\displaystyle \sum _ {i=1}^d \big( \frac{r}{d}\frac{x _ i y _ i}{\sigma _ i^2}
> -\tfrac{1}{2}(\frac{r}{d})^2\frac{x _ i^2+y _ i^2}{\sigma _ i^2}\big)}
> {1-(\frac{r}{d})^2}
> \right).
> $$
>
> First, we provide the following orthonormal basis:
>
> Let the probabilists' Hermite polynomials be defined by
> $$
> H _ n(x) = (-1)^n e^{x^2/2}\frac{d^n}{dx^n} e^{-x^2/2}.
> $$
> For a multi-index
> $\alpha=(\alpha _ 1,\dots,\alpha _ d)\in\mathbb{N}^d$, define
> $$
> H _ \alpha(x)=\prod _ {i=1}^d H _ {\alpha _ i}\Big(\frac{x _ i}{\sigma _ i}\Big),\qquad
> \alpha! = \prod _ {i=1}^d \alpha _ i!,\qquad |\alpha|=\sum _ {i=1}^d\alpha _ i.
> $$
> Then the orthonormal basis is
> $$
> \psi _ \alpha(x)=\frac{H _ \alpha(x)}{\sqrt{\alpha!}}
> = \prod _ {i=1}^d \frac{H _ {\alpha _ i}(x _ i/\sigma _ i)}{\sqrt{\alpha _ i!}}.
> $$
> We have the following Mercer's decomposition:
> $$
> K _ r(x,y)=\sum _ {k=0}^\infty (\frac{r}{d})^k
> \sum _ {|\alpha|=k}\psi _ \alpha(x)\psi _ \alpha(y),
> $$
> which can be simply derived from one-dimensional Mehler's formula, and the multiplicity of eigenvalue $(\frac{r}{d})^k$ is
> $$
> N(d,k)=\binom{d+k-1}{k}.
> $$
> We may notice that the asymptotic properties of the eigenvalues of the above kernel are similar to the large dimensional Gaussian kernel. Since our main theorems (Theorem 3.4 and Theorem 3.5) only depend on eigenvalues, we can similarly apply our theorem to the above kernel. As a result, we may derive the similar exact rates and report the same saturation effects and multiple descent behavior, which (we believe) are universal phenomena for large dimensional KRR. Also, by producing an orthogonal matrix to $x$, **we actually provide a kind of kernel which can be applied to almost all general Gaussian distributions $N(\mu,\Sigma')$, where $\Sigma'$ need not be a diagonal matrix.** We shall add the discussion of the above kernel in the appendix.
>
> **Mehler's formula**
> $$
> \sum _ {n=0}^{\infty} r^n \frac{H _ n(x/\sigma)H _ n(y/\sigma)}{n!}
> = \frac{1}{\sqrt{1-r^2}}
> \exp\left(
> \frac{rxy/\sigma^2 - \tfrac{1}{2}r^2(x^2+y^2)/\sigma^2}{1-r^2}
> \right),
> \quad |r|<1.
> $$
>
> **Q1**: Thank you for your question on the periodic plateau phenomenon and the saturation effect. As is indicated by Theorem 3.4, 3.5, the exact convergence rates are only determined by the eigenvalues of the kernel. As a result, we can report the periodic plateau phenomenon and  same saturation effects for kernels with **staircase descent** eigenvalues (the eigenvalues $\mu _ k\asymp d^{-k}$, with $\mu _ k$'s multiplicity as $N(d,k)\asymp d^k$), such as inner product kernels on sphere, Gaussian kernels and the Mehler kernels on anisotropic Gaussian data displayed in W1. We believe such phenomena are widespread in the large dimensional setting, and further work on the universality of such phenomena is promising.

---

> ### Author Response · Authors · 2025-11-25
> **Another example kernel on $(0,\infty)^d$**
>
> We provide the following kernel, whose domain lies on $(0,\infty) ^ d$ rather than $R ^ d$, to which our theorem can also be applied.
>
> **Theorem** Let $x,y\in(0,\infty)^d$. Assume that each coordinate is independently distributed as $x _ i,y _ i \sim \Gamma(\alpha _ i+1,1), \alpha _ i>-1,  i=1,\dots,d$ with density $\mathrm{d}\mu _ {\alpha _ i}(t)= \frac{1}{\Gamma(\alpha _ i+1)} t^{\alpha _ i} e^{- t}dt.$ Define the anisotropic Laguerre kernel with parameter $r\in(0,1)$:
> $$
> K _ r (x,y) = \prod _ {i=1}^d G _ {r,i}(x _ i,y _ i),
> $$
> where
> $$
> \begin{aligned}
> G _ {r,i}(x _ i,y _ i)
> &=
> \frac{\Gamma(\alpha _ i+1)}{1- \frac{r}{d}}
> \exp\left(-\frac{ \frac{r}{d}(x _ i+y _ i)}{1- \frac{r}{d}}\right)
> ( \frac{r}{d}\cdot x _ iy _ i)^{-\alpha _ i/2}
> I _ {\alpha _ i}\left(\frac{2\sqrt{ \frac{r}{d}x _ iy _ i}}{1- \frac{r}{d}}\right).
> \end{aligned}
> $$
> Here the modified Bessel function is defined by
> $$
> I _ {\alpha _ i}(z)=\sum _ {m=0}^{\infty}\frac{1}{m!\Gamma(m+\alpha _ i+1)}\left(\frac{z}{2}\right)^{2m+\alpha _ i}.
> $$
>
> Then the Mercer decomposition is
> $$
> K _ r (x,y)=\sum _ {k=0}^ \infty  \left(\frac{r}{d}\right)^{k}
> \sum _ {|\beta|=k}
> \widetilde{L} _ {\beta}^{(\boldsymbol{\alpha})}(x)
> \widetilde{L} _ {\beta}^{(\boldsymbol{\alpha})}(y),
> $$
> where
> $$
> \widetilde{L} _ {\beta}^{(\boldsymbol{\alpha})}(x)
> = \prod _ {i=1}^d \widetilde{L} _ {\beta _ i}^{(\alpha _ i)}(x _ i),
> \qquad
> \beta! = \prod _ {i=1}^d \beta _ i!,
> \qquad
> |\beta| = \sum _ {i=1}^d \beta _ i
> $$
> for  multi-index $\boldsymbol{\alpha}=(\alpha_1,\cdots,\alpha_d)$ and $\beta=(\beta _ 1,\dots,\beta _ d)\in\mathbb{N}^d$, and $\widetilde{L} _ {n}^{(\alpha _ i)}$ denote the orthonormal Laguerre polynomial in
> $L^2((0,\infty),\mu _ {\alpha _ i})$.
>
> The Mercer decomposition of the above kernel can be derived from the Laguerre formula (in the end of this comment). Notice that the multiplicity of $\left(\frac{r}{d}\right)^{k}$ is $N(d,k)=\binom{d+k-1}{k}.$ We can also derive the same phenomena as the Gaussian kernel and anisotropic Mehler kernel.
>
> **Laguerre formula**
> $$
> \sum _ {n=0}^{\infty} r^n
> \widetilde{L} _ {n}^{(\alpha _ i)}(x)
> \widetilde{L} _ {n}^{(\alpha _ i)}(y)=\frac{\Gamma(\alpha _ i+1)}{1-r}
> \exp\left(-\frac{r(x+y)}{1-r}\right)
> (rxy)^{-\alpha _ i/2}
> I _ {\alpha _ i}\left(\frac{2\sqrt{rxy}}{1-r}\right).
> $$

---

> > ### Comment · Reviewer_SvDk · 2025-11-27
> >
> > Thank you for the detailed rebuttal. I continue to hold a positive assessment.

---

> > > ### Author Response · Authors · 2025-11-27
> > >
> > > Thank you again for your important suggestions the universality of phenomena in the large dimensional setting and approving the importance of our work and contribution to the community. Please let us know if you have further questions.

---

### Official Review · Reviewer_sN5r · 2025-11-01

**Soundness:** 2
**Presentation:** 1
**Contribution:** 1
**Rating:** 2
**Confidence:** 5

**Summary:**

The paper considers the generalization properties of kernel ridge regression in increasing dimension. The main result is a generalization error of KRR that can be applied to a broad class of kernels (and not only inner product kernels on the sphere as some of the previous). Under source conditions on the target depending on $s$, they characterize the error of kernel ridge regression. This bound applied to the Gaussian kernel with Gaussian data matches the minmax lower bound for large-dimensional Gaussian kernel regression for $\le s<1$ but not $s>1$ and shows that this kernel exhibits the periodic plateau and multiple descent phenomena.

**Strengths:**

1. The topic of the main result in the paper is central to the understanding of kernel ridge regression, and it is especially interesting as it explores non-typical settings beyond the inner-product kernels on the sphere.
2. The bounds on the eigenvalues of Gaussian kernel with Gaussian data are interesting in their own right and can further the understanding of the generalization properties of KRR.

**Weaknesses:**

1. A related and important line of work is missing from the discussion on related works and the context of the paper’s result -  the work on eigenframework (e.g. Simon et al “The Eigenlearning Framework: A Conservation Law Perspective on Kernel Regression and Wide Neural Networks” and many others). Further, the work on the deterministic equivalent of test risk (e.g. Misiakiewicz and Saeed “A non-asymptotic theory of Kernel Ridge Regression: deterministic equivalents, test error, and GCV estimator”) is only briefly discussed because it puts explicit assumptions on kernel eigenfunctions. Both of these lines of work give estimates for the generalization error of KRR with a general input space $\mathcal X$ and kernels beyond the inner product kernels. In particular, both lines of work give estimates on the test risk of KRR in the same form as Theorem 3.4. For example, Thereom 1 in Misiakiewicz and Saeed and the main equation of eigenframework form Simon et al look very similar to Theorem 3.4. It is essential to properly discuss the similarities and differences between this work and existing work, especially when they are so closely related. Furthermore, both of these lines of work consider quantities that are more general to the ones in Definition 3.1 and Eq. (4). Namely, the regularization parameter $\lambda$ is replaced with “effective regularization”. This makes me wonder if the result in this paper misses a key detail of the behavior of KRR. Not discussing the eigenframework is especially alarming.
2. The motivation for considering a Gaussian kernel with $x\sim N(0,\sigma^2 I_d)$ is unclear, even with the remark on line 251. The reasoning there (and in Remark 4.1) is mostly focused on explaining why the proof techniques won’t work in this case. The more interesting question (to me at least) is whether the behavior of Gaussian Kernel with Gaussian data in large dimensions is in any way different from an inner product kernel on the sphere. The discussion in lines 414-423 does not discuss the differences in the behavior but (if I understand correctly) only the similarities (that the periodic plateau phenomenon and multiple descent behavior were shown in these other papers that “consider large dimensional spectral algorithms on the sphere”).
3. The limitations of the result could be expanded by a discussion of the phenomena the result does not capture. It is well known that KRR in increasing dimension n=d^{\gamma} can overfit benignly (e.g. Misiakiewicz 2022, Barzilai and Shamir 2023). The result in this paper does not discuss this regime, namely, the conditions on Eq (5) do not allow for setting $\lambda=0$. This is not the case for prior work (see weakness #1).

**Questions:**

1. In light of weakness #1, how is the main result of this paper (Theorem 3.4.) different/new compared to the already existing results on the deterministic equivalent of test error of KRR (Misiakiewicz and Saeed 2024, Theorem 1)? Are assumptions in Eq (5) not covered by the already existing results? If not, in what ways is it an improvement? Is it the avoidance of reliance on eigenfunctions? Because the result in Misiakiewicz and Saeed also does not rely on eigenfunctions.
2. Why are the $d^{\gamma}$-th moments of $X$ (scaled by $\sigma \sqrt{d}$) important to the generalization behavior of KRR (except in the particular proof approaches)? For large $d$, the distribution $x\sim N(0,\sigma^2 I_d)$ is close to $x\sim d*S^{d-1}$ because of CLT, and on $S^{d-1}$ the Gaussian kernel is an inner product kernel, so the Gaussian kernel considered here is also close to an inner product kernel up to scaling.
3. What other kernels except the Gaussian kernel with Gaussian data is the main result applicable to?
4. Is there a kernel to which the main result applies that is qualitatively different from inner product kernels on the sphere? If not, then what is the contribution of the result?

---

> ### Author Response · Authors · 2025-11-19
>
> **W1**: Thank you for your requirement for a more detailed discussion of related works. We shall provide a more detailed discussion as follows.
>
> *Related work on eigenframework*
>
> Simon et al. developed the eigenlearning framework, which expresses the generalization error of KRR in a fixed-dimensional regime through the notion of learnability and a scalar quantity $\kappa$ that solves a self-consistent implicit equation.
> Their main result (equation (9) in their paper) shows that the generalization error admits the decomposition
> $$
> \mathcal{E}(f)  =\frac{n}{n-\sum _ i \mathcal{L} _ i^2}\left(\sum _ i\left(1-\mathcal{L} _ i\right)^2 f _ i^2+\epsilon^2\right),
> $$
> where $\mathcal{L} _ i$ denotes the learnability of eigenmode $i$.
> The learnability $\mathcal{L} _ i=\frac{\lambda _ i}{\lambda _ i+\kappa}$ is further characterized by the effective regularization $\kappa>0$ via an implicit fixed-point equation of the form
> $$
> n=\sum _ i \frac{\lambda _ i}{\lambda _ i+\kappa}+n\frac{\lambda}{\kappa} ,
> $$
> where $\{\lambda _ i\}$ are the kernel eigenvalues. (A slight difference between the above equation and those in Simon et al. and Misiakiewicz and Saeed is due to the different settings of KRR considered. As a result, $\lambda _ {\text{theirs}} = n \lambda _ {\text{ours}}$.) Also, they obtained a closed form for $\kappa$ when $\lambda=0$ and provide several bounds under various assumptions. Although their results mainly lie on the fixed dimensional setting (which is not the case in ours), the eigenframework provides a valuable and insightful perspective on KRR, and we believe future work on the large dimensional setting is promising.
> **We have added a proper discussion of the eigenframework literature in the revision (highlighted in blue for convenience).**
>
> *Work on the deterministic equivalent of test risk*
>
> For Theorem 1 in Misiakiewicz and Saeed, they claimed that the generalization error can be approximated by a deterministic equivalent ($R _ n(\beta _ *,\lambda)$ in their work), which, at first glance, is similar to our Theorem 3.4 and 3.5. However, their result requires a crucial condition (Assumption 1 in their work), which assumes some concentration inequalities for the first few eigenfunctions and tail parts of target functions. Such concentration inequalities are difficult to verify  directly, which makes **a direct application of Theorem 1 in their work troublesome.**
>
> In order to make use of Theorem 1, **Misiakiewicz and Saeed considered the inner product kernels on sphere, and provided sufficient conditions for Assumption 1**: a hypercontractivity condition $\mathbb{E} _ {\boldsymbol{u}}\left[\left|f _ {\star}(\boldsymbol{u})\right|^q\right] \leq\left(\mathrm{C} _ L q\right)^{q L / 2}||f _ {\star}|| _ {L^2}^q$ for any integer $q\geq2$ and a heavy-tail condition $||\mathrm{P} _ {>L} f _ {\star}|| _ {L^2} \geq ||f _ {\star}|| _ {L^2} / \mathrm{C} _ L$. Here $L$ is an integer satisfying $n\leq d^L$. Such sufficient conditions are crucial to their proof and can not be easily removed. However, **the sufficient conditions do not hold for most of the $f _ {\star}$ when there exists $s>0$ such that $f _ {\star}\in [\mathcal{H}]^s$.** Notice that for large dimensional inner product kernels on sphere, the eigenvalues satisfy $\mu _ i\asymp d^{-i}$, with its multiplicity as $N(d,i)\asymp d^i$, and that $N(d,0)=1$. This implies that $||\mathrm{P} _ {>L} f _ {\star}|| _ {L^2} ^2=\sum _ {i>L}f _ i^2\leq\mu _ 1^{s}\sum _ {i>L}\mu _ i^{-s}f _ i^2\lesssim d^{-s}||f _ {\star}|| _ {[\mathcal{H}]^s}^2=o(1)$. Hence, the heavy-tail condition $||\mathrm{P} _ {>L} f _ {\star}|| _ {L^2} \geq ||f _ {\star}|| _ {L^2} / \mathrm{C} _ L$ only holds for $||f _ {\star}|| _ {L^2}=0$. Therefore, whether assumptions of Theorem 1 in Misiakiewicz and Saeed can be verified under the interpolation space setting remains an open question, even in the case of inner product kernels on sphere.
>
> On the contrary, our Theorem 3.4 and 3.5 can be easily applied to a large class of kernels including **Gaussian kernels on Gaussian measure, Mehler kernels on anisotropic Gaussian distribution $x\sim N(0,\Sigma)$ provided in Q3, and other kernels with staircase descent eigenvalues** under the interpolation space setting. As a result, we can also report similar phenomena such as saturation effects, periodic plateau phenomena and multiple descent behavior for this large class of kernels. **Such difference in the applicability under interpolation space setting makes our results different from Theorem 1 in Misiakiewicz and Saeed's work.**

---

> ### Author Response · Authors · 2025-11-19
>
> *Reasons for not considering effective generalization and whether details of the behavior are missed*
>
> + In response to your concern that the effective regularization is not used in our work, effective regularization $\kappa$ in their work is defined as the solution of
> $$
> n=\sum _ i \frac{\lambda _ i}{\lambda _ i+\kappa}+n
> \frac{\lambda}{\kappa} .
> $$
> By substituting $\lambda _ i$ by the eigenvalues of large dimensional Gaussian kernel (or other kernels such as inner product kernels on sphere, Mehler kernels on anisotropic Gaussian distribution $x\sim N(0,\Sigma)$ in Q3), we derive that $\kappa\asymp n^{-1}$ when $\lambda=o(n^{-1})$, and $\kappa\asymp \lambda$ when $\lambda=\Theta(n^{-1})$, which indicates that the **effective regularization is fully characterized by $n,\lambda$ in the large dimensional setting.**
>
> + For your doubt whether some behaviors are missed, if we understand correctly, your concern about whether our paper overlooks details of KRR behavior relates primarily to the benign overfitting phenomenon in W3. We wish to emphasize that **the primary objective of our work is to identify which kernels—beyond inner product kernels on sphere—exhibit characteristic behaviors such as saturation effects, periodic plateau phenomenon, and multiple descent behavior in the large dimensional setting.** Our findings indicate that Gaussian kernels under Gaussian measure, as well as the Mehler kernels on anisotropic Gaussian distribution $x\sim N(0,\Sigma)$ provided in Q3, also display these phenomena. More importantly, since our main results depend solely on eigenvalue properties, we can in fact extend these observations to all large dimensional kernels with **staircase descent** eigenvalues (the eigenvalues $\mu _ k\asymp d^{-k}$, with $\mu _ k$'s multiplicity as $N(d,k)\asymp d^k$). We believe such phenomena are widespread in the large dimensional setting, and further work on the universality of such phenomena is promising.
>
>     As for the **benign overfitting** behavior, in Zhang et al. 2024, "The phase diagram of kernel interpolation in large dimensions", a complete discussion about the benign overfitting behavior has been provided for inner product kernels on sphere. We believe similar studies on the benign overfitting behavior of other general kernels under the large dimensional setting are promising, but future effort might be needed. Also, we have added a part in the introduction about the benign overfitting behavior, and properly cited the paper you mentioned.

---

> ### Author Response · Authors · 2025-11-19
>
> **W2**: Our motivation for considering kernels other than inner product kernels on sphere consists of two aspects.
>
> + On the one hand, we are interested in **finding out what kernels except inner product kernels on sphere exhibit the saturation effects, periodic plateau phenomenon and multiple descent behavior**, since most former large dimensional papers consider only inner product kernels on sphere.
> As a result, our results imply that such phenomena widely exist in a large group of kernels, including **Gaussian kernel on Gaussian measure, Mehler kernels on anisotropic Gaussian distribution $x\sim N(0,\Sigma)$ provided in Q3, and other kernels with staircase descent eigenvalues**. We believe such phenomena are widespread in the large dimensional setting, and our work can be extended to establish such universality.
>
> + On the other hand, pursuing explicit understanding of kernels other than the inner product on sphere is motivated by a recent work which asserts that **overparametrization leads to adaptivity to low dimensional structure** (Li, et al. 2024, Improving Adaptivity via Over-Parameterization in Sequence Models), where the low dimensional structure cannot be simply defined for inner product kernel on spheres, which is not the case of Gaussian kernels.
> More precisely, let $\{i _ 1, \cdots, i _ p\}$ be a subset of $\{1, 2, \cdots, d\}$, and for any $u = (u _ 1, \cdots, u _ d)$, define its projection as $u' = (u _ {i _ 1}, \cdots, u _ {i _ p})$. Consider a  Gaussian kernel
> $k(u,v)=\exp \left(-\frac{||u-v|| _ 2^2}{2\ell^2d}\right).$
> By Proposition 4.4, if we set $u _ i = 0$ for all $i \notin \{i _ 1, i _ 2, \dots, i _ p\}$ for every element in the RKHS of $k$, the resulting RKHS naturally coincides with that of a lower-dimensional Gaussian kernel
> $k(u',v')=\exp \left(-\frac{||u'-v'|| _ 2^2}{2\ell'^2p}\right)$
> with an appropriate $\ell'$.
> This nested structure implies that if the target function  $f _ \rho^ {\star}$ possesses a low dimensional structure, i.e., $f _ \rho^{\star}(u _ 1,\cdots,u _ d)=g(u _ {i _ 1},\cdots,u _ {i _ p})$, a low dimensional Gaussian kernel can be used to effectively approximate $f _ \rho^{\star}$. In contrast, the RKHS of inner product kernels on the sphere generally lacks such nested RKHS properties. Therefore, a thorough understanding of large dimensional Gaussian kernels is essential for analyzing such problems involving low dimensional structure and sparsity.
>
> **W3**: We have to admit that we have not discussed the benign overfitting in this manuscript, as we planned to find more concrete kernels which we know the Mercer's decomposition exactly so that we could provide a characterization of benign overfitting as Zhang et al. 2024, "The phase diagram of kernel interpolation in large dimensions" did. Please allow us make few more words on our plans.
>
> + First, we want to point out that **our work mainly concerns about what kernels other than inner product kernels on sphere exhibit the saturation effects, periodic plateau phenomenon and multiple descent behavior.** As a result, our results imply that such phenomena widely exist in a large group of kernels, including Gaussian kernel on Gaussian measure, Mehler kernels on anisotropic Gaussian distribution $x\sim N(0,\Sigma)$ provided in Q3 and other kernels with staircase descent eigenvalues. We believe such phenomena are widespread in the large dimensional setting, and our work can be extended to establish such universality.
>
> + Second, for the **benign overfitting** behavior, in Zhang et al., 2024, "The phase diagram of kernel interpolation in large dimensions", a complete discussion about the benign overfitting behavior has been provided for inner product kernels on sphere. We believe similar studies on the benign overfitting behavior of other general kernels under the large dimensional setting are promising, but future effort might be needed. Also, we have added a part in the introduction about the benign overfitting behavior, and properly cited the paper you mentioned.

---

> ### Author Response · Authors · 2025-11-19
>
> **Q1**: For the difference between our work and Misiakiewicz and Saeed 2024, we want to point out that **the result in their work actually makes assumptions on eigenfunctions.** Assumption 1(a) in their work assumes  the concentration inequality of the first few eigenfunctions, which is hard to be verified directly, and the sufficient condition provided by them does not generally hold in the interpolation space setting even in the case of inner product kernels on sphere (see W1 for a detailed discussion).
>
> In contrast, our main results **depend solely on eigenvalue properties**, and hence can be applied to all large dimensional kernels with **staircase descent** eigenvalues (the eigenvalues $\mu _ k\asymp d^{-k}$, with $\mu _ k$'s multiplicity as $N(d,k)\asymp d^k$). As a result, saturation effects, periodic plateau phenomenon and multiple descent behavior all occur for such kernels. We believe these phenomena are widespread in the large dimensional setting, and further work on the universality of such phenomena is promising.
>
> **Q2**: Thank you for your question on the difference between Gaussian kernel and inner product kernels on sphere. Regarding your question on the importance of $d^{\gamma}$-th moments of $X$, we acknowledge that the distribution $x \sim N(0, \sigma^2 I _ d)$ is close to $x \sim d \cdot S^{d-1}$ due to the Central Limit Theorem. However, in the large dimensional setting where both $n$ and $d$ vary, it remains an open question whether $x \sim N(0, \sigma^2 I _ d)$ converges sufficiently quickly to $x \sim d \cdot S^{d-1}$ such that the performance of kernel ridge regression (KRR) remains unaffected. We introduce the $d^{\gamma}$-th moments to illustrate that the rate of this distributional convergence is strongly influenced by the dimension $d$, thereby emphasizing the importance of a separate analysis of large dimensional Gaussian kernel.
>
> Also, we shall provide **a Mehler kernel on anisotropic Gaussian distribution $x\sim N(0,\Sigma)$ in Q3** to eliminate your concern whether our results can only be applied to Gaussian kernel with isotropic Gaussian data.
>
> **Q3**: Thank you for your request for more kernels to which our main results are applicable. Generally speaking, our results are applicable to all kernels with given eigenvalues, without requiring additional properties on the eigenfunctions. As an another example, we shall provide the following kernel:
>
> Let $x,y\in\mathbb{R}^d$. Assume
> $$
> x,y\sim N(0,\Sigma),\qquad
> \Sigma=\operatorname{diag}(\sigma _ 1^2,\dots,\sigma _ d^2),\quad \sigma _ i>0.
> $$
> Define the anisotropic kernel with parameter $r$:
> $$
> K _ r(x,y)
> = \frac{1}{(1-(\frac{r}{d})^2)^{d/2}}
> \exp\left(
> \frac{\displaystyle \sum _ {i=1}^d \big( \frac{r}{d}\frac{x _ i y _ i}{\sigma _ i^2}
> -\tfrac{1}{2}(\frac{r}{d})^2\frac{x _ i^2+y _ i^2}{\sigma _ i^2}\big)}
> {1-(\frac{r}{d})^2}
> \right).
> $$
>
> First, we provide the following orthonormal basis:
>
> Let the probabilists' Hermite polynomials be defined by
> $$
> H _ n(x) = (-1)^n e^{x^2/2}\frac{d^n}{dx^n} e^{-x^2/2}.
> $$
> For a multi-index
> $\alpha=(\alpha _ 1,\dots,\alpha _ d)\in\mathbb{N}^d$, define
> $$
> H _ \alpha(x)=\prod _ {i=1}^d H _ {\alpha _ i}\Big(\frac{x _ i}{\sigma _ i}\Big),\qquad
> \alpha! = \prod _ {i=1}^d \alpha _ i!,\qquad |\alpha|=\sum _ {i=1}^d\alpha _ i.
> $$
> Then the orthonormal basis is
> $$
> \psi _ \alpha(x)=\frac{H _ \alpha(x)}{\sqrt{\alpha!}}
> = \prod _ {i=1}^d \frac{H _ {\alpha _ i}(x _ i/\sigma _ i)}{\sqrt{\alpha _ i!}}.
> $$
> We have the following Mercer's decomposition:
> $$
> K _ r(x,y)=\sum _ {k=0}^\infty (\frac{r}{d})^k
> \sum _ {|\alpha|=k}\psi _ \alpha(x)\psi _ \alpha(y),
> $$
> which can be simply derived from one-dimensional Mehler's formula, and the multiplicity of eigenvalue $(\frac{r}{d})^k$ is
> $$
> N(d,k)=\binom{d+k-1}{k}.
> $$
> We may notice that the asymptotic properties of the eigenvalues of the above kernel are similar to the large dimensional Gaussian kernel. Since our main theorems (Theorem 3.4 and Theorem 3.5) only depend on eigenvalues, we can similarly apply our theorem to the above kernel. As a result, we may derive the similar exact rates and report the same saturation effects and multiple descent behavior, which (we believe) are universal phenomena for large dimensional KRR. Also, by producing an orthogonal matrix to $x$, **we actually provide a kind of kernel which can be applied to almost all general Gaussian distributions $N(\mu,\Sigma')$, where $\Sigma'$ need not be a diagonal matrix.** We shall add the discussion of the above kernel in the appendix.
>
> **Mehler's formula**
> $$
> \sum _ {n=0}^{\infty} r^n \frac{H _ n(x/\sigma)H _ n(y/\sigma)}{n!}
> = \frac{1}{\sqrt{1-r^2}}
> \exp\left(
> \frac{rxy/\sigma^2 - \tfrac{1}{2}r^2(x^2+y^2)/\sigma^2}{1-r^2}
> \right),
> \quad |r|<1.
> $$

---

> ### Author Response · Authors · 2025-11-19
>
> **Q4**: Yes, we have provided a group of kernels in Q3 to which our main results are applicable, whose distribution is completely different from the sphere. For the contribution of our results, the only dependency on eigenvalues of the generalization error of KRR implies that saturation effects, periodic plateau phenomenon and multiple descent behavior exist for all kernels with staircase descent eigenvalues, indicating potential universal phenomena for large dimensional KRR. Please let us know if you have further questions, and it would be our great honor if you could consider raising your score should our response have addressed your concerns.

---

> ### Author Response · Authors · 2025-11-25
> **Another example kernel on $(0,\infty)^d$**
>
> We provide the following kernel, whose domain lies on $(0,\infty) ^ d$ rather than $R ^ d$, to which our theorem can also be applied.
>
> **Theorem** Let $x,y\in(0,\infty)^d$. Assume that each coordinate is independently distributed as $x _ i,y _ i \sim \Gamma(\alpha _ i+1,1), \alpha _ i>-1,  i=1,\dots,d$ with density $\mathrm{d}\mu _ {\alpha _ i}(t)= \frac{1}{\Gamma(\alpha _ i+1)} t^{\alpha _ i} e^{- t}dt.$ Define the anisotropic Laguerre kernel with parameter $r\in(0,1)$:
> $$
> K _ r (x,y) = \prod _ {i=1}^d G _ {r,i}(x _ i,y _ i),
> $$
> where
> $$
> \begin{aligned}
> G _ {r,i}(x _ i,y _ i)
> &=
> \frac{\Gamma(\alpha _ i+1)}{1- \frac{r}{d}}
> \exp\left(-\frac{ \frac{r}{d}(x _ i+y _ i)}{1- \frac{r}{d}}\right)
> ( \frac{r}{d}\cdot x _ iy _ i)^{-\alpha _ i/2}
> I _ {\alpha _ i}\left(\frac{2\sqrt{ \frac{r}{d}x _ iy _ i}}{1- \frac{r}{d}}\right).
> \end{aligned}
> $$
> Here the modified Bessel function is defined by
> $$
> I _ {\alpha _ i}(z)=\sum _ {m=0}^{\infty}\frac{1}{m!\Gamma(m+\alpha _ i+1)}\left(\frac{z}{2}\right)^{2m+\alpha _ i}.
> $$
>
> Then the Mercer decomposition is
> $$
> K _ r (x,y)=\sum _ {k=0}^ \infty  \left(\frac{r}{d}\right)^{k}
> \sum _ {|\beta|=k}
> \widetilde{L} _ {\beta}^{(\boldsymbol{\alpha})}(x)
> \widetilde{L} _ {\beta}^{(\boldsymbol{\alpha})}(y),
> $$
> where
> $$
> \widetilde{L} _ {\beta}^{(\boldsymbol{\alpha})}(x)
> = \prod _ {i=1}^d \widetilde{L} _ {\beta _ i}^{(\alpha _ i)}(x _ i),
> \qquad
> \beta! = \prod _ {i=1}^d \beta _ i!,
> \qquad
> |\beta| = \sum _ {i=1}^d \beta _ i
> $$
> for  multi-index $\boldsymbol{\alpha}=(\alpha_1,\cdots,\alpha_d)$ and $\beta=(\beta _ 1,\dots,\beta _ d)\in\mathbb{N}^d$, and $\widetilde{L} _ {n}^{(\alpha _ i)}$ denote the orthonormal Laguerre polynomial in
> $L^2((0,\infty),\mu _ {\alpha _ i})$.
>
> The Mercer decomposition of the above kernel can be derived from the Laguerre formula (in the end of this comment). Notice that the multiplicity of $\left(\frac{r}{d}\right)^{k}$ is $N(d,k)=\binom{d+k-1}{k}.$ We can also derive the same phenomena as the Gaussian kernel and anisotropic Mehler kernel.
>
> **Laguerre formula**
> $$
> \sum _ {n=0}^{\infty} r^n
> \widetilde{L} _ {n}^{(\alpha _ i)}(x)
> \widetilde{L} _ {n}^{(\alpha _ i)}(y)=\frac{\Gamma(\alpha _ i+1)}{1-r}
> \exp\left(-\frac{r(x+y)}{1-r}\right)
> (rxy)^{-\alpha _ i/2}
> I _ {\alpha _ i}\left(\frac{2\sqrt{rxy}}{1-r}\right).
> $$

---

### Author Response · Authors · 2025-12-01
**A Brief Summary of Our Contributions and Key Rebuttal Points**

We provide a brief summary of our contribution and important questions and responses during the rebuttal period. For a detailed discussion, please refer to the paper or rebuttal of the specific question.

*Summary of Contributions*

We provide the exact rates of generalization error of KRR with **unbounded domains** under the large dimensional setting, which **solely** depends on the eigenvalues rather than the eigenfunctions. As a result, we are able to report saturation effects, periodic plateau phenomenon and multiple descent behavior for **all** kernels with **staircase descent** eigenvalues (the eigenvalues $\mu _ k\asymp d^{-k}$, with $\mu _ k$'s multiplicity as $N(d,k)\asymp d^k$). Such property on eigenvalues holds for a large group of kernels, such as

+ inner product kernels on sphere,

+ Gaussian kernels on isotropic Gaussian measure,

+ Mehler kernels on general Gaussian measure (provided in  Section A.7),

+ Laguerre kernels on $(0,\infty)^d$ (provided at the end of rebuttal to each reviewers).

The broad applicability of our theory to the above kernel classes strongly suggests a **deeper universality on the phenomena of large dimensional KRR**, and our work provides a rigorous foundation for future research in this direction.

*Key Rebuttal Points*

+ **Detailed discussion of Simon et al., Misiakiewicz and Saeed. (W1,Q1 of reviewer sN5r)** For Simon et al., their eigenlearning framework lies on the **fixed dimension setting.** For Misiakiewicz and Saeed, their conditions on eigenfunctions are **difficult to verify in general cases**. Moreover, even for spherical inner product kernels, their sufficient conditions **generally fail under the interpolation space setting.**

+ **More differences between Gaussian kernels and inner product kernels on sphere (W2 of reviewer sN5r, W1 for reviewer rgNk)** We first emphasize that our results can be applied to a large group of kernels with explicit eigenvalues (inner product kernels on sphere, Gaussian kernels, Mehler kernels and Laguerre kernels,...), and Gaussian kernels are only one group of the applicable kernels. Second, besides the differences shown in Section A.1, we also provide a **key structural difference** between the RKHSs. The RKHS of Gaussian kernels possess a **nested structure**(also shown in Section A.1). This structure is particularly relevant for learning **target functions with low-dimensional intrinsic structure**, providing a foundation for analyzing adaptivity in large dimensions.

+ **Why this paper does not report benign overfitting behaviors** Our primary goal is to find out what kernels other than inner product kernels on sphere exhibit the saturation effects, periodic plateau phenomenon and multiple descent behavior, and **reveal the universality of such phenomena** in large dimensional KRR. For the benign overfitting behavior, in Zhang et al., 2024, "The phase diagram of kernel interpolation in large dimensions", a complete discussion about the benign overfitting behavior has been provided for inner product kernels on sphere. Extending this analysis to other kernels in the large dimensional setting is an important but separate direction for future work.

+ **Requests for more applicable kernels (Q3 of reviewer sN5r, W1 of reviewer SvDk, Q1 of reviewer rgNk)** We have provided a **Mehler kernel on general Gaussian distribution** in Section A.7 and responses to each reviewer, and a **Laguerre kernel on $(0,\infty)^d$** at the end of rebuttal to each reviewer, both confirming the wider applicability of our theory.

+ **Requests for experiments (W1,Q2 of reviewer TaT4)** We have added an experiment part in Section A.6, which explicitly shows the saturation effects of large dimensional KRR and how improved smoothness of target function affects the convergence rates.

+ **More discussion with Zhang et al., 2024, "Optimal rates of kernel ridge regression under source condition in large dimensions" (W1,Q4 of reviewer rgNk)** A core limitation of Zhang et al. is its reliance on eigenfunctions (Assumption 3), whose verification strongly relies on the **properties of spherical harmonic functions.** On the contrary, our results **solely** depend on eigenvalues. Such circumvention of eigenfunction assumption enables our framework to be applied to a far broader range of kernels (all kernels with staircase descent eigenvalues), and reveals the **universality of the reported phenomena** in large dimensional KRR.

---

### Meta-Review · Area_Chair_cLjz · 2025-12-21

**Summary:**

This paper established the convergence rate of kernel ridge regression without imposing any assumptions on eigenfunctions. It holds for KRR under unbounded domains in large dimension, polynomial regime $n \sim d^{\gamma}$. Similarly to previous works, the authors derive the exact convergence rates, and the minimax rates, and evidence a multiple descent behavior.

The review evaluation is quite mixed. Two reviewers argue the difference with previous work, espeically with (Zhang et al, 2024). The AC also checked the main result (e.g., Theorem 3.4) in details. After reading, I found that the proof of Theorem 3.4 is almost the same as previous work (Zhang et al. 2024). The authors claim the difference in Remark 3.7 on the eigenfunctions. However, the proof techniques are the same with an additional assumption to control the eigenfunctions in Eq. (22).

To be specific, for the bound of variance, it has two parts. One is for approximation A, with the proof from Li et al. (2023a). The authors clearly state it. However, for approximation B, when compared to (Zhang et al. 2024), the only difference is to bound $||f||_{L^{\inf}}$ by an additional assumption in line 760. In this case, the requirement on the eigenfunctions can be directly removed and thus we don't require it trivially.
Similarly, the proof for the bias is almost the same as  (Zhang et al. 2024) except for some minor modifications.

In this case, I don't think the contribution is sufficient to support this submission and recommend to reject this paper.

**Reviewer Concerns:**

Comparison with previous work, e.g., Theo's work is clearly stated.
However, the comparison with (Zhang et al. 2024) doesn't address the reviewers' concern.

**Reviewer Scores:**

Reviewer sN5r may increase the review score but would keep the negative recommendation of this paper.

Reviewer SvDk and Reviewer TaT4 will maintain their positive score.

Reviewer rgNk would maintain the evaluation as the rebuttal didn't address the concerns on the comparison with (Zhang et al. 2024).

---

### Decision · Program_Chairs · 2026-01-26

Reject